# ERA: Transforming VLMs into Embodied Agents via Embodied Prior Learning and Online Reinforcement Learning

## Abstract

Recent advances in embodied AI highlight the potential of vision language models (VLMs) as agents capable of perception, reasoning, and interaction in complex environments. However, top-performing systems rely on large-scale models that are costly to deploy, while smaller VLMs lack the necessary knowledge and skills to succeed. To bridge this gap, we present *Embodied Reasoning Agent (ERA)*, a two-stage framework that integrates prior knowledge learning and online reinforcement learning (RL). The first stage, *Embodied Prior Learning*, distills foundational knowledge from three types of data: (1) Trajectory-Augmented Priors, which enrich existing trajectory data with structured reasoning generated by stronger models; (2) Environment-Anchored Priors, which provide in-environment knowledge and grounding supervision; and (3) External Knowledge Priors, which transfer general knowledge from out-of-environment datasets. In the second stage, we develop an online RL pipeline that builds on these priors to further enhance agent performance. To overcome the inherent challenges in agent RL, including long horizons, sparse rewards, and training instability, we introduce three key designs: self-summarization for context management, dense reward shaping, and turn-level policy optimization. Extensive experiments on both high-level planning (EB-ALFRED) and low-level control (EB-Manipulation) tasks demonstrate that ERA-3B surpasses both prompting-based large models and previous training-based baselines. Specifically, it achieves overall improvements of 8.4% on EB-ALFRED and 19.4% on EB-Manipulation over GPT-4o, and exhibits strong generalization to unseen tasks. Detailed Ablation studies further validate the effectiveness of different prior datasets and agent RL designs. Overall, ERA offers a practical path toward scalable embodied intelligence, providing methodological insights for future embodied AI systems.

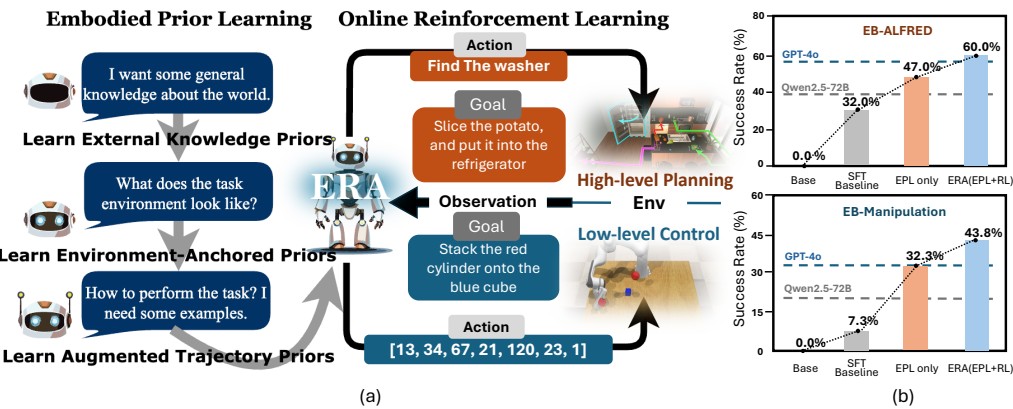

Figure 1: (a) Overview of the ERA framework: Embodied Prior Learning (EPL) finetunes on diverse data sources to provide foundational knowledge, and online RL further improves the agent. (b) ERA (i.e, EPL+RL) boosts a 3B base model to surpass GPT-4o on hold-out evaluation sets.

# 1 INTRODUCTION

Vision language models (VLMs) have shown remarkable capabilities in instruction following, visual understanding, and commonsense as well as mathematical reasoning (OpenAI, 2024; Liu et al., 2024a; Reid et al., 2024; Bai et al., 2025; Zhu et al., 2025). Building on these strengths, researchers are now exploring how to transform VLMs into embodied agents that can operate in interactive environments and tackle real-world tasks (Driess et al., 2023; Huang et al., 2023; 2024; Mu et al., 2024; Liu et al., 2024b; Kim et al., 2024b; Szot et al., 2025). Unlike single-turn question answering, embodied tasks require an agent to actively perceive, reason, and act within a dynamic environment to achieve its goals. This introduces new challenges for VLMs, including long-horizon planning, commonsense reasoning, reliable visual grounding, and spatial awareness (Yang et al., 2025; Cheng et al., 2025).

Recent studies have systematically evaluated VLMs as embodied agents (Liu et al., 2024b; Yang et al., 2025; Cheng et al., 2025; Li et al., 2025b). With carefully designed prompting, large-scale VLMs can solve increasingly complex tasks, but their success comes at high cost: massive models demand expensive hardware, long training cycles, and costly inference, all of which hinder real-world deployment where efficiency is critical. Meanwhile, the performance gap between large and small models remains striking. For example, Claude-3.5-Sonnet achieves 64.0% on EB-ALFRED, compared to only 4.7% for Qwen2.5-VL-7B-Instruct (Yang et al., 2025). This disparity highlights the limitations of smaller models, which often lack embodied knowledge, robust reasoning, and the synergy between high-level planning and low-level grounding. Thus, enabling compact models to master complex embodied tasks remains an open challenge. Recent efforts have explored reinforcement learning (RL) to enhance embodied agents' reasoning capabilities (Zhai et al., 2024; Kim et al., 2025; Zhang et al., 2025c; Wu et al., 2025; Feng et al., 2025b; Wang* et al., 2025), but most apply RL only to static QA-style datasets or focus narrowly on high-level reasoning, leaving low-level embodied tasks underexplored and raising doubts about whether such gains generalize across the full embodied spectrum.

In this paper, we address the gap between large and small VLMs in embodied tasks with the **Embodied Reasoning Agent (ERA)**, a two-stage training framework designed to unlock generalizable embodied skills in VLMs. ERA builds on two ideas: introducing embodied priors into small VLMs, and refining them with online RL. Since general VLMs, especially small ones, lack domain-specific abilities in embodied environments, the first stage, *Embodied Prior Learning*, injects tailored knowledge to strengthen reasoning, perception, and environmental understanding. We categorize three sources of prior knowledge: (i) *Trajectory-Augmented Priors*, which enrich existing trajectories with reasoning annotations from stronger VLMs and rule-based visual description augmentation; (ii) *Environment-Anchored Priors*, which provide in-environment knowledge and grounding in the form of QA pairs beyond agent-collected trajectories; (iii) *External Knowledge Priors*, which transfer general skills (e.g., mathematical reasoning, spatial reasoning) from large-scale out-of-environment data and can be curated at minimal cost. The second stage applies online RL to further enhance agents' performance. Agents are trained with an improved PPO pipeline, with three key designs: efficient context management via self-summarization, dense reward shaping with sub-goal and behavior-shaping rewards, and turn-level policy optimization. These components together enable stable and efficient policy learning in long-horizon, sparse-reward settings.

We evaluate ERA on EmbodiedBench (Yang et al., 2025), focusing on EB-ALFRED (high-level planning) and EB-Manipulation (low-level control), which together offer broad coverage of embodied reasoning tasks. ERA-3B not only surpasses prompting-based large models (e.g., GPT-4o) but also outperform 7B-scale training-based baselines, achieving an average score of 65.2% on EB-ALFRED and 48.3% on EB-Manipulation. Moreover, our ablation studies disentangle the contributions of different priors in the first stage, as well as context management, reward shaping, and turn-level optimization in the RL stage, providing practical insights for building effective training pipelines for embodied agents.

Our main contributions are threefold: (1) We present a comprehensive study on post-training compact VLMs for embodied agents, combining prior knowledge curation for supervised finetuning and long-horizon online RL enhanced by key design choices. (2) We introduce a principled taxonomy of accessible prior knowledge for embodied agents, offering practical guidance for data curation across different task levels. (3) We achieve strong results on both high- and low-level tasks with a 3B model and provide detailed ablations analyzing the impact of each data component and RL design choice.

## 2 RELATED WORK

**Foundation Model–based Embodied Agents.** LLMs and VLMs have been explored as embodied agents, enabling them to perceive complex environments and make sequential decisions. Early work relied on prompting strategies to harness the reasoning and planning capabilities of foundation models (Singh et al., 2022; Song et al., 2023; Hu et al., 2023; Kim et al., 2024a; Shin et al., 2025). Building on this foundation, subsequent research introduced mechanisms to improve decision-making, such as code-based tools (Liang et al., 2023; Silver et al., 2024). More recently, the availability of curated embodied datasets has facilitated supervised finetuning, which has proven effective across both low-level robotic control tasks (Zawalski et al., 2024; Zhao et al., 2025; Lee et al., 2025; Liu et al., 2025a; Kim et al., 2024b; Lu et al., 2025a; Huang et al., 2025; Zhang et al., 2025b) and high-level embodied planning (Wu et al., 2023; Chen et al., 2024a; Ji et al., 2025).

**RL for Embodied Agents.** Beyond supervised learning, RL has become a central approach for training embodied agents (Su & Zhang, 2023; Zhai et al., 2024; Yang et al., 2024; Shu et al., 2025; Cao et al., 2025; Liu et al., 2025b; Kim et al., 2025; Szot et al., 2025; Feng et al., 2025a). A key strength of RL lies in its ability to exploit suboptimal and even failed trajectories, thereby making efficient use of diverse data sources (Song et al., 2024; Wang et al., 2025a). Recent progress further shows that RL can foster reasoning abilities, enabling embodied agents to generalize more effectively to novel tasks (Wu et al., 2025; Wei et al., 2025; Lin et al., 2025). Meanwhile, studies reveal that smaller LLMs and VLMs often lack crucial embodied knowledge, such as spatial reasoning and awareness (Gao et al., 2024; Lee et al., 2025; Sun et al., 2024). To address this gap, grounding embodied knowledge into VLMs prior to RL training has emerged as a promising direction.

## 3 FROM PRIORS TO POLICIES: TRAINING VLMS AS EMBODIED AGENTS

We introduce the *Embodied Reasoning Agent (ERA)*, a two-stage framework for training compact VLMs on both high-level planning and low-level control tasks. High-level tasks emphasize logical reasoning and long-term planning through semantically meaningful actions (e.g., "find a Hand-Towel"), while low-level tasks demand precise perception and fine-grained control, often realized through continuous robotic arm commands (e.g., 7D vectors for translation, rotation, and gripper state). To equip VLMs with these complementary capabilities, ERA combines: (1) **Embodied Prior Learning**, which injects structured perception and reasoning via supervised finetuning on curated prior data, and (2) **Online Reinforcement Learning**, which further enhances embodied performance through dense reward shaping and turn-level policy optimization.

### 3.1 EMBODIED PRIOR LEARNING

To finetune VLMs for embodied tasks, the common strategy is to finetune on task-specific trajectories (Liu et al., 2024b; Yuan et al., 2025; Wu et al., 2025; Feng et al., 2025a). This faces two key challenges: *data scarcity and cost*, since collecting trajectories is expensive, and *limited reasoning supervision*, as datasets like ALFRED typically provide only action sequences without detailed reasoning traces. To address these issues, we curate embodied prior data from diverse sources via three complementary priors: **trajectory-augmented priors**, which enrich trajectories with reasoning and visual descriptions; **environment-anchored priors**, which supply contextual grounding beyond trajectories; and **external knowledge priors**, which transfer general reasoning skills from out-of-domain data. An overview of these curated datasets is shown in Table 11.

### 3.1.1 TRAJECTORY-AUGMENTED PRIORS

Most embodied trajectories contain only observations and actions, lacking step-wise reasoning needed for complex tasks. While some work (Yang et al., 2025) adds high-level rationales, this coarse supervision fails to provide intermediate guidance required for effective error recovery.

To address this limitation, we construct trajectory-augmented priors by enriching every step of the trajectory with explicit reasoning supervision from large VLMs such as GPT-4o. Specifically, for each timestep $t$, we define a structured reasoning representation $z_t = \{z_t^{vis}, z_t^{ref}, z_t^{plan}\}$, where $z_t^{vis}$ is a *visual description* of the current state, $z_t^{ref}$ is a *reflection* on the history to detect and correct potential errors, and $z_t^{plan}$ is a *step-level plan* for achieving the task. By prompting GPT-4o with the language instruction $L$, current observation $I_t$, action history $\{a_0, \ldots, a_{t-1}\}$, and the current action $a_t$, we obtain $z_t$ that enriches the trajectory with structured "inner monologue." This step-level

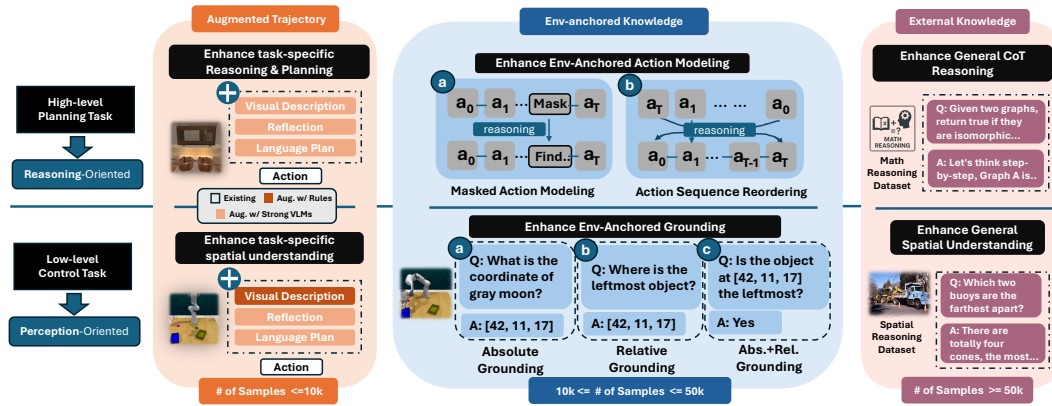

Figure 2: Illustration of Embodied Prior Learning (EPL). EPL leverages three data sources: Augmented trajectory priors, environment-anchored priors, and external knowledge priors.

reasoning has been shown to significantly improve generalization in high-level embodied tasks(Feng et al., 2025a; Zhai et al., 2024). Prompting details of GPT-4o are provided in Appendix H.6.

For low-level manipulation tasks, however, GPT-4o often produces inaccurate visual descriptions, creating misalignments between perception and action. To address this, we adopt a rule-based oracle method that generates ground-truth visual descriptions, ensuring consistency between perception and control. Further details of this rule-based procedure are given in Appendix H.7.

### 3.1.2 ENVIRONMENT-ANCHORED PRIORS

Although trajectories provide direct supervision, they are limited and expensive to collect. This motivates the use of *environment-anchored priors*: auxiliary environment-level signals such as semantic QA and visual grounding, which enrich agents with environment and task understanding.

For **EB-ALFRED**, we curate two datasets: *masked action modeling* and *action sequence reordering*. These are derived from the ALFRED training dataset and adapted via rule-based matching to align with the EB-ALFRED action space (details in Appendix H.8).

- **Masked Action Modeling.** Given an instruction $L$ and an action sequence $\{a_0, a_1, \ldots, a_T\}$, we mask a randomly selected timestep $t \in \{0, \ldots, T\}$, replacing $a_t$ with [MASK]. This produces a query–output pair: $q = (L, \{a_0, \ldots, a_{t-1}, [\text{MASK}], a_{t+1}, \ldots, a_T\})$, $y = (z, a_t)$, where $q$ is the masked trajectory with its instruction, and $y$ is the output: the missing action $a_t$ along with a reasoning trace $z$. Reasoning traces, generated by GPT-4o, justify why $a_t$ is the correct action, providing explicit supervision that strengthens both prediction and interpretability.

- **Action Sequence Reordering.** Here, an action sequence $\{a_0, a_1, \ldots, a_T\}$ is randomly shuffled into a permuted sequence $\{a_{m_0}, a_{m_1}, \ldots, a_{m_T}\}$. The query–output pairs are organized as:

$$q = (L, \{a_{m_0}, a_{m_1}, \ldots, a_{m_T}\}), \quad y = (z, \{a_0, a_1, \ldots, a_T\}),$$

where $q$ is the permuted sequence with its instruction, and $y$ is the correctly ordered sequence accompanied by a reasoning trace $z$ generated by GPT-4o. The reasoning trace explains why the order is correct, enabling the model to better understand temporal dependencies.

**For EB-Manipulation, we curate environment-anchored prior data to capture spatial understanding**, which are crucial for low-level embodied tasks. Using simulated episodes from the VLM-bench training set (Zheng et al., 2022), we combine image observations with ground-truth 3D coordinates to construct three complementary subsets: *absolute coordinate grounding*, *relative coordinate grounding*, and their *combination*. Detailed examples for each subsets are deferred to Appendix H.4.

- **Absolute Coordinate Grounding.** Maps objects to their 3D coordinates, either predicting coordinates from objects or describing objects from coordinates.

- **Relative Coordinate Grounding.** Captures spatial relations (e.g., "leftmost"), where the model predicts target coordinates from relational descriptions.

- **Combined Grounding.** Integrates absolute and relative grounding via binary queries (e.g., "Is the object at [42, 11, 17] the leftmost?"), enabling joint reasoning.

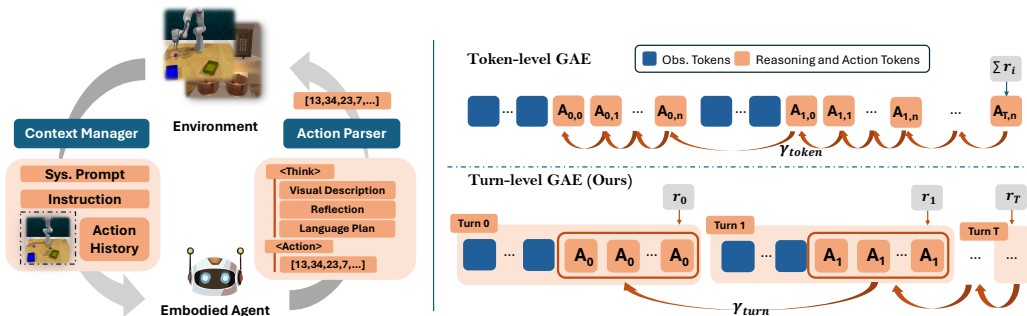

Figure 3: (a) Our agent framework, and (b) a comparison of turn-level GAE and token-level GAE.

### 3.1.3 EXTERNAL KNOWLEDGE PRIORS

While environment-anchored priors provide valuable task-specific knowledge, they are limited in scale compared to general LLM/VLM datasets. To complement them, we introduce *external knowledge priors*: large-scale out-of-environment datasets that transfer abstract reasoning and cross-domain grounding, enabling agents to generalize beyond environment-specific supervision.

For **high-level planning tasks** such as EB-ALFRED, we investigate whether external reasoning datasets can strengthen agents' planning ability. We adopt *OpenO1-SFT dataset* (Open O1 Team, 2024), a text-based supervised fine-tuning dataset designed to activate chain-of-thought reasoning. We sample 10,000 QA pairs to build the dataset. For **low-level control tasks** such as EB-Manipulation, we examine whether external spatial reasoning datasets can enhance agents' visual perception and physical object understanding. To this end, we utilize the *SpaceThinker dataset* (Remyx AI, 2025), a multimodal spatial reasoning corpus synthesized via the VQASynth pipeline. We use its entire 11,413 QA pairs to curate the dataset. By incorporating these external knowledge priors, we complement environment-anchored supervision with large-scale reasoning and grounding signals, equipping VLMs with capabilities that extend beyond the limits of embodied data alone.

### 3.2 ONLINE REINFORCEMENT LEARNING

While Embodied Prior Learning (§3.1) equips agents with foundational skills, online reinforcement learning (RL) is crucial for refining these priors and developing adaptive strategies through environmental interaction. However, applying RL to VLM-based agents presents three major challenges: the need for an **efficient agent design** to manage high computational costs, the difficulty of credit assignment in **long-horizon tasks with sparse rewards**, and the instability of **policy optimization** with conventional token-level methods that are ill-suited for turn-based interactions. To tackle these challenges, we design an online RL pipeline with three key components: (1) an efficient agent framework using self-summarization for context management; (2) a dense reward function providing richer supervision; and (3) turn-level value to stabilize policy optimization.

### 3.2.1 AGENT FRAMEWORK

Our agent framework, illustrated in Figure 3(a), is a unified pipeline designed to process multimodal inputs, including language instructions, visual observations, and interaction history, and generate structured reasoning and executable actions. Specifically, at each turn,

**Structured Reasoning and Action.** At each turn, the agent generates a response in a ReAct-style format that combines a reasoning trace with an executable action (either a string or a 7D vector), separated by special tokens. The reasoning trace is structured into three components, $\mathbf{y_t} = \{y_t^{vis}, y_t^{ref}, y_t^{plan}\}$, where $y_t^{vis}$ is a language description of the current visual state, $y_t^{ref}$ is a reflection and summarization on the action history, and $y_t^{plan}$ is a step-level plan for achieving the task. This structured design elicits the reasoning capability of foundation models while enabling fine-grained optimization during agent training.

**Self-Summarization Context Management.** The context manager organizes historical information and feeds it into the model to ensure continuity of reasoning. A central challenge in long-horizon tasks is the *context explosion problem*: naively retaining the full history of agent outputs (reasoning and action) and environment feedback, $h_t = (\mathbf{y}_1, a_1, e_1, \ldots, \mathbf{y}_{t-1}, a_{t-1}, e_{t-1})$, causes input length to grow linearly with turn number $t$, i.e., $\mathcal{O}(t)$. This is computationally inefficient and may harm performance by diverting attention to irrelevant history. Sliding-window approaches are a common

workaround, but their window sizes are often chosen heuristically rather than principled. In our framework, trajectory-augmented priors train the model to explicitly summarize the any action history in its prompt into a through structured reasoning and reflection at each step. This design allows the agent to compress the entire past into its most recent output $\mathbf{y}_{t-1}$ and we only need a one-step context $h_t = (\mathbf{y}_{t-1}, a_{t-1}, e_{t-1})$, effectively reducing context size to $\mathcal{O}(1)$ while retaining essential information. We refer to this lightweight mechanism as *Self-Summarization*. It enables efficient long-horizon reasoning without sacrificing critical historical context.

### 3.2.2 REWARD DESIGN

Embodied tasks are typically long-horizon (e.g., 20 steps) and often suffer from sparse supervision, where rewards are only given upon task success. To provide richer learning signals, we design a multi-component reward function $r_t$ that integrates task completion, intermediate progress, and behavior shaping. At each turn $t$, the reward is defined as $r_t = r_t^{\text{success}} + r_t^{\text{subgoal}} + r_t^{\text{behavior}}$, with the components detailed below and further elaborated in Appendix F.1.2.

**Success-based Reward** ($r_t^{\text{success}}$). This sparse reward is given at task termination: positive for successful completion and 0 if the episode ends after exceeding the step limit. **Subgoal-based Reward** ($r_t^{\text{subgoal}}$). To provide denser feedback for RL training, subgoal rewards are assigned the first time the agent achieves rule-based subgoals. For high-level planning tasks, subgoal rewards correspond to the proportion of conditions satisfied in the Planning Domain Definition Language (PDDL) specification defined by the simulator. For low-level manipulation tasks, subgoals are defined as the first successful approach of the end-effector to an instruction-referenced object within a predefined distance threshold. **Behavior-Shaping Reward** ($r_t^{\text{behavior}}$). This component shapes task-specific behaviors by rewarding desirable actions and penalizing undesirable ones. For high-level planning, penalties are applied to invalid actions that the environment cannot execute (e.g., attempting to pick up an object while already holding another). For low-level manipulation, rewards are based on the accuracy of the agent's visual grounding, quantified by the ratio of correctly matched attributes against the ground truth. Thresholds on this ratio are used to assign positive or negative values. Full implementation details are provided in Appendix F.1.2.

### 3.2.3 TURN-LEVEL POLICY OPTIMIZATION

Conventional token-level optimization, widely adopted in RLHF and recent agent RL works (Ouyang et al., 2022; Wang et al., 2025b), is not well-suited for multi-turn embodied agents. In embodied tasks, interactions and rewards are inherently defined at the *turn level*. Learning a value function for individual reasoning or action tokens is therefore less meaningful and often leads to high-variance advantage estimates and unstable policy optimization.

To address this challenge, we propose a *turn-level policy optimization* scheme, where the agent's entire response in a turn is treated as a single "action." At each turn $t$, rather than estimating values for every token, we pass only the state input $\mathbf{x}_t$ (observation, instruction, and history) to the value function to obtain a single estimate $V_\phi(\mathbf{x}_t)$. Given turn-level rewards $\{r_t\}_{t=0}^T$ for a trajectory of $T$ turns, we compute the temporal-difference (TD) residual for each turn: $\delta_t = r_t + \gamma V_\phi(\mathbf{x}_{t+1}) - V_\phi(\mathbf{x}_t)$, with terminal bootstrap $V_\phi(\mathbf{x}_{T+1}) = 0$. The turn-level generalized advantage estimate (GAE) is then calculated as: $A_t = \sum_{l=0}^{T-t} (\gamma\lambda)^l \delta_{t+l}$. This advantage estimate $A_t$ is **shared across all tokens within the response** $\mathbf{y}_t$ in turn $t$, ensuring that credit assignment aligns with the natural unit of environment interaction.

We perform parallel rollouts with multiple environments and collect an online buffer $\mathcal{D}$ of turn-level state–response pairs for PPO updates. The value function is trained concurrently by regressing toward a detached target: $\mathcal{L}_{\text{value}}(\phi) = \mathbb{E}_{\mathbf{x}_t \sim \mathcal{D}} \left[ \frac{1}{2} \left( V_\phi(\mathbf{x}_t) - \text{no\_grad} \left( A_t + V_\phi(\mathbf{x}_t) \right) \right)^2 \right]$. Overall, this turn-wise formulation reduces variance in advantage estimation and leads to more stable policy learning for embodied agents. We compare token-level and turn-level GAE in Section 4.4.

## 4 EXPERIMENTS

We conduct comprehensive experiments on both high-level planning and low-level manipulation tasks, aiming to gain deeper insights into how different design choices contribute to embodied agent learning. Specifically, we address the following research questions:

① **Q1**: What performance does ERA achieve compared to strong baselines?
② **Q2**: What role do different prior datasets play in agent performance?

Table 1: Task success rates on the five subsets of EB-ALFRED and EB-Manipulation. The best result in each column is highlighted in **bold**. "Base," "Complex," and "Visual" are seen subsets, while "Common" and "Spatial" are unseen subsets.

| Model | EB-Alfred | | | | | | EB-Manipulation | | | | | |
|---|---|---|---|---|---|---|---|---|---|---|---|---|
| | Avg | Base | Complex | Visual | Common | Spatial | Avg | Base | Complex | Visual | Common | Spatial |
| *Prompting-based MLLMs* | | | | | | | | | | | | |
| GPT-4o | 56.8 | 64 | 68 | 46 | 54 | 52 | 28.9 | 39.6 | 29.2 | 19.4 | 29.2 | 25.0 |
| Claude-3.5-Sonnet | **66.4** | **72** | **76** | 60 | **66** | 58 | 25.4 | 37.5 | 29.2 | 19.4 | 16.7 | 22.9 |
| Gemini-1.5-Pro | 63.2 | 70 | 72 | 58 | 64 | 52 | 21.1 | 14.6 | 22.9 | 16.7 | 14.6 | 35.4 |
| Gemini-2.0-flash | 51.2 | 62 | 54 | 46 | 48 | 46 | 16.7 | 14.6 | 14.6 | 13.9 | 8.3 | 31.3 |
| Llama-3.2-90B-Vision-Ins | 35.2 | 38 | 44 | 28 | 34 | 32 | 14.9 | 10.4 | 16.7 | 10.4 | 12.5 | 20.8 |
| InternVL3-78B | 39.6 | 38 | 46 | 42 | 34 | 38 | 26.3 | 29.2 | 22.9 | 25.0 | 22.9 | 31.3 |
| Qwen2.5-VL-72B-Ins | 40.8 | 50 | 42 | 36 | 42 | 34 | 16.2 | 12.5 | 16.7 | 22.2 | 12.5 | 18.8 |
| Qwen2.5-VL-7B-Ins | 5.2 | 10 | 6 | 2 | 8 | 0 | 9.6 | 8.3 | 8.3 | 5.6 | 8.3 | 16.7 |
| Qwen2.5-VL-3B-Ins | 0 | 0 | 0 | 0 | 0 | 0 | 0 | 0 | 0 | 0 | 0 | 0 |
| *Training-based MLLMs* | | | | | | | | | | | | |
| RL4VLM (3B) | 51.2 | 70 | 70 | 56 | 32 | 28 | 21.9 | 33.3 | 29.2 | 30.6 | 8.3 | 8.3 |
| VAGEN (3B) | 52.8 | 70 | 70 | 58 | 38 | 28 | 22.9 | 35.4 | 31.3 | 29.2 | 8.3 | 10.4 |
| Reinforced Reasoner (7B) | 41.6 | 54 | 46 | 28 | 42 | 38 | - | - | - | - | - | - |
| Robot-R1 (7B) | - | - | - | - | - | - | 11.7 | 12.5 | 6.3 | 16.7 | 8.3 | 14.6 |
| **ERA-3B (EPL-only)** | 56.0 | 68 | 66 | 52 | 44 | 50 | 40.0 | 45.8 | 41.7 | 47.9 | 37.5 | 27.1 |
| **ERA-3B (EPL+RL)** | 65.2 | **72** | 72 | **62** | 54 | **66** | **48.3** | **56.3** | **47.9** | **50.0** | **47.9** | **39.6** |

③ **Q3**: Is Self-Summarization effective for context management?

② **Q4**: How do reward design and turn-level value impact RL performance?

**Experiment Setup.** We build ERA on top of Qwen2.5-VL-3B-Instruct and evaluate models on *EmbodiedBench*, a comprehensive benchmark covering both high-level planning and low-level control. Each benchmark includes diverse subsets targeting distinct capabilities. To assess both in-distribution learning and out-of-distribution generalization, we use base skills, complex instruction following, and visual perception for ***training (seen)*** and hold out commonsense reasoning and spatial awareness subsets for ***testing (unseen)***. Task success rate serves as the primary evaluation metric. Additional details on dataset curation, training hyperparameters, and evaluation are provided in the Appendix E

### 4.1 Q1: WHAT PERFORMANCE DOES ERA ACHIEVE COMPARED TO STRONG BASELINES?

In Table 1, we compare ERA-3B against a diverse set of baselines. These include *prompting-based models*, such as GPT-4o, Claude-3.5-Sonnet, Qwen2.5-VL (3B, 7B, and 72B) and other popular proprietary models. We also compare with *Training-based models*, such as RL4VLM (Zhai et al., 2024), VAGEN (Wang* et al., 2025), Reinforced Reasoner (Wu et al., 2025) and Robot-R1 (Kim et al., 2025). Full reproduction details are provided in Appendix F.

**Overall Results.** ERA establishes a new state-of-the-art among training-based agents, substantially outperforming previous RL baselines on both high-level planning (+12.4% over VAGEN) and low-level manipulation (+25.4% over VAGEN). ERA achieves average success rates of 65.2% on EB-ALFRED and 48.3% on EB-Manipulation, exceeding proprietary models such as GPT-4o by 8.4 and 19.4 points, respectively. Remarkably, these results are obtained with a compact 3B model, underscoring the parameter efficiency of ERA for embodied agents. On EB-ALFRED, ERA-3B is also competitive with the top-performing large proprietary model Claude-3.5-Sonnet (66.4%).

**Unseen Generalization.** On EB-ALFRED, RL4VLM and VAGEN perform comparably to ERA on the three seen subsets but fall far behind on the two unseen subsets. For example, ERA achieves 66% on the *Spatial* subset, outperforming VAGEN (28%) by 38 points. Similar patterns are observed on EB-Manipulation. These results show that ERA learns robust and transferable skills rather than overfitting to the training tasks.

**Benefits of EPL and RL in ERA.** The results further highlight the complementary roles of EPL and RL in ERA. EPL alone provides a strong foundation, reaching success rates of 56.0% on EB-ALFRED and 40.0% on EB-Manipulation. Adding the RL stage yields substantial gains, improving average success rates by 9.2 and 8.3 points, respectively. These improvements are especially pronounced on unseen subsets, with average gains of 13.0 points on EB-ALFRED and 11.5 points on EB-Manipulation. Together, these findings confirm that EPL imparts essential foundational knowledge, while online RL effectively refines these priors to boost generalization.

Table 2: Ablation results on different prior datasets. We report average success rates on both seen and unseen splits. 'Traj-Aug', 'Env-Anc', and 'Ext-Know' denotes Trajectory-Augmented, Environment-Anchored, and External Knowledge Priors. Stage 1 and Stage 2 correspond to EPL and RL, respectively. Numbers in parentheses indicate gains over the raw trajectory baseline.

| Methods | EB-ALFRED | | | | EB-Manipulation | | | |
| --- | --- | --- | --- | --- | --- | --- | --- | --- |
| | Stage 1 | | Stage 1 & 2 | | Stage 1 | | Stage 1 & 2 | |
| | Seen | Unseen | Seen | Unseen | Seen | Unseen | Seen | Unseen |
| Base Model (No prior injected) | – | – | 0 | 0 | – | – | 0 | 0 |
| Raw Trajectory (baseline) | 59.3 | 32.0 | 64.0 | 36.0 | 25.0 | 7.3 | 44.0 | 21.9 |
| + Traj-Aug | 62.0 (+2.7) | 37.0 (+5.0) | 66.7 (+2.7) | 49.0 (+13.0) | 41.4 (+16.4) | 26.1 (+18.8) | 50.3 (+6.3) | 35.5 (+13.6) |
| + Env-Anc | **63.3 (+4.0)** | 39.0 (+7.0) | **70.0 (+6.0)** | 42.0 (+6.0) | 25.0 (+0.0) | 9.4 (+2.1) | 47.2 (+3.2) | 22.9 (+1.0) |
| + Ext-Know | **63.3 (+4.0)** | 35.0 (+3.0) | 68.7 (+4.7) | 46.0 (+10.0) | 30.3 (+5.3) | 18.8 (+11.5) | 48.5 (+4.5) | 27.1 (+5.2) |
| + Traj-Aug + Env-Anc | 62.0 (+2.7) | **47.0 (+15.0)** | 68.7 (+4.7) | **60.0 (+24.0)** | **45.1 (+20.1)** | **32.3 (+25.0)** | **51.4 (+7.4)** | **43.8 (+21.9)** |
| + Traj-Aug + Ext-Know | **63.3 (+4.0)** | 44.0 (+12.0) | 68.7 (+4.7) | 55.0 (+19.0) | 37.9 (+12.9) | 31.3 (+24.0) | **51.4 (+7.4)** | 37.5 (+15.6) |

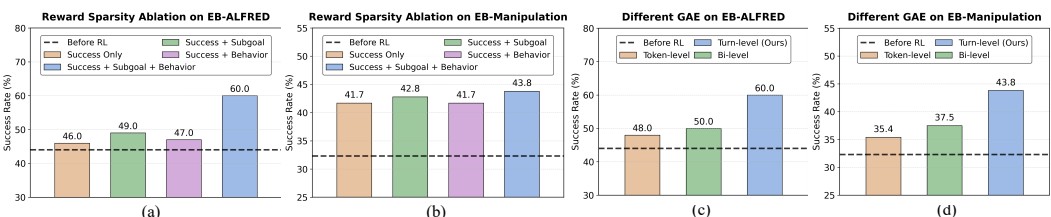

Figure 4: (a)(b) Reward design ablations and (c)(d) Value estimation method comparisons, both on EB-ALFRED and EB-Manipulation.

## 4.2  Q2: WHAT ROLE DO DIFFERENT PRIOR DATASETS PLAY IN AGENT PERFORMANCE?

To evaluate the impact of different prior datasets, we perform ablations on both the EPL (Stage 1) and RL (Stage 2) phases in Table 2.

**Trajectory-Augmented Priors Achieve the Largest Individual Gains in Generalization.** Among the three individual datasets, trajectory augmentation yields the strongest improvements on unseen tasks. On EB-ALFRED, augmenting raw trajectories with structured reasoning improves unseen performance by +13.0% after Stage 2, relative to the raw-trajectory baseline. The effect is even stronger on EB-Manipulation, with a +13.6% gain on unseen tasks. In comparison, Environment-Anchored and External Knowledge Priors achieve relatively modest improvements of +6.0%/+10.0% on EB-ALFRED (unseen) and +1.0%/+5.2% on EB-Manipulation (unseen). These results highlight the importance of structured reasoning in enhancing transfer to novel tasks.

**Environment-Anchored Priors Improve Seen and Unseen Tasks Equally, While External Knowledge Priors Favor Unseen Tasks.** Environment-Anchored Priors produce balanced improvements across seen and unseen subsets. For instance, on EB-ALFRED they deliver a +6% gain over the raw-trajectory baseline for both seen and unseen tasks after stage 2. This consistency suggests that environment-anchored data encode environment-level knowledge that is broadly useful across tasks in similar environments. In contrast, External Knowledge Priors lead to larger improvements on unseen tasks than on seen ones. This indicates that while external data are less environment-specific, they capture general reasoning and grounding skills that support generalization to novel tasks. However, their overall gains remain smaller than trajectory augmentation.

**Combining Trajectory-Augmented and Environment-Anchored Priors Elicit the Best Performance.** We next examine whether combining different priors leads to further gains. Notably, the combination of Trajectory-Augmented and Environment-Anchored Priors achieves the strongest overall results: after Stages 1 and 2, performance reaches 60% on the unseen subsets of EB-ALFRED (+24% over baseline) and 43.8% on the unseen subsets of EB-Manipulation (+21.9% over baseline). While combining Trajectory-Augmented and External Knowledge Priors also produces substantial improvements, the gains are relatively smaller. These findings suggest that environment-anchored data provide explicit, task-relevant supervision that complements trajectory-based priors, whereas external knowledge offers a weaker but more easily obtainable alternative when environment-specific data are costly to curate.

### 4.3 **Q3:** IS SELF-SUMMARIZATION EFFECTIVE FOR CONTEXT MANAGEMENT?

To evaluate the effectiveness of the proposed self-summarization mechanism, we conduct an ablation study in Table 3 comparing unseen task performance with and without self-summarization after Stage 1. The key difference is that the *w/o Self-Summarization* setting excludes the model's generated reflection from the context, while retaining other history information identical to the self-summarization setting. Results show that self-summarization significantly improves the success rate by 8% to 10% across different of history. Notably, with self-

Table 3: Comparison of average success rate (SR) and average input tokens with varying number of history steps included in the context. SR is averaged over unseen subsets.

| History Length | EB-ALFRED | | EB-Manipulation | |
|---|---|---|---|---|
| | SR (%) ↑ | #Input Tokens ↓ | SR (%) ↑ | #Input Tokens ↓ |
| *w/ Self-Summarization* | | | | |
| 1 step (Ours) | **47** | **217.4** | **32.3** | 399.5 |
| 3 steps | 45 | 490.5 | 30.3 | 798.3 |
| 5 steps | 46 | 680.4 | 29.1 | 998.3 |
| *w/o Self-Summarization* | | | | |
| 1 step | 41 | 157.0 | 24.0 | 305.3 |
| 3 steps | 35 | 332.7 | 15.6 | 564.4 |
| 5 steps | 36 | 455.8 | 22.9 | 694.5 |

summarization, including only one-step history is sufficient to outperform its 3- to 5-step history counterpart, while using fewer tokens, likely due to the distraction introduced by redundant information. These findings demonstrate that a concise summary generated by the model can provide a more efficient history representation, enabling the agent to proactively focus on relevant context without being hindered by lengthy histories.

### 4.4 **Q4:** HOW DO REWARD DESIGN AND TURN-LEVEL VALUE IMPACT RL PERFORMANCE?

To assess the effect of two key RL design choices, we conduct ablations in Figure 4, reporting average success rates on unseen subsets of EB-ALFRED and EB-Manipulation. All methods are initialized from the same EPL checkpoint to ensure fair comparison.

**Synergistic Dense Reward Improves Long-Horizon RL.** Reward sparsity poses a major challenge for credit assignment, particularly in long-horizon tasks. As shown in Figure 4, supplementing sparse success-based rewards with two turn-level signals (subgoal-based and behavior-shaping rewards) substantially improves performance, particularly for high-level planning tasks. On EB-ALFRED, the average success rate rises by 14% (46% → 60%) compared to training with only success-based rewards. In contrast, the gain on the shorter-horizon EB-Manipulation benchmark is modest (+2.1%). This disparity shows that dense rewards are especially critical for long-horizon tasks, where they can guide exploration and stabilize credit assignment. Moreover, using only subgoal-based or only behavior-shaping rewards produces limited gains, highlighting the synergistic effect of combining multiple dense reward signals.

**Turn-Level Value Estimation Enhances Policy Learning.** We also compare three value learning schemes: token-level, bi-level (Wang* et al., 2025), and our turn-level GAE. Token-level and bi-level approaches require learning a value function over individual tokens, distributing fine-grained credit across reasoning and action tokens. In contrast, turn-level GAE treats the entire response as a single action and learns a turn-level value function. Results show that turn-level GAE achieves the strongest performance, improving average success rates by 12 points on EB-ALFRED (48% → 60%) and by 8.4 points on EB-Manipulation (35.4% → 43.8%) compared to token-level GAE. Bi-level GAE offers modest gains over token-level GAE, likely because it incorporates partial turn-level credit assignment, but its reliance on unstable token-level value estimation still limits effectiveness. These results validate that turn-level credit assignment yields more stable policy learning.

## 5 CONCLUSION

In this work, we propose a two-stage framework for transforming VLMs into capable embodied reasoning agents. In the first stage, we show that enriching existing trajectory data with structured reasoning, incorporating environment-anchored supervision, and leveraging external knowledge each improve agent performance, with their combination yielding even greater gains. In the second stage, we highlight the importance of careful design choices in context management, dense reward shaping, and turn-level value learning, providing insights into the factors that drive effective RL for embodied agents. These contributions establish a practical and scalable recipe for developing more powerful and efficient VLM-based agents in more realistic settings.

ETHICS STATEMENT

After carefully reviewing the ethical regulations of the conference, to the best of our knowledge, this work does not present any foreseeable ethical concerns. No negative societal or ethical impacts are anticipated for the contribution of this work.

REPRODUCIBILITY STATEMENT

We have made efforts to ensure that our work is reproducible, with details provided in Section 4 and Appendix D, F.2, and G.

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

APPENDIX CONTENTS

## A    LIMITATION

A key limitation of this work is that all evaluations are conducted in simulated environments, without validation on real-world systems. This reflects a common trade-off in agent research: simulations provide standardized and reproducible benchmarks that greatly reduce time, cost, and safety risks, but they inevitably limit real-world applicability. While such practice is common for LLM/VLM-based agents (Zhai et al., 2024; Feng et al., 2025b), real-world testing remains crucial for practical deployment. As future work, we plan to explore deploying our ERA training pipelines in real-world environments.

## B    USE OF LARGE LANGUAGE MODELS STATEMENT

Large language models are used solely to refine writing and correct grammar. They are not used for generating research ideas or shaping the intellectual content of the work.

## C    ADDITIONAL RELATED WORK

**Vision Language Models**    Vision Language Models (VLMs) have been a popular research domain with their ability to combine multi-modal perception with a strong language backbone. Li et al. (2025a) and other works (Yin et al., 2024; Zhang et al., 2024b; Awais et al., 2025) categorize VLMs into subdomains like vision to text (Chen et al., 2024b; Deitke et al., 2024; Bai et al., 2025; Li et al., 2024a;b; Zhang et al., 2024c), vision to action (Sima et al., 2024), and text to vision (Deng et al., 2025; Chen et al., 2025b; Sun et al., 2023a; Zheng et al., 2023), etc. For vision to action, embodied AI serves as a perfect area, as it provides a natural environment and action interface. Kim et al. (2024b); Huang et al. (2023); Xu et al. (2024); Wu et al. (2024) pioneer the relevant field by converting vision input into executable actions. VLMs for reasoning comprise a large portion of the suitable applications. For example, existing works (Zhang et al., 2024d; Lu et al., 2023; Zhang et al., 2024e) utilize VLMs for solving math problems. Besides, video models emerge as an extension to VLMs as well. Chen et al. (2025a), Zhang et al. (2025a) and Tian et al. (2025) synthesize video information into text for further processing. The interleaving interaction between vision information and text reasoning provides better deployment for VLMs (Yuan et al., 2024; Tang et al., 2024).

**RL for LLMs or VLMs**    Reinforcement Learning from Human Feedback (RLHF) has become a cornerstone of modern LLM alignment. Early work such as InstructGPT established the paradigm of training a reward model from human preferences and using PPO to fine-tune the base model, demonstrating substantial improvements in helpfulness and safety (Ouyang et al., 2022; Stiennon et al., 2020). To mitigate the need to explicitly training a reward model, implicit reward models like Direct Preference Optimization (DPO (Rafailov et al., 2023)) have been investigated and successfully applied in scenarios with preference data. Since the success of OpenAI-o1 (Jaech et al., 2024) and DeepSeek-R1 (Guo et al., 2025), reinforcement learning (RL) has been a dominant technique for LLMs post-training, especially for reasoning models. To better handle long autoregressive sequences, new algorithms like GRPO and GSPO (Guo et al., 2025; Shao et al., 2024; Zheng et al., 2025) improve stability via group-based comparisons, and offline/self-training methods such as ReST and ReST-MCTS (Gulcehre et al., 2023; Zhang et al., 2024a) reduce online interaction costs by iteratively filtering and retraining on high-quality outputs. Beyond textual alignment, RL is increasingly applied to reasoning (Lightman et al., 2023) and multimodal alignment: LLaVA-RLHF and RLHF-V (Sun et al., 2023b; Yu et al., 2024) demonstrate that preference optimization can mitigate hallucinations and strengthen grounding in visual-language tasks.

Table 4 provides a comprehensive comparison with existing works, highlighting how ERA itself apart from prior works in several aspects.

| Method | Task Level | Reasoning | SFT | | | RL | | |
|---|---|---|---|---|---|---|---|---|
| | | | Traj.-aug. prior | Env.-anchored prior | Ext.-knowledge prior | In-env. interaction | Process-level reward | Value Learning |
| Reinforced Reasoning (Wu et al., 2025) | High | ✓ | ✓ | × | × | × | ✓ | × |
| CoSo (Feng et al., 2025a) | High | ✓ | ✓ | × | × | ✓ | × | × |
| Embodied-R1 (Yuan et al., 2025) | Low | ✓ | - | - | - | ✓ | × | × |
| Robot-R1 (Kim et al., 2025) | Low | ✓ | ✓ | × | × | - | - | - |
| GEA (Szot et al., 2025) | High & Low | × | ✓ | × | × | ✓ | ✓ | ✓ |
| MolmoAct (Lee et al., 2025) | Low | ✓ | ✓ | ✓ | ✓ | - | - | - |
| RL4VLM (Zhai et al., 2024) | High | ✓ | ✓ | × | × | ✓ | × | ✓ |
| RFTF (Shu et al., 2025) | Low | × | - | - | - | ✓ | ✓ | ✓ |
| Vagen (Wang* et al., 2025) | High | ✓ | - | - | - | ✓ | × | ✓ |
| VLA-RL (Lu et al., 2025b) | Low | × | - | - | - | ✓ | ✓ | ✓ |
| Emma-X (Sun et al., 2024) | Low | ✓ | ✓ | ✓ | × | - | - | - |
| ERA (Ours) | High & Low | ✓ | ✓ | ✓ | ✓ | ✓ | ✓ | ✓ |

Table 4: Comparison of finetuning strategies for embodied VLM/LLM agents. Columns grouped under **SFT** cover different pretraining priors; columns under **RL** cover interactive training signals and method comparison.

# D  ENVIRONMENT

## D.1  PROBLEM FORMULATION

Formally, VLM-based agentic tasks can be modeled as a Partially Observable Markov Decision Process (POMDP) augmented with language, represented by the tuple $(\mathcal{S}, \mathcal{A}, \Omega, \mathcal{T}, \mathcal{O}, L, \mathcal{R})$. Here, $\mathcal{S}$ denotes the full environment state space; $\mathcal{A}$ is the action space; and $\Omega$ is the visual observation space, where each observation $I_t = \mathcal{O}(s_t)$ is generated from the underlying state. The agent also receives a language instruction $L$, which specifies the goal. The reward function $\mathcal{R}$ generally provides a binary signal: 1 if the current state satisfies the instruction, and 0 otherwise. At timestep $t$, the agent maintains a history $h_t = (I_0, a_0, \ldots, I_{t-1}, a_{t-1}, I_t)$ of past observations and actions, and acts according to a policy $\pi(a_t \mid L, h_t)$ parameterized by a VLM. The episode terminates either when the instruction is satisfied or when a maximum horizon is reached. The learning objective is to maximize the expected task success rate: $\max_\pi \mathbb{E}_\pi[\sum_{t=0}^{\tau} \gamma^t r_t]$, where $\tau$ denotes the terminal timestep and $\gamma$ is the discount factor.

## D.2  EB-ALFRED

**Task Description.**  EB-ALFRED is built on the ALFRED dataset and the AI2-THOR simulator (Shridhar et al., 2020; Kolve et al.; Deitke et al., 2020), widely recognized for diverse household tasks and realistic environments in embodied AI. The benchmark evaluates an agent's ability to plan and execute sequences of high-level actions in scenarios such as "Put washed lettuce in the refrigerator." Each task is formally represented in the Planning Domain Definition Language (PDDL) (McDermott et al., 1998), enabling precise evaluation of task and subgoal completion.

The ALFRED dataset spans seven task types: *Pick & Place, Stack & Place, Pick Two & Place, Clean & Place, Heat & Place, Cool & Place, and Examine in Light*. Following LoTa-Bench's implementation for household task planning (Choi et al., 2024), our simulator supports eight high-level action primitives: *pick up*, *open*, *close*, *turn on*, *turn off*, *slice*, *put down*, and *find*. These actions are parameterized by objects (e.g., "find an apple" or "pick up an apple"). The simulator provides both egocentric visual observations and textual feedback, indicating whether an action succeeds or fails (e.g., "failure to pick up an object because another object is already being held").

## D.3  EB-MANIPULATION

**Task Description.**  EB-Manipulation extends VLMbench (Zheng et al., 2022) using the CoppeliaSim/V-REP simulator (Rohmer et al., 2013) to control a 7-DoF Franka Emika Panda robotic arm. It comprises four manipulation categories: (1) *Pick & Place Objects*, (2) *Stack Objects*, (3) *Shape Sorter Placement*, and (4) *Table Wiping*, each with diverse instances varying in color, position, shape, and orientation.

The action space is a 7-dimensional vector specifying end-effector translation, rotation, and gripper state. Actions are executed with automatic motion planning in ABS_EE_POSE_PLAN_WORLD_FRAME mode, which drives the trajectory from the current to the target pose, reducing the agent's burden to predicting keypoints essential for task completion.

# E  ADDITIONAL EXPERIMENTS

## E.1  EFFECT OF RULE-BASED GROUND TRUTH VISUAL DESCRIPTION

Table 5:  Ablation study on the effect of oracle visual description on EPL.

| Method | Seen | Unseen |
|--------|------|--------|
| Raw Trajectory (baseline) | 25.0 | 7.3 |
| + Oracle visual description | 39.6 | 22.9 |
| + Traj-Aug | 41.4 | 26.1 |

Table 5 highlights the impact of incorporating oracle visual descriptions into Embodied Prior Learning (EPL). Starting from the raw trajectory baseline, which yields 25.0% on seen and 7.3% on unseen environments, adding oracle visual descriptions substantially improves performance to 39.6% and 22.9%, respectively. This demonstrates that accurate visual grounding plays a critical role in bridging perception and action. Further enriching trajectories with structured reasoning through Traj-Aug leads to additional gains, reaching 41.4% on seen and 26.1% on unseen environments. These results confirm that both accurate visual descriptions and trajectory-level reasoning are essential for enhancing generalization in embodied agents, with the strongest improvements observed in unseen settings.

## E.2  EFFECT OF DIFFERENT COMPONENTS IN THE ENVIRONMENT-ANCHORED PRIOR DATASET

Table 6:  Ablation study on environment-anchored prior dataset on EB-ALFRED, analyzing the impact of training data from different tasks.

| Data Type | EB-ALFRED | | |
|-----------|-----------|--------|---------|
| | Avg | Common | Spatial |
| Masked Action Modeling only | 41 | 38 | 44 |
| Action Sequence Reordering only | 44 | 40 | 48 |
| All | 47 | 44 | 50 |

Table 7:  Ablation study on environment-anchored prior dataset on EB-MANIPULATION, analyzing the impact of training data from different tasks.

| Data Type | EB-Manpulation | | | | | |
|-----------|----------------|------|---------|--------|--------|---------|
| | Avg | Base | Complex | Visual | Common | Spatial |
| No Comb. Grounding | 36.7 | 45.8 | 37.5 | 45.8 | 35.4 | 18.8 |
| No Relative Grounding | 36.6 | 47.9 | 33.3 | 47.9 | 33.3 | 20.8 |
| No Absolute Grounding | 35.0 | 43.8 | 33.3 | 45.8 | 33.3 | 18.8 |
| All | 40 | 45.8 | 41.7 | 47.9 | 37.5 | 27.1 |

The Environment-Anchored dataset was designed with multiple components to provide diverse training signals. To assess their impact, we conduct ablation experiments on Stage 1 performance using either the full dataset or individual subsets (Table 6 & 7) . On EB-ALFRED, combining Masked Action Modeling with Action Sequence Reordering delivers the strongest results, outperforming either task alone (Avg = 47 vs. 41 or 44). On EB-Manipulation, training jointly on absolute, relative, and compositional grounding data also achieves the best results compared to partial combinations. These findings show that performance gains are maximized when training incorporates multiple complementary task formulations, as joint learning from diverse supervision signals enables the model to capture richer environment knowledge.

## E.3  EFFECT OF DIFFERENT REWARD SPARSITY DESIGN IN RL

Reward sparsity poses a significant challenge for credit assignment in reinforcement learning, an effect that is amplified in long-horizon embodied tasks as discussed in Section 4.4. While the main

Table 8: Ablation of RL reward sparsity of unseen task performance

| Reward Sparsity | EB-ALFRED | | | EB-Manipulation | | |
|---|---|---|---|---|---|---|
| | Avg | Common | Spatial | Avg | Common | Spatial |
| Outcome Only | 46 | 42 | 50 | 41.7 | 47.9 | 35.4 |
| Outcome + Subgoal | 49 | 44 | 54 | 42.8 | 45.9 | 39.6 |
| Outcome + B.S | 47 | 44 | 50 | 41.7 | 47.9 | 35.4 |
| Outcome + Subgoal + B.S | 60 | 54 | 66 | 43.8 | 47.9 | 39.6 |

paper highlights the overall benefit of dense rewards, this section provides a more granular analysis of how different reward components influence agent performance on unseen subsets, drawing detailed insights from Table 8.

For the long-horizon planning tasks in EB-ALFRED, supplementing the sparse, outcome-only success reward with denser signals yields substantial improvements. Subgoal-based rewards, which guide the agent toward intermediate milestones, provide a moderate performance uplift (Avg: 46% → 49%), with a more pronounced effect on the *Spatial* subset where performance increases by 4 points. This suggests that explicit guidance on navigational progress is particularly beneficial for spatial reasoning. In contrast, behavior-shaping rewards, which penalize invalid actions, offer a smaller overall gain but specifically improve performance on the *Common Sense* subset by 2 points, indicating that this signal helps the agent learn logical action constraints. Most notably, we observe a strong synergistic effect when combining all three reward components. The average success rate surges to 60%, a 14-point improvement over the outcome-only baseline that far exceeds the sum of the individual gains from subgoal (+3) and behavior-shaping (+1) rewards. This synergy is particularly evident on the unseen subsets, where performance on *Spatial* tasks increases by 16 points and on *Common Sense* tasks by 12 points. This finding underscores that for complex, long-horizon planning, a composite reward function is critical: subgoal rewards effectively guide exploration toward promising states, while behavior-shaping rewards prune the search space by discouraging invalid action sequences, and their combination enables robust generalization.

In the context of the shorter-horizon, low-level control tasks in EB-Manipulation, the impact of dense rewards is more nuanced, though still beneficial. The overall performance gain from the full reward function is modest (Avg: 41.7% → 43.8%). However, a closer look at the subsets reveals important dynamics. Subgoal-based rewards, defined by the end-effector's proximity to target objects, significantly boost performance on the *Spatial* subset (+4.2 points). This confirms that a dense reward signal directly aligned with a specific task aspect—in this case, spatial precision—can effectively improve that capability. Interestingly, this same reward slightly degrades performance on the *Common Sense* subset, suggesting a potential trade-off where optimizing for spatial proximity may distract from more complex, sequential logic. Furthermore, the behavior-shaping reward, based on visual grounding accuracy, yields no improvement on its own. Yet, when combined with the subgoal reward, it recovers the performance drop on the *Common Sense* subset, revealing a subtle synergistic effect. This indicates that while the overall reward signal is less critical than in long-horizon tasks, a carefully balanced combination of dense rewards is still valuable for shaping specific skills without compromising others.

In summary, this detailed analysis confirms that the utility of dense reward signals is strongly correlated with task horizon length. Furthermore, it reveals that different reward components can target distinct agent capabilities, such as spatial awareness or logical consistency. The non-additive, synergistic effects observed in both high-level and low-level tasks highlight the importance of designing a composite reward function that provides both positive guidance and negative constraints to facilitate effective and generalizable policy learning.

### E.4 EFFECT OF DIFFERENT GAEs IN RL

Table 9: Ablation of Different GAE in RL

| Different GAE | EB-ALFRED | | | | | | EB-Manipulation | | | | | |
|---|---|---|---|---|---|---|---|---|---|---|---|---|
| | Avg | Base | Complex | Visual | Common | Spatial | Avg | Base | Complex | Visual | Common | Spatial |
| Token-level GAE | 58.8 | 70 | 72 | 56 | 40 | 56 | 40.0 | 43.9 | 47.9 | 37.5 | 45.8 | 25.0 |
| Bi-level GAE | 60.4 | 72 | 70 | 60 | 44 | 56 | 42.1 | 47.9 | 47.9 | 39.6 | 39.6 | 35.4 |
| Turn-level GAE (Ours) | 65.2 | 72 | 72 | 62 | 54 | 66 | 48.3 | 56.3 | 47.9 | 50.0 | 47.9 | 39.6 |

Vision-Language-Model-based agents generate sequences of tokens that collectively constitute a single atomic action from the environment's perspective. This creates a granularity mismatch for standard token-level advantage estimation, which improperly distributes credit within a single coherent action sequence. To address this, we compare three schemes for Generalized Advantage Estimation (GAE): token-level, bi-level (Wang* et al., 2025), and our proposed turn-level GAE. As detailed in Table 9, aligning credit assignment with the unit of interaction by using turn-level GAE consistently yields the best performance across both high-level planning and low-level control tasks.

For the long-horizon planning tasks in EB-ALFRED, our turn-level GAE demonstrates substantially improved generalization on unseen subsets. It achieves a 65.2% average success rate, a gain of 6.4 points over token-level GAE. The improvements are most striking on the unseen *Common Sense* and *Spatial* subsets, where performance increases by 14 and 10 points, respectively. This highlights that a stable, turn-level credit assignment is critical for learning complex reasoning and planning policies. By treating the entire reasoning-and-action sequence as a single unit, the agent can more effectively learn the causal link between its high-level strategy and the resulting outcome, avoiding the high variance associated with token-level signals.

This advantage is also pronounced in the low-level control tasks of EB-Manipulation. Turn-level GAE again achieves the highest average success rate (43.9%) and delivers remarkable improvements on subsets requiring precise perception and control. For instance, performance on the *Visual* subset improves by 12.5 points and on the *Spatial* subset by a significant 14.6 points compared to the token-level baseline (25.0% → 39.6%). This demonstrates that a holistic credit assignment helps the model better learn the coupling between its generated visual descriptions and the corresponding multi-dimensional control actions, leading to more accurate spatial manipulation.

In contrast, the bi-level GAE, which represents a hybrid approach, offers only modest and sometimes inconsistent gains over the token-level baseline. While it incorporates some turn-level signal, its continued reliance on token-level value estimation appears to limit its effectiveness and stability.

In summary, these results provide strong evidence that matching the temporal unit of credit assignment to the agent's action abstraction is critical for reducing variance and learning generalizable policies. The turn-level GAE proves to be a more stable and effective method for training sequence-generating agents in interactive environments.

## F    IMPLEMENTATION DETAILS

### F.1    ALGORITHM DETAILS

#### F.1.1    DETAILED ANALYSIS OF EFFICIENT CONTEXT MANAGEMENT

This section provides a more detailed analysis of our investigation into efficient context management, expanding upon the discussion in the main paper and drawing deeper insights from the data presented in Table 3.

**Task-Dependent Sensitivity to Interaction History.**    Our experiments reveal a clear distinction in how interaction history affects performance on high-level planning versus low-level manipulation tasks. This divergence stems from the fundamental characteristics of each domain.

- **High-level planning (EB-ALFRED)** is defined by long horizons and partial observability. The agent's current visual input rarely captures the complete state of the environment; for example, it cannot see objects in other rooms or recall which containers it has already checked. Consequently, a memory of past actions, observations, and discoveries is crucial for effective long-term planning. This dependency is empirically confirmed in our ablation study. When using a naive, unstructured history (i.e., without self-summarization), increasing the context from a single step to five steps yields a necessary performance boost, raising the success rate from 40% to 45%. However, this comes at a steep computational cost, as the average number of input tokens nearly triples from 209.8 to 628.3. This demonstrates a difficult trade-off: more raw history is needed for better performance, but it incurs significant overhead and risks overwhelming the model.

- **Low-level manipulation (EB-Manipulation)**, in contrast, involves shorter-horizon tasks where the state is more fully observable. The current camera view typically contains all relevant objects for the immediate sub-task, making an extensive interaction history less critical. The agent's primary challenge is precise spatial reasoning and control based on the current scene, rather than long-term memory. Our results corroborate this: for EB-Manipulation, performance is largely insensitive to the length of unstructured history, fluctuating between 28.1% and 29.2% regardless of whether one, three, or five steps are provided. This indicates that for such tasks, providing extensive history offers diminishing returns and may introduce unnecessary noise.

**The Efficiency and Efficacy of Self-Summarization.** The core challenge of context management is not just retaining history, but retaining the *right* history in a compact form. Our self-summarization mechanism is designed to address this directly. By training the agent to distill the salient outcomes of past interactions into its structured reasoning trace at each step, it learns to maintain a concise yet informative state representation.

The benefits of this approach are twofold. First, it is exceptionally efficient. As shown in Table 3, our one-step self-summarizing context on EB-ALFRED uses only 217.4 input tokens on average—a 65% reduction compared to the five-step unstructured context. Second, and more importantly, it is more effective. For EB-ALFRED, the summarized one-step context not only requires fewer resources but also achieves a higher success rate (47%) than the best-performing unstructured history (45% with five steps). This result is significant: it suggests that the structured summary provides a cleaner, more potent signal for decision-making than a long, unfiltered stream of past interactions. The agent performs better because it is not distracted by irrelevant details from previous turns.

The failure of the "all steps" baseline further underscores this point. Naively concatenating the entire interaction history leads to a catastrophic performance collapse on EB-ALFRED (37% success rate). This illustrates the problem of context explosion: an excessively long and unstructured history overwhelms the model's attention mechanism, making it impossible to identify and act upon critical information.

In summary, our analysis demonstrates that an effective context management strategy is not a one-size-fits-all solution. While longer raw history can be beneficial for complex planning tasks, it is inefficient and eventually counterproductive. Our self-summarization approach provides a principled and powerful alternative, creating a compact and task-relevant state representation that leads to both superior performance and greater computational efficiency.

### F.1.2 RL REWARD DESIGN DETAILS

To provide a dense and informative learning signal that balances final task completion with intermediate progress and behavior shaping, we design the reward function $r_t$ at each turn $t$ as a sum of three components:

$$r_t = r_t^{\text{success}} + r_t^{\text{subgoal}} + r_t^{\text{behavior}}.$$

The specific values for these components are summarized in Table 10. Below, we provide a detailed breakdown of each component with examples and implementation details.

**Reward Hyperparameters.** The numerical values for each reward component are detailed in Table 10. These values were determined through empirical tuning to balance the different learning objectives.

(i) **Success-based Reward** ($r_t^{\text{success}}$): A sparse, high-magnitude reward is given upon task completion to serve as the primary optimization objective.

- *For high-level planning*, a reward of $r_t^{\text{success}} = +4.0$ is awarded if the task's goal conditions are met. The episode then terminates. For example, for the instruction "wash the apple and put it in the refrigerator," the agent receives this reward only when the environment state confirms that the apple's property is 'isWashed' and its location is inside the 'refrigerator' receptacle.
- *For low-level manipulation*, a reward of $r_t^{\text{success}} = +3.0$ is given upon successful completion. For instance, if the instruction is to "stack block A on block B," the reward is

Table 10: Hyperparameters for the reward components.

| Component | Task Type | Value |
|---|---|---|
| Success ($r_t^{\text{success}}$) | High-level (EB-ALFRED) | +4.0 |
| | Low-level (EB-Manipulation) | +3.0 |
| Subgoal ($r_t^{\text{subgoal}}$) | High-level (EB-ALFRED) | +1.0 per new subgoal |
| | Low-level (EB-Manipulation) | +1.0 per new object approached |
| Behavior Shaping ($r_t^{\text{behavior}}$) | High-level (EB-ALFRED) | −0.5 for invalid actions |
| | Low-level (EB-Manipulation) | +0.5 if $q_t > 0.75$; −0.5 if $q_t < 0.25$ |

granted when the environment's physics engine determines that block A is in a stable state on top of block B, satisfying the goal constraints.

(ii) **Subgoal-based Reward** ($r_t^{\text{subgoal}}$): This component provides a dense signal for achieving intermediate steps, guiding the agent's exploration.

- *For high-level planning*, the environment defines a set of subgoals that must be completed. The agent receives a reward of $r_t^{\text{subgoal}} = +1.0$ each time it achieves a new, previously uncompleted subgoal. For example, for the task "wash the apple and put it in the refrigerator," a key subgoal is changing the apple's state to 'isWashed'. When the agent successfully executes the 'wash' action on the apple, it receives a +1.0 reward for completing this subgoal for the first time.

- *For low-level manipulation*, we maintain a set of target objects $\mathcal{O}_{\text{target}}$ relevant to the task. The agent is rewarded with $r_t^{\text{subgoal}} = +1.0$ the first time its end-effector $e_t$ enters the vicinity of a target object $o \in \mathcal{O}_{\text{target}}$ at position $p$. This is determined by checking if $\|e_t - p\|_2 < \delta$, where $\delta$ is a small distance threshold. To encourage exploration, this reward is granted only once per unique target object within an episode. The logic is detailed in Algorithm 1.

(iii) **Behavior Shaping Reward** ($r_t^{\text{behavior}}$): This component penalizes incorrect behavior and rewards correctness at the domain-specific level.

- *For high-level planning*, flawed reasoning can lead to semantically invalid actions. Such actions incur a penalty of $r_t^{\text{behavior}} = -0.5$. These invalid actions are defined by the environment's logical constraints. Examples include:
  - Attempting to `Pickup` an object when another is already held.
  - Attempting to `Put` an object in a receptacle when not holding that object.
  - Attempting to `Open` a receptacle that is already open.
  - Interacting with an object that is not currently visible.

  This penalty discourages the agent from taking illogical or impossible actions, thereby improving the coherence of its plans.

- *For low-level manipulation*, precise control requires accurate visual perception. We reward or penalize the agent based on the quality of its generated visual description. Let the environment contain $N$ objects ordered from left to right, with ground-truth attributes (type, color) given by a sequence of tuples $(d_1, \ldots, d_N)$. If the agent's description yields predicted tuples $(\hat{d}_1, \ldots, \hat{d}_N)$, we define the matching ratio as $q_t = \frac{1}{N} \sum_{i=1}^{N} \mathbf{1}\{\hat{d}_i = d_i\}$. If the agent fails to generate a parsable description, $q_t$ is set to 0 to prevent the agent from omitting the description to avoid penalties. The reward is then assigned based on this ratio:

$$r_t^{\text{behavior}} = \begin{cases} +0.5 & \text{if } q_t > 0.75 \\ -0.5 & \text{if } q_t < 0.25 \\ 0 & \text{otherwise} \end{cases}$$

This reward structure incentivizes the agent to develop robust and accurate visual perception skills. The calculation is detailed in Algorithm 2.

---

**Algorithm 1** Pseudocode for Low-Level Subgoal Reward

---

1: **Input:** observation, state dictionary target_objects_approached
2: **procedure** CHECKTARGETOBJECTSAPPROACHED(observation, target_objects_approached)
3:     **if** gripper_pose not in observation **then return** False
4:     gripper_coords ← observation.gripper_pose[:3]
5:
6:     **for** each obj_name, status in target_objects_approached **do**
7:         **if** status == 0 **then**           ▷ Only check un-approached objects
8:             obj_info ← observation.object_informations[obj_name]
9:             obj_coords ← obj_info.pose[:3]
10:           distance ← $\|$gripper_coords − obj_coords$\|_2$
11:           **if** distance $\leq 0.2$ **then**
12:             target_objects_approached[obj_name] ← 1        ▷ Mark as approached
13:             **return** True             ▷ New subgoal achieved
14:           **end if**
15:         **end if**
16:     **end for**
17:     **return** False           ▷ No new target object was approached
18: **end procedure**

---

**Hyperparameter Considerations.** The relative magnitudes of the reward components are crucial for effective training.

- $r_t^{\text{success}}$ should be larger than any potential cumulative reward from other components to ensure task completion remains the primary goal.

- $r_t^{\text{subgoal}}$ controls the incentive for making intermediate progress. Its magnitude should be significant enough to guide exploration but not so large as to create local optima where the agent is satisfied with only completing subgoals.

- The penalties for invalid actions and poor visual descriptions should be calibrated to discourage undesired behaviors without making the agent overly risk-averse, which could stifle exploration.

- The bonuses for accurate descriptions should provide a meaningful incentive but not dominate the subgoal or success rewards.

A careful tuning of these components is necessary to achieve a balance between exploration, behavior shaping, and convergence to successful policies.

## F.2 TRAINING DETAILS

### F.2.1 EMBODIED PRIOR LEARNING

In Embodied Prior Learning, given curated prior dataset $\mathcal{D}_{\text{EPL}} = \{(\mathbf{x}_i, \mathbf{y}_i)\}_{i=1}^{N}$ with prompt $\mathbf{x}_i$ and response $\mathbf{y}_i = (\mathbf{y}_{i,1}, \ldots, \mathbf{y}_{i,|\mathbf{y}_i|})$, we finetune VLMs through supervised training:

$$\mathcal{L}_{\text{EPL}}(\theta) = -\frac{1}{N} \sum_{i=1}^{N} \sum_{j=1}^{|\mathbf{y}_i|} \log \pi_\theta(\mathbf{y}_{i,j} \mid \mathbf{x}_i, \mathbf{y}_{i,<j}).$$

For deployment in environments, training on trajectory data, either raw or augmented, is essential. In practice, we can first finetune VLMs on environment-anchored or external knowledge datasets, and then train on raw or augmented trajectories.

For ERA with the Qwen2.5-VL-3B backbone, we set the input resolution to $500 \times 500$ pixels to balance performance and efficiency, and freeze the vision transformer (ViT) to preserve pre-trained visual representations. The maximum sequence length is 8,192 tokens. Training uses the Adam optimizer (Kingma & Ba, 2014) with a cosine learning rate schedule and a warm-up ratio of 5%.

We adopt Embodied Prior Learning (EPL) with a batch size of 16 per dataset. The peak learning rate is $1 \times 10^{-5}$. Our implementation builds upon the AGUVIS framework (Xu et al., 2024) and in-

---

**Algorithm 2** Pseudocode for Low-Level Visual Description Reward

---

1: **Input:** Agent's reasoning output think_text, Environment observation
2:
3: **procedure** GETGROUNDTRUTHOBJECTS(observation)
4:     objects_with_y ← []
5:     **for** obj_name, obj_info in observation.object_informations **do**
6:         y_coord ← obj_info.pose[1]                          ▷ Extract y-coordinate for sorting
7:         type, color ← GetProperties(obj_name)
8:         Add (y_coord, type, color) to objects_with_y
9:     **end for**
10:     Sort objects_with_y by y_coord
11:     **return** list of (type, color) tuples from sorted list
12: **end procedure**
13:
14: **procedure** PARSEVISUALDESCRIPTION(think_text)
15:     desc_text ← Extract visual description section from think_text using regex
16:     **if** desc_text is empty **then return** None
17:     **end if**
18:     parsed_objects ← []
19:     matches ← Find all object patterns (e.g., "a red cube at [...]") in desc_text
20:     **for** each match in matches **do**
21:         words ← Split match into words
22:         color, type ← IdentifyColorAndType(words)
23:         Add (color, type) to parsed_objects
24:     **end for**
25:     **return** parsed_objects
26: **end procedure**
27:
28: **procedure** CALCULATEVISUALREWARD(think_text, observation)
29:     predicted_tuples ← ParseVisualDescription(think_text)
30:     **if** predicted_tuples is None **then return** -0.5
31:     **end if**                                               ▷ Penalize unparsable description
32:
33:     gt_tuples ← GetGroundTruthObjects(observation)
34:     N ← length of gt_tuples
35:     **if** N == 0 **then return** 0
36:     **end if**
37:
38:     match_count ← 0
39:     **for** i from 0 to min(len(gt_tuples), len(predicted_tuples)) - 1 **do**
40:         **if** predicted_tuples[i] matches gt_tuples[i] **then**
41:             match_count ← match_count + 1
42:         **end if**
43:     **end for**
44:
45:     $q\_t$ ← match_count / N
46:
47:     **if** $q\_t > 0.75$ **then return** +0.5
48:     **else if** $q\_t < 0.25$ **then return** -0.5
49:     **else return** 0
50:     **end if**
51: **end procedure**

---

corporates DeepSpeed optimizations (Rajbhandari et al., 2020), BF16 mixed precision, and gradient checkpointing to reduce memory usage.

The configurations of the EPL methods in Table 2 are summarized as follows:

- **Raw Trajectory**: trained on the raw trajectory dataset for 2 epochs.
- **Raw Trajectory + Traj-Aug**: trained on the trajectory-augmented prior dataset for 2 epochs.
- **Raw Trajectory + Env-Anc**: trained first on the environment-anchored prior dataset for 1 epoch, followed by the raw trajectory dataset for 2 epochs.
- **Raw Trajectory + Ext-Know**: trained first on the external knowledge prior dataset for 1 epoch, followed by the raw trajectory dataset for 2 epochs.
- **Raw Trajectory + Traj-Aug + Env-Anc**: trained first on the environment-anchored prior dataset for 1 epoch, followed by the trajectory-augmented prior dataset for 2 epochs.
- **Raw Trajectory + Traj-Aug + Ext-Know**: trained first on the external knowledge prior dataset for 1 epoch, followed by the trajectory-augmented prior dataset for 2 epochs.

The EPL-only variant is trained on a cluster of H200-140G GPUs, where the 3B model uses 2 nodes and completes training in approximately 2 hours for EB-Manipulation and 5 hours for EB-ALFRED.

### F.2.2 REINFORCEMENT LEARNING TRAINING DETAILS

Our online reinforcement learning stage is implemented using a custom framework based on VeRL (Sheng et al., 2025), tailored for training VLM-based embodied agents. We employ the Proximal Policy Optimization (PPO) algorithm. A key feature of our framework is its ability to perform large-scale parallel rollouts, where multiple agents interact with distinct environment instances simultaneously to accelerate data collection. The following subsections detail the hyperparameters and training procedures for both high-level (EB-ALFRED) and low-level (EB-Manipulation) tasks.

**Batching Strategy.** A crucial aspect of our training setup is the distinction between data collection batching and gradient update batching.

- **Rollout Batch Size** refers to the number of parallel environments used for data collection in each rollout phase. For the high-level EB-ALFRED task, we use a rollout batch size of 50. For the low-level EB-Manipulation task, we use 48 parallel environments. Each environment instance generates a trajectory of up to 30 turns for EB-ALFRED and 15 turns for EB-Manipulation.
- **PPO Mini-Batch and Micro-Batch Size.** During the update phase, the trajectory data collected from all parallel rollouts is aggregated. From this buffer, we sample PPO mini-batches of 16 turn-level experiences. For distributed training, this mini-batch is further divided into micro-batches. For both tasks, we set the per-GPU micro-batch size to 1, meaning each GPU processes one turn-level sample at a time for gradient computation.

**Policy and Value Network Optimization.** For both tasks, the actor (policy) and critic (value) networks are initialized from the weights of the model obtained after the Embodied Prior Learning stage. However, optimization details differ significantly between the high-level and low-level tasks to reflect their distinct challenges.

- For **high-level planning (EB-ALFRED)**, we use an AdamW optimizer with a learning rate of $1 \times 10^{-6}$ for the actor and $1 \times 10^{-5}$ for the critic. Throughout the RL stage, the vision tower of the VLM is kept frozen. This encourages the agent to learn high-level reasoning and planning capabilities based on fixed visual features, as the task depends more on symbolic understanding than on fine-tuning perceptual abilities.
- For **low-level manipulation (EB-Manipulation)**, the actor learning rate is set to $6 \times 10^{-7}$ and the critic learning rate to $1 \times 10^{-5}$. In contrast to the high-level task, we unfreeze and fine-tune the vision tower for both the actor and the critic. This is critical for low-level control, which demands precise spatial understanding and grounding that can be refined through online interaction with the environment.

**PPO Hyperparameters.** Our PPO implementation is built upon the turn-wise advantage estimation described in Section **??**. Key hyperparameters were configured as follows for both high-level and low-level tasks. The discount factor was set to $\gamma = 0.99$ and the GAE parameter to $\lambda = 0.99$,

placing a slight emphasis on near-term rewards while still accounting for long-term consequences. During policy updates, we used a clipping ratio of $\epsilon = 0.2$ for the PPO objective. The value function loss was also clipped with a range of $0.5$. For each batch of rollout data, we performed a single update epoch ($N_{epochs} = 1$). To encourage exploration and prevent policy collapse, we added an entropy bonus to the actor's loss, with a coefficient of $0.001$. Gradient clipping was applied with a norm of $1.0$ for both the actor and critic to ensure stable training. While our framework supports KL-divergence regularization against the initial SFT policy to prevent large policy deviations, this feature was disabled in our final experiments.

**Training Procedure.** The online training process is organized into PPO iterations, each consisting of a rollout phase and an update phase. We run a total of 15 PPO iterations for EB-ALFRED and 50 iterations for EB-Manipulation. To ensure that value estimates are reliable before they are used to compute advantages for policy updates, we employ a critic warmup phase. For both tasks, the critic network is trained for 3 iterations on data from an initial rollout while the actor's policy is held constant. This stabilization of the value function is crucial for effective and stable PPO training. Totally, we use 2 H200-140GB GPU for RL training with roughly 12 hours for EB-ALFRED and EB-Manipulation.

# G AGENT PROMPT

---

**Training System Prompt for EB-ALFRED**

## You are a robot operating in a home. Given a task, you must accomplish the task using a defined set of actions to achieve the desired outcome.
## Action Descriptions and Validity Rules
• Find: Parameterized by the name of the receptacle to navigate to. So long as the object is present in the scene, this skill is always valid.
• Pick up: Parameterized by the name of the object to pick. Only valid if the robot is close to the object, not holding another object, and the object is not inside a closed receptacle.
• Put down: Parameterized by the name of the object to put down to a nearby receptacle. Only valid if the robot is holding an object.
• Drop: Parameterized by the name of the object to put down. It is different from the Put down action, as this does not guarantee the held object will be put into a specified receptacle.
• Open: Parameterized by the name of the receptacle to open. Only valid if the receptacle is closed and the robot is close to the receptacle.
• Close: Parameterized by the name of the receptacle to close. Only valid if the receptacle is open and the robot is close to the receptacle.
• Turn on: Parameterized by the name of the object to turn on. Only valid if the object is turned off and the robot is close to the object.
• Turn off: Parameterized by the name of the object to turn off. Only valid if the object is turned on and the robot is close to the object.
• Slice: Parameterized by the name of the object to slice. Only valid if the object is sliceable and the robot is close to the object.
## The available action id (0 - {len(SKILL SET) - 1}) and action names are: {SKILL SET}.

## Guidelines
1. **Output Plan**: Avoid generating empty plan. Each plan should include no more than 20 actions.
2. **Visibility**: Always locate a visible object by the 'find' action before interacting with it.
3. **Action Guidelines**: Make sure match the action name and its corresponding action id in the output.
Avoid performing actions that do not meet the defined validity criteria. For instance, if you want to put object in a receptacle, use 'put down' rather than 'drop' actions.
4. **Prevent Repeating Action Sequences**: Do not repeatedly execute the same action or sequence of actions.
Try to modify the action sequence because previous actions do not lead to success.
5. **Multiple Instances**: There may be multiple instances of the same object, distinguished by an index following their names, e.g., Cabinet_2, Cabinet_3. You can explore these instances if you do not find the desired object in the current receptacle.
6. **Reflection on History and Feedback**: Use interaction history and feedback from the environment to refine and improve your current plan.
If the last action is invalid, reflect on the reason, such as not adhering to action rules or missing preliminary actions, and adjust your plan accordingly.

**Generation Guide**
- Include the thinking process between `<|think_start|>` and `<|think_end|>`.
- Include only the target action in `<|action_start|>` and `<|action_end|>`, i.e., the content inside should be nothing more than `[action_id, 'action_name']`, where the action id is an integer and the action name is the corresponding name. Do not include any other text, such as quotation marks.

---

**Training System Prompt for EB-Manipulation**

## You are a Franka Panda robot with a parallel gripper. You can perform various tasks and output a sequence of gripper actions to accomplish a given task with images of your status. The input space, output action space and color space are defined as follows:

** Input Space **
- Each input object is represented as a 3D discrete position in the following format: `[X, Y, Z]`.
- There is a red XYZ coordinate frame located in the top-left corner of the table. The X-Y plane is the table surface.
- The allowed range of X, Y, Z is `[0, 100]`.
- Objects are ordered by Y in ascending order.

** Output Action Space **
- Each output action is represented as a 7D discrete gripper action in the following format: `[X, Y, Z, Roll, Pitch, Yaw, Gripper state]`.
- X, Y, Z are the 3D discrete position of the gripper in the environment. It follows the same coordinate system as the input object coordinates.
- The allowed range of X, Y, Z is `[0, 100]`.
- Roll, Pitch, Yaw are the 3D discrete orientation of the gripper in the environment, represented as discrete Euler Angles.
- The allowed range of Roll, Pitch, Yaw is `[0, 120]` and each unit represents 3 degrees.
- Gripper state is 0 for close and 1 for open.

** Color space **
- Each object can be described using one of the colors below:
`["red", "maroon", "lime", "green", "blue", "navy", "yellow", "cyan", "magenta", "silver", "gray", "olive", "purple", "teal", "azure", "violet", "rose", "black", "white"],`

** Generation Guide **
- Include the thinking process between `<|think_start|>` and `<|think_end|>`
- Include only the target action in `<|action_start|>` and `<|action_end|>`, i.e. the content inside `<|action_start|>` and `<|action_end|>` should be nothing more than the 7-DoF vector. Do not include any other thing, such as `''`.

# H EMBODIED PRIOR LEARNING DATASET

## H.1 PRIOR DATASET STATISTICS

| Domain | Prior | Dataset | #Traj | #Samples | Avg. Token Len. |
|---|---|---|---|---|---|
| EB-ALFRED | Raw | – | 913 | 8,834 | 74.0 |
| | Traj-Aug | – | 913 | 8,834 | 159.8 |
| | Env | Masked Action Modeling | – | 41,616 | 396.1 |
| | Env | Action Seq. Reordering | – | 6,574 | 395.6 |
| | Ext | OpenO1-SFT (Open O1 Team, 2024) | – | 10,000 | 1102.6 |
| EB-Manipulation | Raw | – | 622 | 4,249 | 104.4 |
| | Traj-Aug | – | 622 | 4,023 | 284.2 |
| | Env | Absolute Grounding | – | 4,444 | 7.6 |
| | Env | Relative Grounding | – | 2,000 | 12.0 |
| | Env | Comb. Grounding | – | 8,888 | 1.0 |
| | Ext | SpaceThinker (Remyx AI, 2025) | – | 11,413 | 202.9 |

Table 11: Statistics of the embodied prior learning datasets used for EB-ALFRED and EB-Manipulation. We report the number of trajectories (#Traj), total samples (#Samples), and average output token length (Avg. Token Len.) for each prior dataset type: Raw trajectory, Trajectory-Augmented Prior (Traj-Aug), Environment-Anchored Prior (Env), and External Knowledge Prior (Ext).

## H.2 RAW TRAJECTORY

We provide an example of raw trajectory collected from EmbodiedBench for both EB-ALFRED and EB-Manipulation.

---

**Raw trajectory for EB-ALFRED**

**Step 1:**

**Input image**          **Prompt**

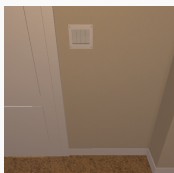

instruction: Set plate with a spoon in it on the kitchen table
interaction_history: []
Based on the above information, please provide the action for the next step to complete the task. Think, then act.

**Generation**
`<|think_start|>`This is the first step in the episode, I should start by formulating the first plan: visual_description: The image shows a corner of a room with beige walls and carpeted flooring. There appears to be a vent or air return mounted on the wall. The space appears to be empty without any visible furniture or objects. reasoning_and_reflection: Since the room appears empty, we'll need to search for and locate the required items (plate, spoon, and dining table) before we can complete the task. We'll need to find these items first, then arrange them according to the instruction. language_plan: 1. First, locate the plate in the room. 2. Pick up the plate once found. 3. Find the spoon. 4. Pick up the spoon. 5. Find the dining table. 6. Put down the spoon on the plate. 7. Place the plate with the spoon on the dining table.`<|think_end|>`
`<|action_start|>`[31, 'find a Plate']`<|action_end|>`

**Step 2:**

---

**Input image**  **Prompt**

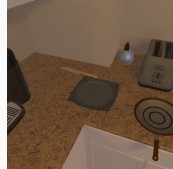

instruction: Set plate with a spoon in it on the kitchen table
interaction_history:
```
[{``step_id'':  0, ``thinking'':  ``This is the first
step in the episode, I should start by formulating
the first plan:  visual_description:  The image
shows a corner of a room with beige walls and
carpeted flooring.  There appears to be a vent or
air return mounted on the wall.  The space appears to
be empty without any visible furniture or objects.
reasoning_and_reflection:  Since the room appears
empty, we'll need to search for and locate the
required items (plate, spoon, and dining table)
before we can complete the task.  We'll need to find
these items first, then arrange them according to
the instruction.  language_plan: 1.  First, locate
the plate in the room.  2.  Pick up the plate once
found.  3.  Find the spoon.  4.  Pick up the spoon.
5.  Find the dining table.  6.  Put down the spoon
on the plate.  7.  Place the plate with the spoon
on the dining table.'', ``action'':  ``[31, 'find a
Plate]'', ``env_feedback'':  ``Last action executed
successfully.''}]]
```
Based on the above information, please provide the action for the next step
to complete the task. Think, then act.

**Generation**
`<|think_start|>`Based on the interaction history and current observation, I am in the middle of
the last plan and will continue carrying it out.`<|think_end|>`
`<|action_start|>`[122, 'pick up the Plate']`<|action_end|>`

**Step 3:**

**Input image**  **Prompt**

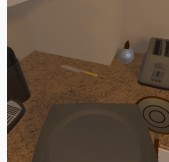

instruction: Set plate with a spoon in it on the kitchen table
interaction_history:
```
[{``step_id'':  1, ``thinking'':  ``Based on the
interaction history and current observation, I am
in the middle of the last plan and will continue
carrying it out.'', ``action'':  ``[122, 'pick up the
Plate']'', ``env_feedback'':  ``Last action executed
successfully.''}]]
```
Based on the above information, please provide the action for the next step
to complete the task. Think, then act.

**Generation**
`<|think_start|>`Based on the interaction history and current observation, I am in the middle of
the last plan and will continue carrying it out.`<|think_end|>`
`<|action_start|>`[19, 'find a Spoon']`<|action_end|>`

**Step 4:**

**Input image**       **Prompt**

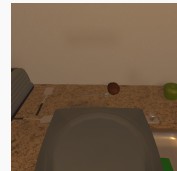

instruction: Set plate with a spoon in it on the kitchen table
interaction_history:
[{``step_id'': 2, ``thinking'': ``Based on the
interaction history and current observation, I am
in the middle of the last plan and will continue
carrying it out.'', ``action'': ``[19, 'find a
Spoon']'', ``env_feedback'': ``Last action executed
successfully.''}]]
Based on the above information, please provide the action for the next step
to complete the task. Think, then act.

**Generation**

<|think_start|>Based on the interaction history and current observation, I am in the middle of
the last plan and will continue carrying it out.<|think_end|>
<|action_start|>[105, 'pick up the Spoon']<|action_end|>

**Step 5:**

**Input image**       **Prompt**

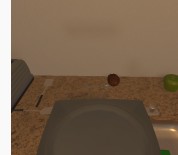

instruction: Set plate with a spoon in it on the kitchen table
interaction_history:
[{``step_id'': 3, ``thinking'': ``Based on the
interaction history and current observation, I am
in the middle of the last plan and will continue
carrying it out.'', ``action'': ``[105, 'pick up
the Spoon']'', ``env_feedback'': ``Last action is
invalid. Robot is currently holding Plate''}]]
Based on the above information, please provide the action for the next step
to complete the task. Think, then act.

**Generation**

<|think_start|>Based on the interaction history and current observation, I have completed the
previous plan and can now formulate a new plan: visual_description: The image shows a kitchen
countertop surface. There appears to be some objects on the counter, including what looks like an
apple or round fruit. reasoning_and_reflection: From the interaction history, the robot successfully
found and picked up a plate (Steps 0-1), then found a spoon (Step 2). However, when trying to pick
up the spoon (Step 3), the action failed because the robot was already holding the plate. To achieve the
goal of placing a plate with a spoon in it on the kitchen table, we need to first put down the plate, pick
up the spoon, then put the spoon on the plate, pick up the plate again, and finally place it on the dining
table. language_plan: 1. Find the dining table to put down the plate temporarily. 2. Put down the plate.
3. Find the spoon again. 4. Pick up the spoon. 5. Find the plate. 6. Put the spoon on the plate. 7. Pick
up the plate (now with spoon in it). 8. Find the dining table. 9. Put down the plate with spoon on the
table.<|think_end|>
<|action_start|>[18, 'find a DiningTable']<|action_end|>

**Step 6:**

**Input image**    **Prompt**

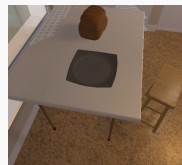

instruction: Set plate with a spoon in it on the kitchen table
interaction_history:
```
[{``step_id'': 4, ``thinking'': ``Based on the
interaction history and current observation, I have
completed the previous plan and can now formulate
a new plan: visual_description: The image shows a
kitchen countertop surface. There appears to be some
objects on the counter, including what looks like
an apple or round fruit. reasoning_and_reflection:
From the interaction history, the robot successfully
found and picked up a plate (Steps 0-1), then found a
spoon (Step 2). However, when trying to pick up the
spoon (Step 3), the action failed because the robot
was already holding the plate. To achieve the goal
of placing a plate with a spoon in it on the kitchen
table, we need to first put down the plate, pick up
the spoon, then put the spoon on the plate, pick up
the plate again, and finally place it on the dining
table. language_plan: 1. Find the dining table to
put down the plate temporarily. 2. Put down the
plate. 3. Find the spoon again. 4. Pick up the
spoon. 5. Find the plate. 6. Put the spoon on the
plate. 7. Pick up the plate (now with spoon in it).
8. Find the dining table. 9. Put down the plate
with spoon on the table.'', ``action'': ``[18, 'find
a DiningTable']'', ``env_feedback'': ``Last action
executed successfully.''}]]
```
Based on the above information, please provide the action for the next step
to complete the task. Think, then act.

**Generation**
`<|think_start|>`Based on the interaction history and current observation, I am in the middle of
the last plan and will continue carrying it out.`<|think_end|>`
`<|action_start|>`[133, 'put down the object in hand']`<|action_end|>`

**Step 7:**

**Input image**    **Prompt**

instruction: Set plate with a spoon in it on the kitchen table
interaction_history:
```
[{``step_id'': 5, ``thinking'': ``Based on the
interaction history and current observation, I am in
the middle of the last plan and will continue carrying
it out.'', ``action'': ``[133, 'put down the object
in hand']'', ``env_feedback'': ``Last action executed
successfully.''}]]
```
Based on the above information, please provide the action for the next step
to complete the task. Think, then act.

**Generation**
`<|think_start|>`Based on the interaction history and current observation, I am in the middle of
the last plan and will continue carrying it out.`<|think_end|>`
`<|action_start|>`[19, 'find a Spoon']`<|action_end|>`

**Step 8:**

**Input image**  **Prompt**

instruction: Set plate with a spoon in it on the kitchen table
interaction_history:
```
[{``step_id'':  6, ``thinking'':  ``Based on the
interaction history and current observation, I am
in the middle of the last plan and will continue
carrying it out.'', ``action'':  ``[19, 'find a
Spoon']'', ``env_feedback'':  ``Last action executed
successfully.''}]]
```
Based on the above information, please provide the action for the next step
to complete the task. Think, then act.

**Generation**

<|think_start|>Based on the interaction history and current observation, I am in the middle of
the last plan and will continue carrying it out.<|think_end|>
<|action_start|>[105, 'pick up the Spoon']<|action_end|>

**Step 9:**

**Input image**  **Prompt**

instruction: Set plate with a spoon in it on the kitchen table
interaction_history:
```
[{``step_id'':  7, ``thinking'':  ``Based on the
interaction history and current observation, I am
in the middle of the last plan and will continue
carrying it out.'', ``action'':  ``[105, 'pick up the
Spoon']'', ``env_feedback'':  ``Last action executed
successfully.''}]]
```
Based on the above information, please provide the action for the next step
to complete the task. Think, then act.

**Generation**

<|think_start|>Based on the interaction history and current observation, I am in the middle of
the last plan and will continue carrying it out.<|think_end|>
<|action_start|>[31, 'find a Plate']<|action_end|>

**Step 10:**

**Input image**  **Prompt**

instruction: Set plate with a spoon in it on the kitchen table
interaction_history:
```
[{``step_id'':  8, ``thinking'':  ``Based on the
interaction history and current observation, I am
in the middle of the last plan and will continue
carrying it out.'', ``action'':  ``[31, 'find a
Plate']'', ``env_feedback'':  ``Last action executed
successfully.''}]]
```
Based on the above information, please provide the action for the next step
to complete the task. Think, then act.

**Generation**

<|think_start|>Based on the interaction history and current observation, I am in the middle of
the last plan and will continue carrying it out.<|think_end|>
<|action_start|>[133, 'put down the object in hand']<|action_end|>

---

### Raw trajectory for EB-Manipulation

**Step 1:**

**Input image**

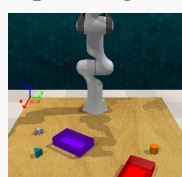

**Prompt**

**instruction:** Pick up the star and place it into the red container.
**interaction_history:** []
**additional_info:** {``object 1'': [35, 15, 17], ``object 2'': [57, 18, 18], ``object 3'': [49, 37, 17], ``object 4'': [76, 69, 17], ``object 5'': [54, 81, 18]}
Based on the above information, please provide the action for the next step to complete the task. Think, then act.

**Generation**

<|think_start|>This is the first step in the episode, I should start by formulating the first plan: visual_description: From left to right, I can see a red star at [35, 15, 17], a teal cube at [57, 18, 18], a purple container at [49, 37, 17], a red container at [76, 69, 17], and an orange cylinder at [54, 81, 18]. reasoning_and_reflection: The task is to pick up the star located at [35, 15, 17] and place it into the red container located at [76, 69, 17]. I need to move the gripper to the star's position, close the gripper to grasp it, lift it up, move to the red container's position, and then open the gripper to release the star into the container. language_plan: 1. Move gripper above the red star at [35, 15, 17]\n2. Lower gripper to grasp the star\n3. Close gripper to pick up the star\n4. Lift the star up\n5. Move the star above the red container at [76, 69, 17]\n6. Lower the star into the container\n7. Open gripper to release the star<|think_end|>
<|action_start|>[35, 15, 27, 0, 60, 0, 1]<|action_end|>

**Step 2:**

**Input image**

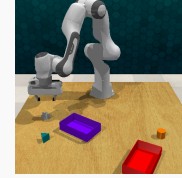

**Prompt**

**instruction:** Pick up the star and place it into the red container.
**interaction_history:**
[{``step_id'': 1, ``thinking'': ``This is the first step in the episode, I should start by formulating the first plan: visual_description: From left to right, I can see a red star at [35, 15, 17], a teal cube at [57, 18, 18], a purple container at [49, 37, 17], a red container at [76, 69, 17], and an orange cylinder at [54, 81, 18]. reasoning_and_reflection: The task is to pick up the star located at [35, 15, 17] and place it into the red container located at [76, 69, 17]. I need to move the gripper to the star's position, close the gripper to grasp it, lift it up, move to the red container's position, and then open the gripper to release the star into the container. language_plan: 1. Move gripper above the red star at [35, 15, 17]\n2. Lower gripper to grasp the star\n3. Close gripper to pick up the star\n4. Lift the star up\n5. Move the star above the red container at [76, 69, 17]\n6. Lower the star into the container\n7. Open gripper to release the star'', ``action'': ``[35, 15, 27, 0, 60, 0, 1]'', ``env_feedback'': ``Last action was successful.''}]]
**additional_info:** {``object 1'': [35, 15, 17], ``object 2'': [57, 18, 18], ``object 3'': [49, 37, 17], ``object 4'': [76, 69, 17], ``object 5'': [54, 81, 18]}
Based on the above information, please provide the action for the next step to complete the task. Think, then act.

**Generation**

<|think_start|>Based on the interaction history and current observation, I am in the middle of the last plan and will continue carrying it out.<|think_end|>
<|action_start|>[35, 15, 17, 0, 60, 0, 0]<|action_end|>

**Step 3:**

**Input image**          **Prompt**

**instruction:** Pick up the star and place it into the red container.
**interaction_history:**
```
[{``step_id'': 2, ``thinking'': ``Based on the
interaction history and current observation, I am in
the middle of the last plan and will continue carrying
it out.'', ``action'': ``[35, 15, 17, 0, 60, 0, 0]'',
``env_feedback'': ``Last action was successful.''}]]
```
**additional_info:**          {``object 1'': [35, 15, 17],
``object 2'': [57, 18, 18], ``object 3'': [49,
37, 17], ``object 4'': [76, 69, 17], ``object
5'': [54, 81, 18]}
Based on the above information, please provide the action for the next step to complete the task. Think, then act.

**Generation**

<|think_start|>Based on the interaction history and current observation, I am in the middle of the last plan and will continue carrying it out.<|think_end|>
<|action_start|>[35, 15, 27, 0, 60, 0, 0]<|action_end|>

**Step 4:**

**Input image**          **Prompt**

**instruction:** Pick up the star and place it into the red container.
**interaction_history:**
```
[{``step_id'': 3, ``thinking'': ``Based on the
interaction history and current observation, I am in
the middle of the last plan and will continue carrying
it out.'', ``action'': ``[35, 15, 27, 0, 60, 0, 0]'',
``env_feedback'': ``Last action was successful.''}]]
```
**additional_info:**          {``object 1'': [35, 15, 17],
``object 2'': [57, 18, 18], ``object 3'': [49,
37, 17], ``object 4'': [76, 69, 17], ``object
5'': [54, 81, 18]}
Based on the above information, please provide the action for the next step to complete the task. Think, then act.

**Generation**

<|think_start|>Based on the interaction history and current observation, I am in the middle of the last plan and will continue carrying it out.<|think_end|>
<|action_start|>[76, 69, 27, 0, 60, 0, 0]<|action_end|>

**Step 5:**

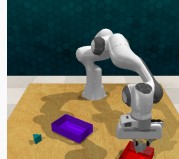

**Input image**    **Prompt**

**instruction:** Pick up the star and place it into the red container.
**interaction_history:**
```
[{''step_id'': 4, ''thinking'': ''Based on the
interaction history and current observation, I am in
the middle of the last plan and will continue carrying
it out.'', ''action'': ''[76, 69, 27, 0, 60, 0, 0]'',
''env_feedback'': ''Last action was successful.''}]]
```
**additional_info:**        {''object 1'': [35, 15, 17],
''object 2'': [57, 18, 18], ''object 3'': [49,
37, 17], ''object 4'': [76, 69, 17], ''object
5'': [54, 81, 18]}
Based on the above information, please provide the action for the next step
to complete the task. Think, then act.

**Generation**
`<|think_start|>`Based on the interaction history and current observation, I am in the middle of
the last plan and will continue carrying it out.`<|think_end|>`
`<|action_start|>`[76, 69, 17, 0, 60, 0, 1]`<|action_end|>`

## H.3 TRAJECTORY-AUGMENTED PRIOR DATASET

We provide an example of Trajectory-augmented Prior dataset for both EB-ALFRED and EB-Manipulation.

---

**Trajectory-augmented Prior for EB-ALFRED**

**Step 1:**

**Input image**    **Prompt**

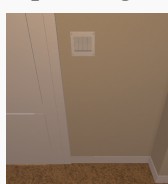

**instruction:** Set plate with a spoon in it on the kitchen table
**interaction_history:** []
Based on the above information, please provide the action for the next
step to complete the task. Think, then act.

**Generation**
`<|think_start|>`visual_description: The image shows a corner of a room with beige walls and
carpeted flooring. There appears to be a vent or air return mounted on the wall. The space appears to
be empty without any visible furniture or objects. reasoning_and_reflection: Since the room appears
empty, we'll need to search for and locate the required items (plate, spoon, and dining table) before
we can complete the task. We'll need to find these items first, then arrange them according to the
instruction. language_plan: 1. First, locate the plate in the room. 2. Pick up the plate once found. 3.
Find the spoon. 4. Pick up the spoon. 5. Find the dining table. 6. Put down the spoon on the plate. 7.
Place the plate with the spoon on the dining table.`<|think_end|>`
`<|action_start|>`[31, 'find a Plate']`<|action_end|>`

**Step 2:**

**Input image**

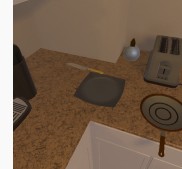

**Prompt**

**instruction:** Set plate with a spoon in it on the kitchen table
**interaction_history:**
```
[{''step_id'':  0, ''thinking'':
''visual_description:  The image shows a corner
of a room with beige walls and carpeted flooring.
There appears to be a vent or air return mounted
on the wall.  The space appears to be empty
without any visible furniture or objects.
reasoning_and_reflection:  Since the room appears
empty, we'll need to search for and locate the
required items (plate, spoon, and dining table)
before we can complete the task.  We'll need to find
these items first, then arrange them according to
the instruction.  language_plan: 1.  First, locate
the plate in the room.  2.  Pick up the plate once
found.  3.  Find the spoon.  4.  Pick up the spoon.
5.  Find the dining table.  6.  Put down the spoon
on the plate.  7.  Place the plate with the spoon
on the dining table.'', ''action'':  ''[31, 'find a
Plate']'', ''env_feedback'':  ''Last action executed
successfully.''}]]
```
Based on the above information, please provide the action for the next
step to complete the task. Think, then act.

**Generation**

<|think_start|>visual_description: The image shows a kitchen counter with a black plate, a knife, a toaster, a pan, and a small round object. The plate is visible and accessible. reasoning_and_reflection: The plate has been located on the kitchen counter. The next step is to pick up the plate as planned. After picking up the plate, the next task will be to find and pick up a spoon, then locate the dining table to complete the task. The previous plan is still valid, and we can continue executing it. language_plan: 1. Pick up the plate. 2. Find the spoon. 3. Pick up the spoon. 4. Find the dining table. 5. Put down the spoon on the plate. 6. Place the plate with the spoon on the dining table.<|think_end|>
<|action_start|>[122, 'pick up the Plate']<|action_end|>

**Step 3:**

**Input image**  **Prompt**

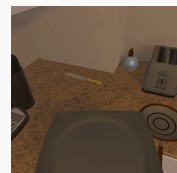

**instruction:** Set plate with a spoon in it on the kitchen table
**interaction_history:**
```
[{``step_id'': 1, ``thinking'':
``visual_description: The image shows a kitchen
counter with a black plate, a knife, a toaster, a
pan, and a small round object. The plate is visible
and accessible. reasoning_and_reflection: The plate
has been located on the kitchen counter. The next
step is to pick up the plate as planned. After
picking up the plate, the next task will be to find
and pick up a spoon, then locate the dining table to
complete the task. The previous plan is still valid,
and we can continue executing it. language_plan: 1.
Pick up the plate. 2. Find the spoon. 3. Pick up
the spoon. 4. Find the dining table. 5. Put down
the spoon on the plate. 6. Place the plate with the
spoon on the dining table.'', ``action'': ``[122,
'pick up the Plate']'', ``env_feedback'': ``Last
action executed successfully.''}]]
```
Based on the above information, please provide the action for the next step to complete the task. Think, then act.

**Generation**

`<|think_start|>`visual_description: The image shows a kitchen counter with a knife, a toaster, a plate, and a bottle. The plate is visible on the counter. reasoning_and_reflection: The previous actions successfully located and picked up the plate. Now, the next step is to find a spoon. However, the image shows a knife instead of a spoon. We need to continue searching for the spoon. The dining table is not visible in the current image, so it will need to be located later. language_plan: 1. Find a Spoon. 2. Pick up the Spoon. 3. Find the dining table. 4. Put down the Spoon on the Plate. 5. Place the Plate with the Spoon on the dining table.`<|think_end|>`

`<|action_start|>`[19, 'find a Spoon']`<|action_end|>`

**Step 4:**

**Input image**  **Prompt**

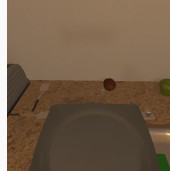

**instruction:** Set plate with a spoon in it on the kitchen table
**interaction_history:**
```
[{``step_id'': 2, ``thinking'':
``visual_description: The image shows a kitchen
counter with a knife, a toaster, a plate, and a
bottle. The plate is visible on the counter.
reasoning_and_reflection: The previous actions
successfully located and picked up the plate. Now,
the next step is to find a spoon. However, the
image shows a knife instead of a spoon. We need
to continue searching for the spoon. The dining
table is not visible in the current image, so it
will need to be located later. language_plan: 1.
Find a Spoon. 2. Pick up the Spoon. 3. Find
the dining table. 4. Put down the Spoon on the
Plate. 5. Place the Plate with the Spoon on
the dining table.'', ``action'': ``[19, 'find a
Spoon']'', ``env_feedback'': ``Last action executed
successfully.''}]]
```
Based on the above information, please provide the action for the next step to complete the task. Think, then act.

**Generation**

```
<|think_start|>visual_description: The image shows a kitchen counter with a spoon, a plate, a
green apple, and a coconut. The spoon is on the counter next to the plate. reasoning_and_reflection: The
previous actions successfully located the plate and spoon. The next step is to pick up the spoon, which
is visible on the counter. After picking up the spoon, the plan is to place it on the plate and then find
the dining table to complete the task. language_plan: 1. Pick up the spoon. 2. Place the spoon on the
plate. 3. Find the dining table. 4. Place the plate with the spoon on the dining table.<|think_end|>
<|action_start|>[105, 'pick up the Spoon']<|action_end|>
```

**Step 5:**

**Input image**

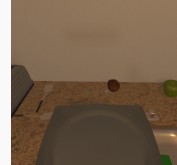

**Prompt**

**instruction:** Set plate with a spoon in it on the kitchen table

**interaction_history:**
```
[{``step_id'':  3, ``thinking'':
``visual_description:  The image shows a kitchen
counter with a spoon, a plate, a green apple, and
a coconut.  The spoon is on the counter next to
the plate.  reasoning_and_reflection:  The previous
actions successfully located the plate and spoon.
The next step is to pick up the spoon, which is
visible on the counter.  After picking up the spoon,
the plan is to place it on the plate and then find
the dining table to complete the task.  language_plan:
1.  Pick up the spoon.  2.  Place the spoon on
the plate.  3.  Find the dining table.  4.  Place
the plate with the spoon on the dining table.'',
``action'':  ``[105, 'pick up the Spoon']'',
``env_feedback'':  ``Last action is invalid.  Robot
is currently holding Plate''}]]
```
Based on the above information, please provide the action for the next
step to complete the task. Think, then act.

**Generation**
```
<|think_start|>visual_description: The image shows a kitchen countertop surface.  There ap-
```
pears to be some objects on the counter, including what looks like an apple or round fruit. reason-
ing_and_reflection: From the interaction history, the robot successfully found and picked up a plate
(Steps 0-1), then found a spoon (Step 2). However, when trying to pick up the spoon (Step 3), the ac-
tion failed because the robot was already holding the plate. To achieve the goal of placing a plate with
a spoon in it on the kitchen table, we need to first put down the plate, pick up the spoon, then put the
spoon on the plate, pick up the plate again, and finally place it on the dining table. language_plan: 1.
Find the dining table to put down the plate temporarily. 2. Put down the plate. 3. Find the spoon again.
4. Pick up the spoon. 5. Find the plate. 6. Put the spoon on the plate. 7. Pick up the plate (now with
spoon in it). 8. Find the dining table. 9. Put down the plate with spoon on the table.<|think_end|>
```
```
<|action_start|>[18, 'find a DiningTable']<|action_end|>
```

**Step 6:**

**Input image**

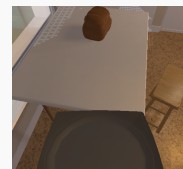

**Prompt**

**instruction:** Set plate with a spoon in it on the kitchen table
**interaction_history:**
```
[{``step_id'': 4, ``thinking'':
``visual_description: The image shows a kitchen
countertop surface. There appears to be some objects
on the counter, including what looks like an apple
or round fruit. reasoning_and_reflection: From the
interaction history, the robot successfully found
and picked up a plate (Steps 0-1), then found a spoon
(Step 2). However, when trying to pick up the spoon
(Step 3), the action failed because the robot was
already holding the plate. To achieve the goal of
placing a plate with a spoon in it on the kitchen
table, we need to first put down the plate, pick up
the spoon, then put the spoon on the plate, pick up
the plate again, and finally place it on the dining
table. language_plan: 1. Find the dining table to
put down the plate temporarily. 2. Put down the
plate. 3. Find the spoon again. 4. Pick up the
spoon. 5. Find the plate. 6. Put the spoon on the
plate. 7. Pick up the plate (now with spoon in it).
8. Find the dining table. 9. Put down the plate
with spoon on the table.'', ``action'': ``[18, 'find
a DiningTable']'', ``env_feedback'': ``Last action
executed successfully.''}]]
```
Based on the above information, please provide the action for the next
step to complete the task. Think, then act.

**Generation**

<|think_start|>visual_description: The robot is near a table with a loaf of bread on it. The robot
is holding a plate. reasoning_and_reflection: The robot successfully found the dining table and is ready
to put down the plate. The next step is to put down the plate, find the spoon, and then place the
spoon on the plate. Finally, the robot will pick up the plate with the spoon and place it on the table.
language_plan: 1. Put down the plate on the table. 2. Find the spoon. 3. Pick up the spoon. 4. Find the
plate. 5. Put the spoon on the plate. 6. Pick up the plate with the spoon. 7. Find the dining table. 8.
Put down the plate with the spoon on the table.<|think_end|>
<|action_start|>[133, 'put down the object in hand']<|action_end|>

**Step 7:**

**Input image**      **Prompt**

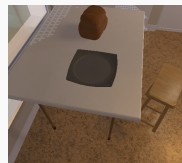

**instruction:** Set plate with a spoon in it on the kitchen table
**interaction_history:**

```
[{``step_id'': 5, ``thinking'':
``visual_description:  The robot is near a table with
a loaf of bread on it.  The robot is holding a plate.
reasoning_and_reflection:  The robot successfully
found the dining table and is ready to put down the
plate.  The next step is to put down the plate, find
the spoon, and then place the spoon on the plate.
Finally, the robot will pick up the plate with the
spoon and place it on the table.  language_plan:
1.  Put down the plate on the table.  2.  Find
the spoon.  3.  Pick up the spoon.  4.  Find the
plate.  5.  Put the spoon on the plate.  6.  Pick
up the plate with the spoon.  7.  Find the dining
table.  8.  Put down the plate with the spoon on the
table.'', ``action'':  ``[133, 'put down the object
in hand']'', ``env_feedback'':  ``Last action executed
successfully.''}]]
```
Based on the above information, please provide the action for the next
step to complete the task. Think, then act.

**Generation**

<|think_start|>visual_description: The plate is on the kitchen table next to a loaf of bread. There is a stool nearby. reasoning_and_reflection: The plate has been successfully placed on the kitchen table. The next step is to find and pick up a spoon to place it on the plate. The previous plan was interrupted after putting down the plate, so the next action is to locate the spoon and continue with the plan to place it on the plate. language_plan: 1. Find the spoon. 2. Pick up the spoon. 3. Find the plate on the kitchen table. 4. Put the spoon on the plate. 5. Pick up the plate with the spoon in it. 6. Put down the plate with the spoon on the kitchen table.<|think_end|>
<|action_start|>[19, 'find a Spoon']<|action_end|>

**Step 8:**

**Input image**      **Prompt**

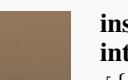

**instruction:** Set plate with a spoon in it on the kitchen table
**interaction_history:**

```
[{``step_id'': 6, ``thinking'':
``visual_description:  The plate is on the kitchen
table next to a loaf of bread.  There is a stool
nearby.  reasoning_and_reflection:  The plate has
been successfully placed on the kitchen table.  The
next step is to find and pick up a spoon to place
it on the plate.  The previous plan was interrupted
after putting down the plate, so the next action is
to locate the spoon and continue with the plan to
place it on the plate.  language_plan:  1.  Find
the spoon.  2.  Pick up the spoon.  3.  Find the
plate on the kitchen table.  4.  Put the spoon on
the plate.  5.  Pick up the plate with the spoon
in it.  6.  Put down the plate with the spoon on
the kitchen table.'', ``action'':  ``[19, 'find a
Spoon']'', ``env_feedback'':  ``Last action executed
successfully.''}]]
```
Based on the above information, please provide the action for the next
step to complete the task. Think, then act.

**Generation**

`<|think_start|>`visual_description: The kitchen counter has a spoon, a fork, a green apple, and a brown object. The sink is visible, and there are some utensils near it. reasoning_and_reflection: The previous plan involved putting down the plate to pick up the spoon. The first three actions were executed successfully, and the robot is now ready to pick up the spoon. The next steps involve placing the spoon on the plate, picking up the plate with the spoon, and placing it on the dining table. language_plan: 1. Pick up the spoon. 2. Find the plate. 3. Put the spoon on the plate. 4. Pick up the plate (now with spoon in it). 5. Find the dining table. 6. Put down the plate with spoon on the table.`<|think_end|>`

`<|action_start|>`[105, 'pick up the Spoon']`<|action_end|>`

**Step 9:**

**Input image**

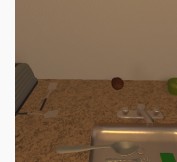

**Prompt**

**instruction:** Set plate with a spoon in it on the kitchen table
**interaction_history:**
```
[{``step_id'': 7, ``thinking'':
``visual_description: The kitchen counter has a
spoon, a fork, a green apple, and a brown object.
The sink is visible, and there are some utensils near
it. reasoning_and_reflection: The previous plan
involved putting down the plate to pick up the spoon.
The first three actions were executed successfully,
and the robot is now ready to pick up the spoon. The
next steps involve placing the spoon on the plate,
picking up the plate with the spoon, and placing it
on the dining table. language_plan: 1. Pick up the
spoon. 2. Find the plate. 3. Put the spoon on the
plate. 4. Pick up the plate (now with spoon in it).
5. Find the dining table. 6. Put down the plate
with spoon on the table.'', ``action'': ``[105,
'pick up the Spoon']'', ``env_feedback'': ``Last
action executed successfully.''}]]
```
Based on the above information, please provide the action for the next step to complete the task. Think, then act.

**Generation**

`<|think_start|>`visual_description: The image shows a kitchen counter with a spoon, fork, and a green object on a tray. There is also a brown object and a green apple on the counter. reasoning_and_reflection: The previous plan involved finding the plate to place the spoon on it. The spoon is already visible on a tray, which might be the plate. The next step is to confirm the presence of the plate and place the spoon on it. The plan needs to be adjusted to ensure the spoon is placed on the plate and then the plate is moved to the dining table. language_plan: 1. Find the plate (tray) on the counter. 2. Put the spoon on the plate. 3. Pick up the plate with the spoon. 4. Find the dining table. 5. Put down the plate with the spoon on the dining table.`<|think_end|>`

`<|action_start|>`[31, 'find a Plate']`<|action_end|>`

**Step 10:**

**Input image** **Prompt**

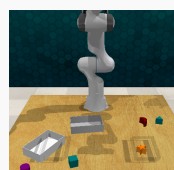

**instruction:** Set plate with a spoon in it on the kitchen table
**interaction_history:**
```
[{``step_id'': 8, ``thinking'':
``visual_description:  The image shows a kitchen
counter with a spoon, fork, and a green object on
a tray.  There is also a brown object and a green
apple on the counter.  reasoning_and_reflection:  The
previous plan involved finding the plate to place
the spoon on it.  The spoon is already visible on a
tray, which might be the plate.  The next step is to
confirm the presence of the plate and place the spoon
on it.  The plan needs to be adjusted to ensure the
spoon is placed on the plate and then the plate is
moved to the dining table.  language_plan:  1.  Find
the plate (tray) on the counter.  2.  Put the spoon
on the plate.  3.  Pick up the plate with the spoon.
4.  Find the dining table.  5.  Put down the plate
with the spoon on the dining table.'', ``action'':
``[31, 'find a Plate']'', ``env_feedback'':  ``Last
action executed successfully.''}]]
```
Based on the above information, please provide the action for the next step to complete the task. Think, then act.

**Generation**
`<|think_start|>`visual_description: The kitchen table has a loaf of bread, a plate, and a spoon on it. The spoon is not on the plate. reasoning_and_reflection: The robot has successfully found the dining table and put down the plate. It then found and picked up the spoon. The next step is to place the spoon on the plate, pick up the plate with the spoon, and place it on the table. The previous plan was interrupted after picking up the spoon, so the next step is to put the spoon on the plate. language_plan: 1. Find the plate. 2. Put the spoon on the plate. 3. Pick up the plate with the spoon. 4. Find the kitchen table. 5. Put down the plate with the spoon on the table.`<|think_end|>`
`<|action_start|>`[133, 'put down the object in hand']`<|action_end|>`

---

**Trajectory-augmented Prior for EB-Manipulation**

**Step 1:**

**Input image** **Prompt**

**instruction:** Pick up the star and place it into the silver container.
**interaction_history:** []
**additional_info:** ``{``object 1'': [73, 15, 18],
``object 2'': [57, 20, 18], ``object 3'': [66,
38, 18], ``object 4'': [33, 43, 17], ``object
5'': [57, 74, 17], ``object 6'': [28, 80, 17],
``object 7'': [26, 91, 17]}``
Based on the above information, please provide the action for the next step to complete the task. Think, then act.

**Generation**
`<|think_start|>`visual_description: From left to right, I can see a purple cylinder at [73, 15, 18], a white container at [57, 20, 18], a teal cube at [66, 38, 18], a silver container at [33, 43, 17], an orange star at [57, 74, 17], a maroon moon at [28, 80, 17], and a teal triangular at [26, 91, 17]. reasoning_and_reflection: The task requires picking up the orange star located at [57, 74, 17] and placing it into the silver container at [33, 43, 17]. I will need to move the gripper to the star position, grasp it, lift it up, move it to the silver container position, and then release it. language_plan: 1. Move gripper above the orange star\n2. Lower gripper and open it to prepare for grasping\n3. Close gripper

to grasp the star\n4. Lift the star up\n5. Move the star above the silver container\n6. Lower the star into the container\n7. Open gripper to release the star<|think_end|>
<|action_start|>[57, 74, 27, 0, 60, 90, 1]<|action_end|>

**Step 2:**

**Input image**          **Prompt**

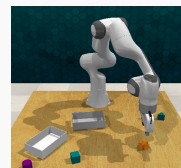

**instruction:** Pick up the star and place it into the silver container.
**interaction_history:**
```
[{``step_id'': 0, ``thinking'':
``visual_description: From left to right, I can see a
purple cylinder at [73, 15, 18], a white container at
[57, 20, 18], a teal cube at [66, 38, 18], a silver
container at [33, 43, 17], an orange star at [57,
74, 17], a maroon moon at [28, 80, 17], and a teal
triangular at [26, 91, 17]. reasoning_and_reflection:
The task requires picking up the orange star located
at [57, 74, 17] and placing it into the silver
container at [33, 43, 17]. I will need to move
the gripper to the star position, grasp it, lift it
up, move it to the silver container position, and
then release it. language_plan: 1. Move gripper
above the orange star\n2. Lower gripper and open
it to prepare for grasping\n3. Close gripper to
grasp the star\n4. Lift the star up\n5. Move the
star above the silver container\n6. Lower the star
into the container\n7. Open gripper to release the
star'', ``action'': ``[57, 74, 27, 0, 60, 90, 1]'',
``env_feedback'': ``Last action was successful.''}]]
```
**additional_info:**          {``object 1'': [73, 15, 18],
``object 2'': [55, 21, 17], ``object 3'': [66,
38, 18], ``object 4'': [34, 46, 18], ``object
5'': [56, 74, 17], ``object 6'': [26, 79, 17],
``object 7'': [26, 91, 17]}
Based on the above information, please provide the action for the next step to complete the task. Think, then act.

**Generation**
<|think_start|>visual_description: From left to right, I can see a purple cylinder at [73, 15, 18], a white container at [55, 21, 17], a teal cube at [66, 38, 18], a silver container at [34, 46, 18], an orange star at [56, 74, 17], a maroon moon at [26, 79, 17], and a teal triangular at [26, 91, 17]. reasoning_and_reflection: The task involves picking up the orange star and placing it into the silver container. I have already moved the gripper above the star. The next step is to lower the gripper and open it to prepare for grasping the star. After that, I will close the gripper to grasp the star, lift it, move it to the container, and release it. I need to ensure that each step is executed precisely to avoid dropping the star or missing the container. language_plan: 1. Lower gripper and open it to prepare for grasping\n2. Close gripper to grasp the star\n3. Lift the star up\n4. Move the star above the silver container\n5. Lower the star into the container\n6. Open gripper to release the star<|think_end|>
<|action_start|>[57, 74, 17, 0, 60, 90, 0]<|action_end|>

**Step 3:**

2538
2539
2540
2541
2542
2543
2544
2545
2546
2547
2548
2549
2550
2551
2552
2553
2554
2555
2556
2557
2558
2559
2560
2561
2562
2563
2564
2565
2566
2567
2568
2569
2570
2571
2572
2573
2574
2575
2576
2577
2578
2579
2580
2581
2582
2583
2584
2585
2586
2587
2588
2589
2590
2591

**Input image**

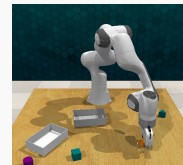

**Prompt**

**instruction:** Pick up the star and place it into the silver container.
**interaction_history:**
[{``step_id'': 1, ``thinking'':
``visual_description: From left to right, I can see a
purple cylinder at [73, 15, 18], a white container at
[55, 21, 17], a teal cube at [66, 38, 18], a silver
container at [34, 46, 18], an orange star at [56,
74, 17], a maroon moon at [26, 79, 17], and a teal
triangular at [26, 91, 17]. reasoning_and_reflection:
The task involves picking up the orange star and
placing it into the silver container. I have already
moved the gripper above the star. The next step
is to lower the gripper and open it to prepare for
grasping the star. After that, I will close the
gripper to grasp the star, lift it, move it to the
container, and release it. I need to ensure that
each step is executed precisely to avoid dropping the
star or missing the container. language_plan: 1.
Lower gripper and open it to prepare for grasping\n2.
Close gripper to grasp the star\n3. Lift the star
up\n4. Move the star above the silver container\n5.
Lower the star into the container\n6. Open gripper
to release the star'', ``action'': ``[57, 74, 17,
0, 60, 90, 0]'', ``env_feedback'': ``Last action was
successful.''}]]
**additional_info:** {``object 1'': [73, 15, 18],
``object 2'': [55, 21, 17], ``object 3'': [66,
38, 18], ``object 4'': [32, 40, 17], ``object
5'': [56, 73, 16], ``object 6'': [27, 81, 18],
``object 7'': [26, 91, 17]}
Based on the above information, please provide the action for the next
step to complete the task. Think, then act.

**Generation**
<|think_start|>visual_description: From left to right, I can see a purple cylinder at [73, 15,
18], a white container at [55, 21, 17], a teal cube at [66, 38, 18], a silver container at [32, 40, 17],
an orange star at [56, 73, 16], a maroon moon at [27, 81, 18], and a teal triangular at [26, 91, 17].
reasoning_and_reflection: The task involves transferring the star from its current position to the silver
container. I have already moved the gripper above the star and lowered it to grasp the star. The next
step is to close the gripper to secure the star, lift it, move it to the container, and release it. I need to
ensure that the gripper is properly aligned and that the star is securely held before moving it to avoid
dropping it. language_plan: 1. Close gripper to grasp the star\n2. Lift the star up\n3. Move the
star above the silver container\n4. Lower the star into the container\n5. Open gripper to release the
star<|think_end|>
<|action_start|>[57, 74, 25, 0, 60, 90, 0]<|action_end|>

**Step 4:**

**Input image**

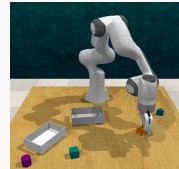

**Prompt**

**instruction:** Pick up the star and place it into the silver container.
**interaction_history:**
```
[{''step_id'': 2, ''thinking'':
''visual_description: From left to right, I can see a
purple cylinder at [73, 15, 18], a white container at
[55, 21, 17], a teal cube at [66, 38, 18], a silver
container at [32, 40, 17], an orange star at [56,
73, 16], a maroon moon at [27, 81, 18], and a teal
triangular at [26, 91, 17]. reasoning_and_reflection:
The task involves transferring the star from its
current position to the silver container. I have
already moved the gripper above the star and lowered
it to grasp the star. The next step is to close
the gripper to secure the star, lift it, move it
to the container, and release it. I need to ensure
that the gripper is properly aligned and that the
star is securely held before moving it to avoid
dropping it. language_plan: 1. Close gripper to
grasp the star\n2. Lift the star up\n3. Move the
star above the silver container\n4. Lower the star
into the container\n5. Open gripper to release the
star'', ''action'': ''[57, 74, 25, 0, 60, 90, 0]'',
''env_feedback'': ''Last action was successful.''}]]
```
**additional_info:** {''object 1'': [73, 15, 18],
''object 2'': [55, 21, 17], ''object 3'': [66,
38, 18], ''object 4'': [31, 40, 18], ''object
5'': [58, 72, 24], ''object 6'': [27, 81, 18],
''object 7'': [26, 91, 17]}
Based on the above information, please provide the action for the next step to complete the task. Think, then act.

**Generation**
<|think_start|>visual_description: From left to right, I can see a purple cylinder at [73, 15, 18], a white container at [55, 21, 17], a teal cube at [66, 38, 18], a silver container at [31, 40, 18], an orange star at [58, 72, 24], a maroon moon at [27, 81, 18], and a teal triangular at [26, 91, 17]. reasoning_and_reflection: The task is to place the star into the silver container. I have already moved the gripper to the star's position, grasped it, and lifted it up. The next step is to move the star to the position above the silver container, lower it into the container, and release it. I need to ensure the star is securely placed inside the container before releasing it. language_plan: 1. Move the star above the silver container\n2. Lower the star into the container\n3. Open gripper to release the star<|think_end|>
<|action_start|>[33, 43, 25, 0, 60, 90, 0]<|action_end|>

**Step 5:**

**Input image**

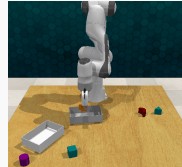

**Prompt**

**instruction:** Pick up the star and place it into the silver container.
**interaction_history:**
```
[{``step_id'': 3, ``thinking'':
``visual_description:  From left to right, I can
see a purple cylinder at [73, 15, 18], a white
container at [55, 21, 17], a teal cube at [66,
38, 18], a silver container at [31, 40, 18], an
orange star at [58, 72, 24], a maroon moon at [27,
81, 18], and a teal triangular at [26, 91, 17].
reasoning_and_reflection:  The task is to place the
star into the silver container.  I have already moved
the gripper to the star's position, grasped it, and
lifted it up.  The next step is to move the star to
the position above the silver container, lower it
into the container, and release it.  I need to ensure
the star is securely placed inside the container
before releasing it.  language_plan:  1.  Move the
star above the silver container\n2.  Lower the star
into the container\n3.  Open gripper to release the
star'', ``action'':  ``[33, 43, 25, 0, 60, 90, 0]'',
``env_feedback'':  ``Last action was successful.''}]]
```
**additional_info:**     `{``object 1'': [73, 15, 18], ``object 2'': [55, 21, 17], ``object 3'': [66, 38, 18], ``object 4'': [34, 41, 25], ``object 5'': [32, 42, 17], ``object 6'': [27, 80, 18], ``object 7'': [26, 91, 17]}`
Based on the above information, please provide the action for the next step to complete the task. Think, then act.

**Generation**

`<|think_start|>`visual_description: From left to right, I can see a purple cylinder at [73, 15, 18], a white container at [55, 21, 17], a teal cube at [66, 38, 18], an orange star at [34, 41, 25], a silver container at [32, 42, 17], a maroon moon at [27, 80, 18], and a teal triangular at [26, 91, 17]. reasoning_and_reflection: The task is to place the star into the silver container. I have already picked up the star and moved it above the container. The next steps involve lowering the star into the container and releasing it. I need to ensure the gripper is positioned correctly above the container before lowering it. Once the star is inside the container, I will open the gripper to release it and then move the gripper back to a neutral position. language_plan: 1. Move the star above the silver container\n2. Lower the star into the container\n3. Open gripper to release the star\n4. Move gripper back to a neutral position`<|think_end|>`
`<|action_start|>`[33, 43, 17, 0, 60, 90, 1]`<|action_end|>`

## H.4 ENVIRONMENT-ANCHORED PRIOR DATASET

We provide an example of Environment-Anchored Prior dataset for both EB-ALFRED and EB-Manipulation.

2700
2701
2702
2703
2704
2705
2706
2707
2708
2709
2710
2711
2712
2713
2714
2715
2716
2717
2718
2719
2720
2721
2722
2723
2724
2725
2726
2727
2728
2729
2730
2731
2732
2733
2734
2735
2736
2737
2738
2739
2740
2741
2742
2743
2744
2745
2746
2747
2748
2749
2750
2751
2752
2753

---

### Environment-Anchored Prior Dataset for EB-ALFRED (Masked Action Modeling)

**Prompt**

You are a household assistant. You are given an instruction: "Place the clean bar soap from the bin into the tub." and an incomplete action sequence: ["find a garbagecan", "pick up the soapbar", "find a sinkbasin", "[MASK]", "find a Faucet", "turn on the Faucet", "turn off the Faucet", "find a soapbar", "pick up the soapbar", "find a bathtubbasin", "put down the object in hand"]. Please identify the missing action to complete the sequence.

**Generation**

To complete the logical chain for the instruction "Place the clean bar soap from the bin into the tub.", we need to ensure that the sequence of actions aligns with the goal of cleaning the soap and then placing it in the tub. Let's break down the sequence step by step:

1. **Find a garbagecan**: This step is likely to ensure that any debris or used soap can be discarded if necessary.
2. **Pick up the soapbar**: This is the initial step to get the soap ready for cleaning.
3. **Find a sinkbasin**: The sinkbasin is where the soap will be cleaned.
4. **[MASK]**: This is the missing action that needs to be identified.
5. **Find a Faucet**: The faucet is needed to provide water for cleaning the soap.
6. **Turn on the Faucet**: This action provides the water necessary for cleaning.
7. **Turn off the Faucet**: This action stops the water flow after the soap is cleaned.
8. **Find a soapbar**: This step is redundant because the soapbar was already picked up earlier. However, it might be included to ensure the soap is still in hand.
9. **Pick up the soapbar**: This step is also redundant but ensures the soap is ready to be moved.
10. **Find a bathtubbasin**: The bathtubbasin is the final destination for the clean soap.
11. **Put down the object in hand**: This action places the clean soap in the bathtubbasin.

Given the context, the missing action should logically be an action that prepares the soap for cleaning. The most appropriate action to fill the gap is "put down the object in hand". This ensures that the soap is placed in the sinkbasin before the water is turned on, allowing it to be properly cleaned. After the soap is cleaned and the faucet is turned off, the sequence continues with finding the bathtubbasin and placing the clean soap in it. Therefore, the missing action is "put down the object in hand". To complete the logical chain for the instruction "Place the clean bar soap from the bin into the tub.", the missing action is "put down the object in hand". The complete and correct sequence is: ["find a garbagecan", "pick up the soapbar", "find a sinkbasin", "put down the object in hand", "find a Faucet", "turn on the Faucet", "turn off the Faucet", "find a soapbar", "pick up the soapbar", "find a bathtubbasin", "put down the object in hand"].

---

**Environment-Anchored Prior Dataset for EB-ALFRED (Action Sequence Reordering)**

**Prompt**

You are a household assistant. You are given the instruction: "Relocate the remote from the chair to the lamp." The randomized action sequences are ["pick up the remotecontrol", "find a floorlamp", "find an armchair", "turn on the floorlamp"]. Your task is to generate the correct sequence of actions to accomplish the instruction.

**Generation**

To accomplish the task of relocating the remote from the chair to the lamp, I need to follow a logical sequence of actions. First, I must locate the armchair where the remote is currently placed. Once I have found the armchair, I can proceed to pick up the remote control from it. Next, I need to find the floorlamp where I will place the remote. Finally, to ensure the lamp is functional and the remote is easily accessible, I will turn on the floorlamp. This sequence ensures that I complete the task as instructed, moving the remote from the chair to the lamp and verifying that the lamp is operational.

As a household assistant, to accomplish the instruction "Relocate the remote from the chair to the lamp.", the correct sequence of actions is: ["find an armchair", "pick up the remotecontrol", "find a floorlamp", "turn on the floorlamp"].

---

**Environment-Anchored Prior Dataset for EB-Manipulation (Absolute Coordinate Grounding)**

**Input image:**

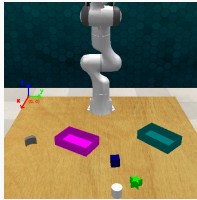

**Example 1:**
**Prompt**
What is the 3D coordinate of the gray moon?

**Generation**
[42, 11, 17]

**Example 2:**
**Prompt**
What object is located at [42, 11, 17]?

**Generation**
The gray moon

---

---

**Environment-Anchored Prior Dataset for EB-Manipulation (Relative Coordinate Grounding)**

**Input image:**

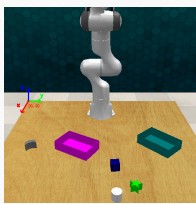

**Example 1:**
**Prompt**
What is the 3D location of the leftmost object?

**Generation**
[42, 11, 17]

**Example 2:**
**Prompt**
What is the 3D location of the rightmost object?"

**Generation**
[41, 80, 18]

---

**Environment-Anchored Prior Dataset for EB-Manipulation (Combination of Absolute & Relative Grounding)**

**Input image:**

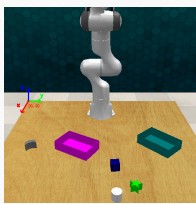

**Example 1:**
**Prompt**
Is the object located at [42, 11, 17] the leftmost in the scene?

**Generation**
Yes

**Example 2:**
**Prompt**
Is the object located at [42, 11, 17] the rightmost in the scene?

**Generation**
No

---

### H.5 EXTERNAL KNOWLEDGE PRIOR DATASET

We provide an example of External Knowledge Prior dataset for both EB-ALFRED and EB-Manipulation.

**External Knowledge Prior Dataset for EB-ALFRED (OpenO1-SFT)**

**Prompt**

## You are a robot who is great at reasoning. Given a instruction, you must generate a thought process and then an output.
Question: To find the number of positive five-digit numbers that have a 1, 3, 7, or 5 in them.

**Generation**

Alright, I need to find the number of positive five-digit numbers that have a 1, 3, 5, or 7 in them. First, let's clarify the problem to ensure I understand it correctly. It seems that the task is to count all five-digit numbers where at least one of the digits is either 1, 3, 5, or 7.
To approach this, I can consider the total number of five-digit numbers and then subtract those that do not contain any of the digits 1, 3, 5, or 7.

**Step 1: Calculate the Total Number of Five-Digit Numbers**

Five-digit numbers range from 10000 to 99999. To find the total number of five-digit numbers:

- The first digit (ten-thousands place) can be from 1 to 9 (since it cannot be 0), giving 9 options.
- The remaining four digits (thousands, hundreds, tens, and ones places) can each be from 0 to 9, giving 10 options each.

Total five-digit numbers = $9 \times 10 \times 10 \times 10 \times 10 = 90,000$

**Step 2: Calculate the Number of Five-Digit Numbers Without 1, 3, 5, or 7**

If a number does not contain any of the digits 1, 3, 5, or 7, the allowed digits are 0, 2, 4, 6, 8, and 9.

- The first digit cannot be 0 (to maintain a five-digit number), so it can be 2, 4, 6, 8, or 9. That's 5 options.
- The remaining four digits can be 0, 2, 4, 6, 8, or 9, giving 6 options each.

Number of five-digit numbers without 1, 3, 5, or 7 = $5 \times 6 \times 6 \times 6 \times 6 = 5 \times 6^4$

Calculating $6^4$:
$6^2 = 36$
$6^3 = 216$
$6^4 = 1296$

So, number = $5 \times 1296 = 6480$

**Step 3: Subtract to Find the Desired Number**
Number of five-digit numbers with at least one digit being 1, 3, 5, or 7 = Total five-digit numbers - Numbers without 1, 3, 5, or 7

$= 90,000 - 6,480 = 83,520$
But wait, let's double-check to ensure there are no mistakes in the calculation.

**Verification:**
Total five-digit numbers: 90,000
Numbers without 1, 3, 5, or 7: 5 (first digit) $\times 6^4$ (remaining digits) $= 5 \times 1296 = 6480$
Subtracting: $90,000 - 6,480 = 83,520$
Everything seems correct. There are no overlapping cases or double-counting issues since the approach uses complementary counting.

**External Knowledge Prior Dataset for EB-Manipulation (Spacethinker)**

**Input image**

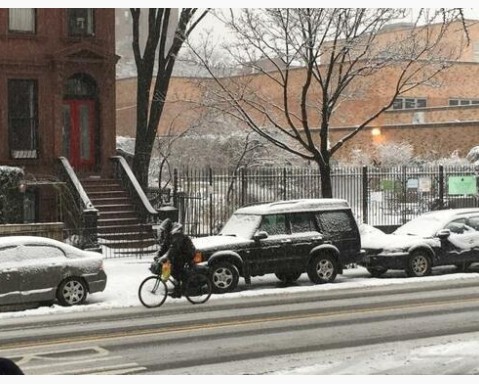

**Prompt**
## You are a robot who is great at spatial reasoning. Given a instruction, you must generate a thought process and then an output.
Question: How far apart are the snow-covered parked cars and the snowy street with parked cars?

**Generation**
I see a street covered with snow, and several parked cars also dusted with snow. Additionally, there is a cyclist riding along the road. The snow covers both the cars and the roadway, suggesting a contiguous snowy scene. Now, determining the distance between "snow-covered parked cars" and the snowy street with parked cars"suggests considering the separation between two distinct areas within this same scene.both the cars and the street are snow-covered, it might seem as if they are tightly connected in context; however, finding a distance here usually involves looking at the lateral space due to the lack of any actual intervening physical barrier or wide expanse in the image. I estimate this minimal distance to be the space separating the parked cars from the street edge directly in front of them, without a significant gap in the visible scene.these observations, the notion of any small distance like 20.33 inches feels particularly plausible, most likely representing the buffer or visual gap where the snow transitions from the cars to the street with virtually no space. This would particularly make sense if the focus were on something like the offset from the curb which is often minor., although exactness in this scenario might be less definitive visually, conceptually a small distance effectively represents the snow transition directly from car to the street, accounting for negligible separation.
There is essentially no significant distance between the snow-covered parked cars and the snowy street with parked cars, with the understanding that 20.33 inches could be seen as the minimal or visual margin separating the snow on the cars from the street-level snow.

## H.6 PROMPT FOR GENERATING REASONING AUGMENTATION

Prompt

For the following task: {}

You have generated the following multi-step plan to complete the task:
{}

You have executed the first {} actions of the plan. The next action to be executed is {}. Now you need to follow the multi-step plan to generate the next multi-step plan including `"visual_state_description"`, `"reasoning_and_reflection"`, and `"language_plan"` in the format of a JSON object. Make sure the step number in the language plan starts from 1.

## H.7 Rule-based ground truth visual description generation for EB-Manipulation

Based on the failure analysis in EmbodiedBench Yang et al. (2025), we observe that visual perception errors are the most frequent source of failure for VLMs in EB-Manipulation. To mitigate this issue, we adopt a rule-based approach to generate ground-truth visual descriptions, which are incorporated as additional inputs when collecting new successful trajectories. Specifically, we extract `object_name--color--coordinate` tuples from the observation data and generate the final oracle visual description using a rule-based template.

## H.8 Rule-based action mapping for EB-ALFRED

First, we examined the raw actions in the original ALFRED dataset along with the new actions introduced in EB-ALFRED, and manually defined a set of mapping rules. Each raw action was then mapped, via these rules, to one or more corresponding actions in EB-ALFRED. After transforming the original action sequence into the new EB-ALFRED action space, we executed the mapped sequence in the simulator to verify that all mappings were valid.

# I ERROR ANALYSIS

To understand how the EPL and RL stages in ERA reduce different types of errors, we conduct an error analysis on unseen subsets of both the high-level planning task EB-ALFRED (100 tasks) and the low-level control task EB-Manipulation (98 tasks). We categorize into 3 types of error: *(i) Perception errors*: incorrect descriptions of the current state; *(ii) Reasoning errors*: mistakes in reasoning about the current state or reflecting on history; *(iii) Planning errors*: mistakes in planning future steps. The results, shown in Figure 5, reveal distinct patterns across task levels. In high-level tasks, reasoning and planning errors are dominant, while in low-level tasks, perception and reasoning errors are more prevalent. Across both settings, EPL and RL consistently reduce errors, but their effects differ in granularity: **EPL contributes to reducing all error types, whereas RL is especially effective at lowering reasoning and planning errors.** When combined, EPL and RL (i.e., ERA) achieve the lowest error rates across all categories. We provide a deeper case analysis for ERA in Appendix I.

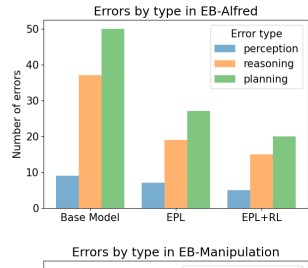

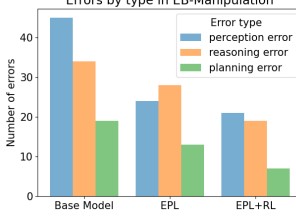

Figure 5: Comparing error statistics in two benchmarks.

Our embodied agent demonstrates qualitatively different types of failure across **eb-alfred** (high-level tasks) and **eb-manipulation** (low-level tasks). While the agent shows encouraging levels of competence, a closer look reveals systematic patterns where capabilities are emerging but not yet fully reliable.

## I.1 FAILURE MODES IN EB-ALFRED.

In high-level planning and reasoning tasks, two broad categories of failure can be identified: *strategic rigidity* and *action-level inconsistency*.

**Strategic rigidity.** (1) **Limited reflection:** one recurring observation is that the agent does not consistently adjust its behavior in response to environmental signals. For example, when an attempted action fails (e.g., trying to open a cabinet that is already open), the agent sometimes repeats the same command multiple times until the maximum step limit is reached. This suggests that reflection mechanisms—such as reconsidering recent outcomes or modifying plans dynamically—have not been deeply internalized. A plausible explanation is tied to the training curriculum: in Stage 1 (SFT), the agent was mostly exposed to successful demonstrations where explicit reflection was unnecessary, and in Stage 2 (RL), the incentives may not have been strong enough to encourage learning reflection strategies. This points to reflection as a promising area for future augmentation, potentially through targeted data augmentation or specialized reward shaping.

(2) **Conservative exploration:** when tasked with locating an item, the agent tends to pursue a fixed search plan. If the initial attempt fails (e.g., looking for a mug in the first cupboard), it often does not adaptively explore other plausible locations, but instead persists with its initial trajectory. This "plan-first-and-stick-to-it" tendency is advantageous when the initial guess is correct, but can lead to stagnation otherwise. Such rigidity highlights that while the agent has acquired a notion of planning, it still lacks mechanisms for broadening the search space when initial strategies fail. Improving adaptive exploration remains a key direction, for example by introducing curiosity-driven objectives or uncertainty-based exploration bonuses.

**Action-level inconsistency.** (3) **Challenges in action sequencing:** even when the agent's global plan is appropriate, the precise ordering of low-level actions is sometimes inconsistent. A typical example occurs when holding one object while needing to manipulate another: the agent may attempt to pick up the new object without first releasing the one already in hand. These mis-sequencing errors suggest that high-level intentions are successfully maintained but the grounding of those intentions into motor-level action chains is less robust. This could be due to limited diversity in training data that emphasizes successful sequences, leaving the agent underexposed to edge cases requiring careful ordering.

(4) **Reasoning–action misalignment:** another pattern is a disconnect between verbalized reasoning and executed actions. For instance, the agent may correctly articulate the need to place a cup on a table, yet follow this with an incongruent command such as "find a wine bottle." These mismatches often recur across episodes, suggesting that the reasoning component generalizes better than the action-generation component, which may have overfit to spurious correlations in the training distribution. This partial decoupling indicates that while reasoning ability is promising, the mapping from reasoning to action needs more grounding and regularization to avoid drift.

I.2 FAILURE MODES IN EB-MANIPULATION.

In low-level manipulation tasks, the challenges are more sensorimotor in nature. Four representative patterns can be identified.

(1) **Underutilization of visual feedback:** the agent sometimes executes subsequent actions as though a prior action has succeeded, even when perceptual evidence indicates otherwise. For example, it may fail to grasp an object, yet proceed as if the object were in hand. This suggests that perception and action verification are not yet tightly coupled: the visual module detects the state of the environment, but the policy does not consistently integrate this feedback to update its internal state. Strengthening this integration could improve the reliability of sequential manipulation.

(2) **Limited error recovery:** once an execution error occurs, the agent has difficulty restoring the task flow. For instance, if the robotic arm collides with an obstacle and becomes stuck, the agent often continues issuing commands without attempting to disengage or reset the arm's configuration. Similarly, if an object slips from its grasp, the agent typically does not pause to re-attempt the grasp but proceeds as though the object were still held. These behaviors indicate that the agent has not yet learned systematic error-recovery strategies such as backtracking, re-initializing poses, or retrying actions with adjusted parameters. Incorporating explicit "recovery demonstrations" or adding intrinsic rewards for restoring feasible states could strengthen this capability.

(3) **Limited orientation and geometry awareness:** beyond recovering from errors, the agent also struggles with fine-grained spatial reasoning about object shapes and poses. For example, when placing a star-shaped block into a sorter, the agent frequently attempts insertion without adjusting the orientation, causing the block to catch on the edge. Unlike error recovery, which involves resuming after failure, this limitation reflects insufficient awareness of object geometry during action selection. The current policy seems to maintain a coarse representation of object location (e.g., "the star is near the slot") but not the detailed geometric constraints required for precise placement. Addressing this may require augmenting perception with pose estimation modules, or training with tasks that explicitly demand orientation-sensitive manipulation.

(4) **Instruction interpretation challenges:** in some cases, the agent misinterprets instructions, especially those involving novel linguistic constructs. For example, in the instruction "put the left object in the right container," the agent may incorrectly treat the container itself as an object to be moved, leading to placing one container inside another. A particularly striking case arises with "put the right object in the right container," where the agent may attempt to put a container into itself. While these errors can appear humorous, they underscore the limitations of current instruction-grounding. They also suggest that although the agent can parse familiar commands, its ability to generalize compositional semantics to unseen phrasings is still limited.

I.3 COMPARISON AND DISCUSSION.

Taken together, these observations highlight complementary challenges across levels of abstraction. In high-level tasks (**eb-alfred**), the main bottlenecks arise from rigidity in strategy adaptation and inconsistencies in aligning reasoning with action. These are primarily issues of decision-making flexibility and reasoning–execution coupling. In low-level tasks (**eb-manipulation**), the core difficulties lie in three different but interconnected domains: (i) integrating perceptual feedback to confirm action outcomes, (ii) developing robust recovery mechanisms when execution goes off track, and (iii) reasoning about object geometry and orientation for precise manipulation. In addition, both domains share the broader challenge of generalizing instruction understanding to unseen linguistic forms.

Overall, high-level planning emphasizes adaptability and coherence, while low-level manipulation emphasizes precision, recovery, and grounding. Addressing these limitations in tandem will be crucial for developing embodied agents capable of both flexible reasoning and robust physical interaction.

## J CASE ANALYSIS

We present two trajectory examples in Figure 6, which demonstrate that ERA exhibits sophisticated long-term reasoning, planning, and action execution across both tasks.

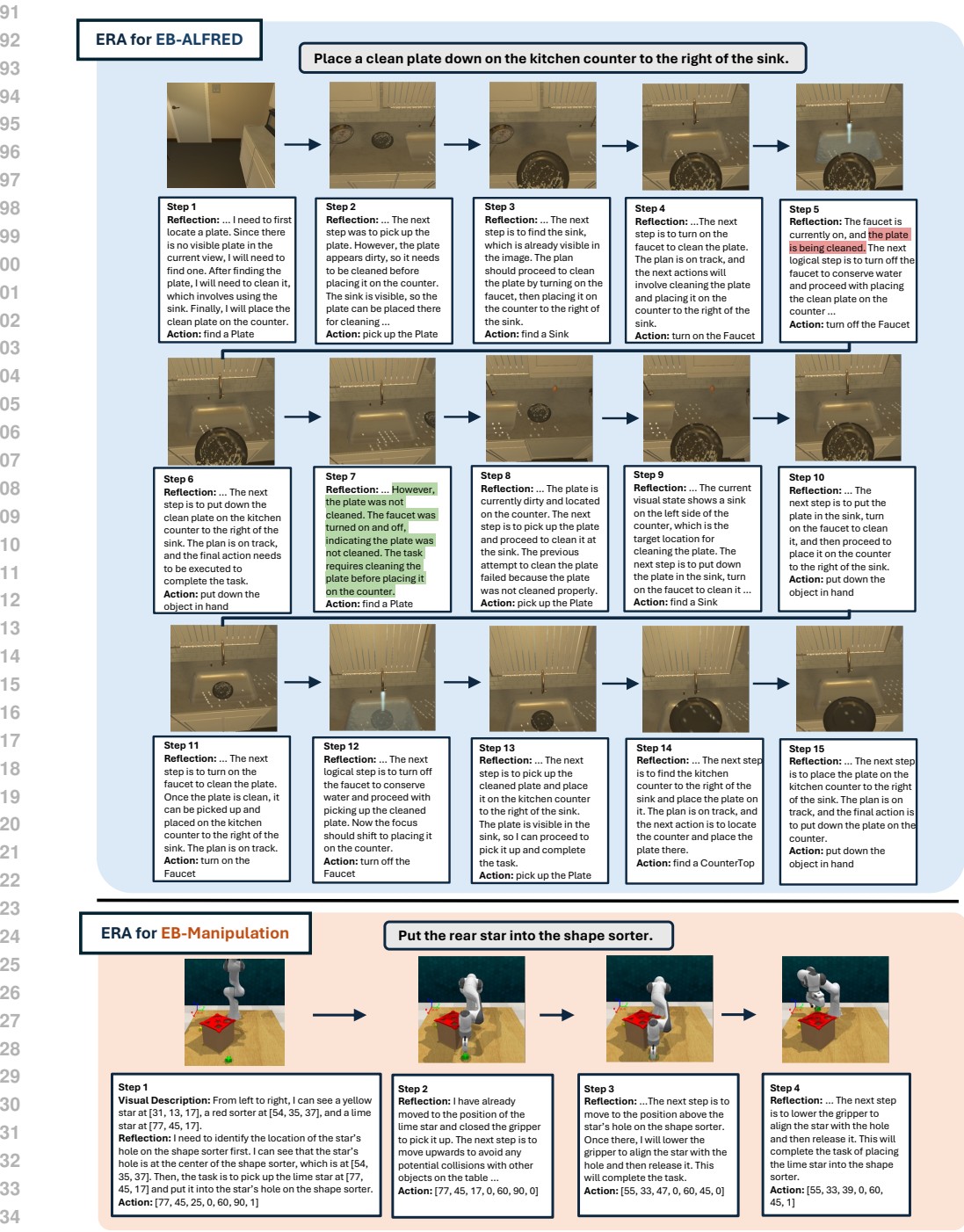

Figure 6: Examples of ERA performing step-by-step reasoning and actions: (a) on EB-ALFRED, it identifies and reflects on earlier errors; (b) on EB-Manipulation, it accurately places the star into the correct slot of the shape sorter.

**Human Instruction: Pick up the object known for bending light into a colorful display and place it into the navy blue container used for keeping things organized.**

**1. EPL only**

**Planning Step 1:**
**Visual description:** From left to right, I can see an orange container at [37, 25, 17], a navy container at [60, 42, 17], an azure triangular prism at [40, 43, 18], a yellow star at [35, 57, 17], an azure cylinder at [35, 75, 18], and a yellow cube at [74, 77, 18].

**Reflection:** The instruction asks to pick up the object that bends light into a colorful display, which refers to the yellow cube at [74, 77, 18], and place it into the navy blue container at [60, 42, 17]. The plan involves moving the gripper to the cube's position, closing the gripper to pick it up ...

**Language plan:**
1. Move to the yellow cube at [74, 77, 18]
2. Close the gripper to pick up the star ...

[74, 77, 26, 0, 60, 90, 1]

**Planning Step 2:**
**Visual description:** From left to right, I can see an orange container at [36, 23, 17], a navy container at [60, 42, 17], an azure triangular prism at [39, 42, 17], a yellow star at [35, 57, 17], an azure cylinder at [35, 75, 18], and a yellow cube at [74, 77, 18].

**Reflection:** The task is to pick up the yellow cube and place it into the navy blue container. The first action has been executed, which involved moving the gripper to the position above the yellow cube. The next step is to lower the gripper to the cube's position and close the gripper to pick it up ...

**Language plan:**
1. Lower the gripper to the yellow cube at [74, 77, 18] ...

[74, 77, 18, 0, 60, 90, 0]

**2. ERA (EPL + RL)**

**Planning Step 1:**
**Visual description:** From left to right, I can see an orange container at [37, 25, 17], a navy blue container at [60, 42, 17], an azure triangular prism at [40, 43, 18], a yellow star at [35, 57, 17], an azure cylinder at [35, 75, 18], and a yellow cube at [74, 77, 18].

**Reflection:** The instruction asks to pick up the object that bends light into a colorful display, which refers to the azure triangular prism at [40, 43, 18]. The destination is the navy blue container at [60, 42, 17]. The plan is to move the gripper above the triangular prism, open the gripper, lower to grasp it ...

**Language plan:**
1. Move gripper above the azure triangular prism at [40, 43, 26] with gripper open ...

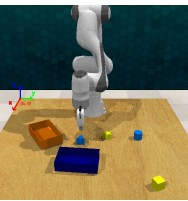

[40, 43, 26, 0, 60, 90, 1]

**Planning Step 2:**
**Visual description:** From left to right, I can see an orange container at [38, 24, 17], a navy blue container at [60, 42, 17], an azure triangular prism at [40, 42, 17], a yellow star at [35, 57, 17], an azure cylinder at [35, 74, 18], and a yellow cube at [74, 77, 18].

**Reflection:** The task is to pick up the azure triangular prism and place it into the navy blue container. The first action has been executed, which involved moving the gripper above the triangular prism with the gripper open. The next step is to lower the gripper to the triangular prism's position while keeping the gripper open, so it can be grasped ...

**Language plan:**
1. Lower gripper to the triangular prism at [40, 43, 18] with gripper open.

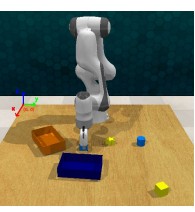

[40, 43, 18, 0, 60, 90, 1]

Figure 7: Reflection Error Example in EB-Manipulation. ERA successfully identified the correct target object: the azure triangular prism, while EPL mistakenly selected the yellow cube.

**Human Instruction: Pick up the twinkling object reminiscent of night-time skies and constellations and place it into the red storage bin.**

## 1. EPL only

**Planning Step 1:**
**Visual description:** From left to right, I can see a silver star at [35, 15, 17], a teal triangular at [57, 18, 18], a violet container at [49, 37, 17], a red container at [76, 69, 17], and an orange cylinder at [54, 81, 18].

**Reflection:** The instruction asks to pick up the twinkling object reminiscent of night-time skies and constellations, which refers to the silver star at [35, 15, 17], and place it into the red storage bin at [76, 69, 17]. The plan involves moving the gripper to the star's position, closing the gripper to pick it up ...

**Language plan:**
1. Move to the star's position at [35, 15, 17].
2. Close the gripper to pick up the star ...

[35, 15, 26, 0, 60, 90, 1]

**Planning Step 5:**
**Visual description:** From left to right, I can see a silver star at [37, 13, 27], a teal triangular at [57, 18, 17], a violet container at [50, 37, 17], a red container at [76, 68, 17], and an orange cylinder at [54, 81, 18].

**Reflection:** ... The previous actions have successfully moved the gripper to the position above the red container with the star securely held. The next step is to lower the gripper to the container's height and open the gripper to release the star into the container. This will complete the task as the star will be placed inside the red container.

**Language plan:**
1. Lower the gripper to the height of the red container at [76, 69, 17] ...

[76, 69, 17, 0, 60, 90, 1]

## 2. ERA (EPL + RL)

**Planning Step 1:**
**Visual description:** From left to right, I can see a silver star at [35, 15, 17], a teal triangular at [57, 18, 18], a violet container at [49, 37, 17], a red container at [76, 69, 17], and an orange cylinder at [54, 81, 18].

**Reflection:** The instruction asks to pick up the object reminiscent of night-time skies and constellations, which corresponds to the silver star at [35, 15, 17]. The target placement is the red container at [76, 69, 17]. The plan involves moving the gripper to the star's position, closing the gripper to pick it up ...

**Language plan:**
1. Move to the star's position.
2. Close the gripper to pick up the star ...

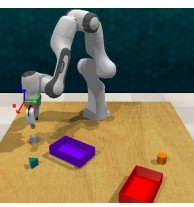

[35, 15, 26, 0, 60, 90, 1]

**Planning Step 5:**
**Visual description:** From left to right, I can see a teal triangular at [57, 18, 17], a violet container at [50, 35, 17], a silver star at [76, 66, 26], a red container at [77, 67, 17], and an orange cylinder at [54, 82, 18].

**Reflection:** The task is to place the silver star into the red container. The previous actions have successfully moved the gripper to the position above the red container with the star securely held. The next step is to open the gripper to release the star into the container. This will complete the task as the star will be placed inside the red container.

**Language plan:**
1. Open the gripper to place the star into the red container.

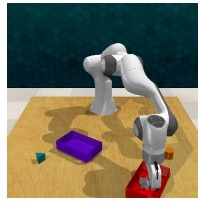

[76, 69, 17, 0, 60, 90, 1]

Figure 8: Successful reflection Example in EB-Manipulation. Both agents were able to identify the silver star as the target object.

**Human Instruction: Put two spray bottles in the cabinet under the sink against the wall.**

**1. EPL only**

**Planning Step 4:**
**Visual description:** The cabinet under the sink is open, and a blue spray bottle is being held above it.

**Reflection:** The first spray bottle has been successfully picked up and the cabinet is open, ready to receive the bottle. The next step is to place the spray bottle inside the cabinet. After placing the first bottle, I need to locate and pick up the second spray bottle to complete the task.

**Language plan:**
1. Put down the spray bottle inside the cabinet.
2. Find the second spray bottle …

put down the object in hand

**Planning Step 5:**
**Visual description:** The cabinet under the sink is open, and one spray bottle is already inside. Another spray bottle is visible on the floor near the cabinet.

**Reflection:** The first spray bottle has been successfully placed inside the cabinet. The next step is to locate the second spray bottle, which is visible on the floor near the cabinet, and pick it up to complete the task.

**Language plan:**
1. Find the SprayBottle on the floor.
2. Pick up the SprayBottle …

find a SprayBottle

**2. ERA (EPL + RL)**

**Planning Step 4:**
**Visual description:** The robot is holding a blue spray bottle in front of an open cabinet under the sink.

**Reflection:** The task is to place two spray bottles in the cabinet under the sink. The first spray bottle has been picked up and the cabinet is open, ready to receive the bottle. The next step is to put the spray bottle into the cabinet. After placing the first bottle, the cabinet should be closed, and the process repeated for the second spray bottle.

**Language plan:**
1. Put down the spray bottle in the cabinet.
2. Close the cabinet …

put down the object in hand

**Planning Step 6:**
**Visual description:** The image shows a cabinet under the sink with a tiled floor. The cabinet doors appear to be closed.

**Reflection:** The previous actions successfully located and placed one spray bottle inside the cabinet under the sink. The cabinet was then closed. To complete the task, we need to find and place a second spray bottle inside the cabinet. The next step is to locate the second spray bottle and repeat the process of opening the cabinet, placing the spray bottle inside, and closing the cabinet.

**Language plan:**
1. Find the second spray bottle …

find a SprayBottle_2

Figure 9: Planning Error Example in EB-ALFRED. ERA successfully identified the second spray bottle as SprayBottle_2 while EPL repeatedly located the same SprayBottle.

**Human Instruction: The room is dark; can you help me turn something on? At the same time, hold a water can.**

### 1. EPL only

**Planning Step 1:**
**Visual description:** The image shows a living room with two green armchairs and a sofa. There is a red pillow on the sofa and a newspaper on it. The room has windows with white frames, and there is a plant visible outside.

**Reflection:** The task requires turning on a light while holding a watering can. The first step is to locate the watering can and pick it up. Then, find a light source and turn it on.

**Language plan:**
1. locate the watering can and pick it up.
2. find a light source and turn it on.

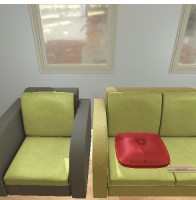

find a WateringCan

**Planning Step 4:**
**Visual description:** The image shows a watering can being held in front of a floor lamp. The lamp is currently off.

**Reflection:** The previous steps involved finding and picking up the watering can, and locating the floor lamp. The next step is to turn on the floor lamp while holding the watering can. The lamp is visible and within reach, so the action to turn it on should be valid.

**Language plan:**
1. Turn on the floor lamp.

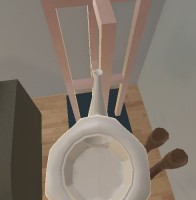

turn on the FloorLamp

### 2. ERA (EPL + RL)

**Planning Step 1:**
**Visual description:** The image shows a living room with two green armchairs and a sofa. On the sofa, there is a red cushion and a newspaper. The room has windows with white frames, and the walls are light gray.

**Reflection:** The task requires turning on a light while holding a watering can. The first step is to locate the watering can and pick it up. Then, find a light source and turn it on.

**Language plan:**
1. locate the watering can and pick it up.
2. find a light source and turn it on.

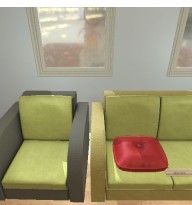

find a WateringCan

**Planning Step 5:**
**Visual description:** The image shows a watering can being held in front of a floor lamp. The lamp is currently off.

**Reflection:** The previous plan involved finding and picking up the watering can, which has been successfully completed. The next step is to turn on the floor lamp while holding the watering can. The floor lamp is visible and within reach, so the action to turn it on should be valid.

**Language plan:**
1. Turn on the FloorLamp while holding the WateringCan.

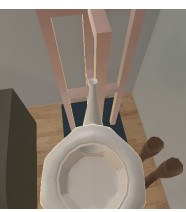

turn on the FloorLamp

Figure 10: Successful reflection Example in EB-ALFRED. Both agents were able to identify the FloorLamp as the target object.

# K  REAL WORLD DEMONSTRATION

We demonstrate that a policy trained entirely in simulation can be directly deployed on a real X-Arm6 without any additional calibration or fine-tuning, as shown in Figure 11, highlighting its strong zero-shot generalization to real-world scenes.

**Human Instruction: Pick up the red cube and place it into the blue container**

**Planning Step 1:**
Visual description: From left to right, I can see a blue container at [7, 26, 13], a red cube at [47, 63, 7], a lime cube at [8, 87, 7], and a rose container at [24, 119, 13].
Reflection:The task is to pick up the red cube located at [47, 63, 7] and place it into the blue container located at [7, 26, 13]. The plan involves moving the gripper to the cube's position, closing the gripper to pick it up, moving to the blue container's position, and opening the gripper to release the cube into the container.
Language plan: 1. Move to the red cube's position at [47, 63, 7]. 2. Close the gripper to pick up the red cube...

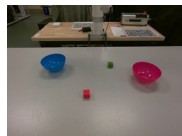

[47, 63, 13, 0, 60, 90, 1]

**Planning Step 1:**
Visual description:From left to right, I can see a blue container at [7, 23, 14], a red cube at [47, 63, 7], and a rose container at [25, 119, 12].
Reflection: The task is to pick up the red cube and place it into the blue container. The first action of moving to the red cube's position has been executed. The next step is to close the gripper to pick up the red cube. After picking it up, the plan involves lifting the cube slightly to avoid any collision with other objects or the table surface. Then, the gripper will move to the blue container's position, and finally, the gripper will open to release the cube into the container. This sequence ensures that the cube is safely transferred from its initial position to the container without any obstruction.
Language plan: ..., 2. Lift the red cube slightly to avoid collision. 3. Move to the blue container's position at [7, 26, 13]...

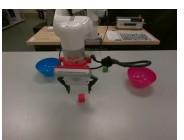

[47, 63, 7, 0, 60, 90, 0]

**Planning Step 1:**
Visual description: From left to right, I can see a blue container at [7, 25, 13], and a rose container at [27, 119, 12].
Reflection: The task is to pick up the red cube and place it into the blue container. The first two actions have been executed: moving to the red cube's position and closing the gripper to pick it up. The next step is to lift the red cube slightly to avoid any collision with other objects or the table surface. After lifting, the gripper will move to the blue container's position, and finally, the gripper will open to release the cube into the container. This sequence ensures that the cube is safely transferred to the container without any obstruction.
Language plan: ..., 2. Move to the blue container's position at [7, 26, 17]. 3. Open the gripper to place the red cube into the blue container....

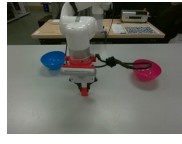

[47, 63, 17, 0, 60, 90, 0]

**Planning Step 1:**
Visual description: From left to right, I can see a blue container at [5, 22, 15], a red cube at [8, 87, 7], and a rose container at [24, 119, 13].
Reflection: The task is to place the red cube into the blue container. The previous actions have successfully picked up the red cube and lifted it to avoid any collision. The next step is to move the gripper, with the cube, to the position above the blue container. Once there, the gripper will open to release the cube into the container. This will complete the task as the cube will be placed inside the blue container.
Language plan: 1. Move to the position above the blue container at [7, 26, 27].

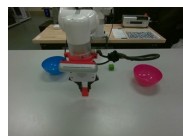

[7, 26, 27, 0, 60, 90, 0]

**Planning Step 1:**
Visual descriptionFrom left to right, I can see a blue container at [3, 25, 12], a red cube at [8, 87, 7], and a rose container at [23, 120, 12].
Reflection: The task is to place the red cube into the blue container. The previous actions have successfully moved the gripper with the red cube to the position above the blue container. The next step is to open the gripper to release the cube into the container. This will complete the task as the cube will be placed inside the blue container. After releasing the cube, the gripper should be moved back to a neutral position to be ready for any subsequent tasks.
Language plan: 1. Open the gripper to place the red cube into the blue container.

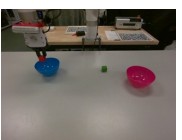

[7, 26, 19, 0, 60, 90, 1]

Figure 11: Examples of ERA performing step-by-step reasoning and actions on EB-Manipulation in real world task without any finetuning.

