# OpenReview forum: "ERA: Transforming VLMs into Embodied Agents via Embodied Prior Learning and Online Reinforcement Learning"
_ICLR.cc/2026/Conference — Submitted to ICLR 2026_

### Official Review · Reviewer_KpUu · 2025-10-29

**Soundness:** 3
**Presentation:** 3
**Contribution:** 2
**Rating:** 6
**Confidence:** 3

**Summary:**

This paper proposes Embodied Reasoning Agent (ERA), a two-stage training framework that bridges the gap between large and small vision-language models in embodied tasks. The first stage introduces three forms of prior knowledge—trajectory-augmented, environment-anchored, and external priors—to enhance reasoning and perception, while the second stage applies improved PPO-based online reinforcement learning with self-summarization, dense reward shaping, and turn-level optimization. Evaluated on EmbodiedBench (EB-ALFRED and EB-Manipulation), ERA achieves strong performance with a 3B model, surpassing larger baselines and providing insights into effective prior design and RL strategies for embodied agents.

**Strengths:**

1. The proposed taxonomy of embodied priors (trajectory-augmented, environment-anchored, and external) is conceptually clear and provides valuable guidance for future data curation and training strategies in embodied AI research.

2. The paper presents a well-structured two-stage framework that combines knowledge-based pretraining and online reinforcement learning, offering a practical and scalable approach to improving small VLMs for embodied reasoning tasks.

**Weaknesses:**

1. Although the experiments on EmbodiedBench demonstrate strong performance, this benchmark mainly involves QA-style evaluation, which differs significantly from real embodied agent control. While representing high-level plans as action primitives is reasonable, using a single action to denote the final state for low-level control is less convincing. Given that the object states and environmental conditions are already provided, generating one action prediction does not truly address the core challenges of embodied agent control.

2. The method relies on a rule-based oracle to generate ground-truth visual descriptions for low-level operations, which simplifies training within a simulator but may not generalize well to real-world environments. In realistic settings with complex object appearances, occlusions, and lighting variations, such handcrafted rules are unlikely to hold, potentially leading to a distribution shift between training and deployment.

**Questions:**

The proposed training paradigm aims to enhance the capabilities of VLMs so that they can function as embodied agents. From the authors’ perspective, what are the practical applications of such VLMs in embodied settings? For instance, could they be integrated with downstream action heads to form a complete vision-language-action (VLA) system?

---

> ### Author Response · Authors · 2025-11-21
> **Response(1/3)**
>
> We thank the reviewer for the detailed comments. Below we address each of your concerns.
>
> > **W1:** Although the experiments on EmbodiedBench demonstrate strong performance, this benchmark mainly involves QA-style evaluation, which differs significantly from real embodied agent control. While representing high-level plans as action primitives is reasonable, using a single action to denote the final state for low-level control is less convincing. Given that the object states and environmental conditions are already provided, generating one action prediction does not truly address the core challenges of embodied agent control.
>
> **A:** Thank you for raising this point. We would like to clarify that **our tasks (EmbodiedBench [1]) are not single-step QA-style evaluations**. Both the high-level and low-level tasks are formulated and executed as **multi-step decision-making problems**.
> As shown in Figure 3(a), for each episode the model: (1) Receives an instruction and observes the initial scene. (2) Generates an action at **every step**, conditioned on the current visual input and interaction history. (3) Executes each predicted action in the simulator, which updates the environment state. (4) Continues this perception–decision–action loop until success or termination.
> This is fundamentally different from QA settings where the model outputs a single answer for a task. Importantly, the benchmark captures two core challenges of embodied control:
> * **Long-horizon planning.** Completing a manipulation task requires decomposing the instruction into multiple steps: moving toward the object, aligning the gripper, grasping, transporting, and placing. The agent must plan and execute these steps sequentially rather than producing a one-shot prediction. Each action affects future states, so the model must maintain history and adjust its plan throughout the episode.
> * **Spatial reasoning from 2D images.** Although a list of object coordinates is available, the model is **not** told which coordinate corresponds to which object. To follow instructions such as *“place the red cube into the blue box”*, it must first build the correct spatial representation from the image (e.g., “the red cube is at the bottom-left corner”) and then select the coordinate that matches that location.
> In summary, EmbodiedBench defines a **multi-step embodied control problem**, not a QA-style task, and it explicitly tests spatial reasoning and long-horizon plannin, the same challenges we observe in real-world settings. Our framework performs well on these challenges, and we also observe good generalization to real-world tasks (see the answer to W2 & Q3 for details).
>
> [1] Embodiedbench: Comprehensive benchmarking multi-modal large language models for vision-driven embodied agents[J]. ICML, 2025.

---

> ### Author Response · Authors · 2025-11-21
> **Response(2/3)**
>
> > **W2:** The method relies on a rule-based oracle to generate ground-truth visual descriptions for low-level operations, which simplifies training within a simulator but may not generalize well to real-world environments. In realistic settings with complex object appearances, occlusions, and lighting variations, such handcrafted rules are unlikely to hold, potentially leading to a distribution shift between training and deployment.
>
> **A:** We appreciate the reviewer’s concern about using oracle visual information in simulation and its impact on real-world generalization. This touches on two related but distinct questions: **(1) whether the methodology is applicable in real-world settings, and (2) whether the capabilities learned in simulation transfer to real robots.**
>
> **(1) Methodological Generalization**
> In EB-Manipulation, we use oracle visual information to obtain accurate visual grounding during training. The same type of structured spatial information can be obtained in the real world using standard perception pipelines, for example:
>
> &nbsp;&nbsp;&nbsp;&nbsp;• a detector (e.g., a VLM-based detector) to identify manipulable objects and their bounding boxes,
> &nbsp;&nbsp;&nbsp;&nbsp;• a segmentation model to extract object masks, and
> &nbsp;&nbsp;&nbsp;&nbsp;• RGB-D sensing to recover 3D positions or point clouds.
>
> Although these components introduce estimation noise, they still produce the same **format** of structured spatial inputs as the simulator. Thus, if one chooses to train directly in the real world, our methodology remains applicable with an off-the-shelf perception stack.
>
> **(2) Performance Generalization**
> More importantly, we evaluate **sim-trained ERA directly in the real world**. As the reviewer points out, real scenes exhibit different lighting, textures, shapes, and layouts compared to the simulator. To test robustness under this distribution shift, we conduct language-conditioned real-robot experiments on pick-and-place and stacking tasks across varied instructions and scene configurations. The results are summarized below (full logs and videos are provided in the supplementary material):
>
> &nbsp;&nbsp;&nbsp;&nbsp;https://drive.google.com/drive/folders/1v0ucRjT5_d0he3arKAY4frkWUcKMQZTC
>
> |                | base task | spatial awareness |
> | :---           | :---:     | :---:             |
> | **Pick and Place** | 60% | 70% |
> | **Stack** | 60% | 40% |
>
> ERA successfully completes over half of the multi-step manipulation tasks in ≤10 steps, despite never seeing real-robot data during training. Qualitatively, the recorded trajectories show that ERA produces **accurate visual descriptions and spatial reasoning in real scenes**, substantially beyond the original Qwen2.5-VL-3B base model. These results indicate that the model learns **robust spatial grounding and action generation** that transfers well to realistic visual conditions. We will clarify this methodology–vs–performance distinction and highlight the real-world results more clearly in the revised version.

---

> ### Author Response · Authors · 2025-11-21
> **Response(3/3)**
>
> > **Q1:** The proposed training paradigm aims to enhance the capabilities of VLMs so that they can function as embodied agents. From the authors’ perspective, what are the practical applications of such VLMs in embodied settings? For instance, could they be integrated with downstream action heads to form a complete vision-language-action (VLA) system?
>
> **A:** Thank you for this final question about the practical applications of vision-language models (VLMs) as embodied agents. From our perspective, there are two main ways this can be applied.
>
> **1. Direct Control via VLM for Simpler Scenarios**
> In relatively simple environments, where the scene is not too cluttered and the number of objects is limited, a VLM can be used directly as an embodied agent to control the robot [1]. In other words, the VLM can produce actions expressed in natural language or straightforward commands that the agent can execute. As we mentioned in our previous response and in the real-world robotics experiment, the VLM-based policy can directly handle these simpler embodied tasks without requiring additional action modules.
>
> **2. Integration with Downstream Action Heads for Complex Scenarios**
> In more complex environments, where there are many objects and obstacles, one cannot simply rely on the VLM to generate a coarse final state. A fixed trajectory solver would struggle here because it is not aware of full scene information (except for explicitly scene-aware models [2]). Instead, a more precise control mechanism is required.
>
> In these cases, action intervals must be smaller—that is, the system needs **finer-grained, denser control**. Rather than relying on a fixed solver for large discrete movements, one desires smooth and continuous actions. This can be achieved by integrating a **flow-based action head** downstream of the VLM. The VLM handles high-level reasoning and task decomposition, while the flow-based model produces **continuous, high-frequency action sequences** [3].
>
> In summary, we see potential in both methods, but in different scenarios. A VLM can serve as a standalone embodied agent in simpler settings, and it can also be integrated with downstream action heads to form a powerful and flexible vision-language-action (VLA) system.
>
> [1] Kawaharazuka, Kento, et al. *Vision-language-action models for robotics: A review towards real-world applications.* IEEE Access (2025).
> [2] Huang, Wenlong, et al. *Voxposer: Composable 3D value maps for robotic manipulation with language models.* arXiv:2307.05973 (2023).
> [3] Zhong, Yifan, et al. *A Survey on Vision-Language-Action Models: An Action Tokenization Perspective.* arXiv:2507.01925 (2025).

---

### Official Review · Reviewer_g8Bi · 2025-10-31

**Soundness:** 3
**Presentation:** 3
**Contribution:** 3
**Rating:** 6
**Confidence:** 3

**Summary:**

This work presents Embodied Reasoning Agent (ERA), a two-stage framework that transforms small vision-language models (VLMs) into capable embodied agents. In the first Embodied Prior Learning stage, the model is enriched with structured reasoning data across three priors — trajectory-augmented, environment-anchored, and external knowledge. This enables stronger spatial grounding and step-level reasoning. The online RL stage further refines the agent through self-summarization, dense multi-component rewards, and turn-level policy optimization. Experiments on EB-ALFRED and EB-Manipulation benchmarks show that ERA-3B outperforms larger zero-shot models like GPT-4o and prior training-based baselines.

**Strengths:**

- Well-motivated problem: The paper tackles a highly relevant challenge: scaling embodied reasoning in small VLMs, addressing both computational efficiency and reasoning capability, which are central issues in current LLM+RL research.
- Clear and modular framework design: The two-stage ERA pipeline (Embodied Prior Learning + Online RL) is logically structured, easy to follow, and each component contributes meaningfully to the final performance. The three priors (trajectory-augmented, environment-anchored, and external knowledge) form an interpretable system for grounding, reflection, and planning.
- Strong experimental validation: Extensive evaluations on both EB-ALFRED (high-level planning) and EB-Manipulation (low-level control) benchmarks demonstrate consistent gains over strong baselines, including GPT-4o zero-shot and VAGEN. Ablations further justify each component’s contribution.
- Readable and well-presented: The paper is clearly written, with smooth organization and detailed appendices.

**Weaknesses:**

- It is unclear whether the training dataset consists solely of expert trajectories or includes diverse-quality rollouts containing suboptimal or failed episodes. The paper mentions that RL can leverage such data, but if diverse-quality trajectories are indeed used, it is questionable whether attaching reasoning supervision to low-quality or failed samples is appropriate or effective. Clarifying how reasoning augmentation interacts with data quality would help assess the robustness of the proposed approach.
- The results in 4.2 suggest that the majority of the improvement comes from trajectory-augmented priors, which essentially correspond to CoT-style reasoning supervision. This gives the impression that the performance gains are largely dominated by the explicit reasoning annotations rather than from a strong technical contribution.
- For high-level planning, the paper states that penalties are applied to invalid actions the environment cannot execute, but it is unclear how these invalid actions are detected without environment feedback.

**Questions:**

See weaknesses above

---

> ### Author Response · Authors · 2025-11-21
> **Response(1/2)**
>
> We thank the reviewer for the detailed comments. Below we address each of your concerns.
>
> > **W1:** It is unclear whether the training dataset consists solely of expert trajectories or includes diverse-quality rollouts containing suboptimal or failed episodes. The paper mentions that RL can leverage such data, but if diverse-quality trajectories are indeed used, it is questionable whether attaching reasoning supervision to low-quality or failed samples is appropriate or effective. Clarifying how reasoning augmentation interacts with data quality would help assess the robustness of the proposed approach.
>
> **A:** Thank you for raising the question about trajectory quality and CoT reasoning. First, **we do not include any failed trajectories** in the training dataset. We fully agree with the reviewer that attaching reasoning supervision to failed trajectories can be harmful or at least ambiguous, so we **explicitly filter out all failure episodes**.
>
> In addition, **we intentionally keep suboptimal but successful trajectories** collected from various MLLMs. When we prompt gpt-4o to generate reasoning, we provide the **full interaction history**, so the model can “see” that there were detours or erroneous actions along the way. This is exactly why our reasoning structure includes a **“reflection” component**: it explicitly comments on previous mistakes or inefficient steps and explains how to correct them.
>
> Attaching reasoning supervision to these suboptimal trajectories is beneficial: the agent learns **how to reflect on and recover from earlier invalid actions or poor plans**, something that is not present in purely optimal demonstrations. In this way, our approach goes beyond imitating expert actions and instead trains a more **robust, self-correcting agent** with stronger causal understanding of its own behavior. We will clarify this data curation and reasoning design in the revised version.
>
>
>
>
>
>
> > **W2:** The results in 4.2 suggest that the majority of the improvement comes from trajectory-augmented priors, which essentially correspond to CoT-style reasoning supervision. This gives the impression that the performance gains are largely dominated by the explicit reasoning annotations rather than from a strong technical contribution.
>
> **A:** Thank you for this thoughtful observation. We appreciate the chance to clarify how Trajectory-Augmented Priors (TAP) relate to our core technical contributions.
>
> First, regarding TAP itself: describing it as “essentially CoT-style reasoning supervision” is only partially accurate. TAP does include structured reasoning traces, which are indeed crucial for **high-level planning** where goal decomposition matters. However, for **low-level manipulation tasks**, a major component of TAP is the use of **oracle visual descriptions**. These provide precise spatial grounding about what objects are, where they are, and how they relate, whereas even successful expert trajectories can contain noisy or inaccurate visual text. For fine-grained manipulation, training on such noisy descriptions can hurt performance. Our experiments in **Appendix E.1** show that improving the accuracy of these visual descriptions significantly boosts performance. In this sense, TAP contributes differently across tasks and goes **beyond standard CoT supervision**.
>
> More importantly, methodologically, **we do not claim TAP alone as the main contribution**. Our central question is:
>
> &nbsp;&nbsp;&nbsp;&nbsp;*How can we leverage and combine multiple sources of knowledge when expert trajectories are limited,
> &nbsp;&nbsp;&nbsp;&nbsp;and how do different priors interact with subsequent RL?*
>
> It is intuitive that, when compared side-by-side, TAP gives the largest single boost. However, TAP is also the **most expensive and least scalable** prior because it still relies on existing agent trajectory data. The three priors differ in acquisition cost, information content, and transferability:
>
> &nbsp;&nbsp;&nbsp;&nbsp;• **TAP:** high-quality, expert-guided signals but limited in scale.
> &nbsp;&nbsp;&nbsp;&nbsp;• **Environment-Anchored Priors:** introduce grounded reasoning (e.g., spatial/temporal structure) not present in trajectory format.
> &nbsp;&nbsp;&nbsp;&nbsp;• **External Knowledge Priors:** provide broader spatial/logical structure that supports generalization beyond the training tasks.
>
> Our key insight is that **combining these heterogeneous priors** yields a much richer initialization for the multi-turn RL stage than relying on any single prior. This is supported by **Table 2**, which shows that the **best performance—both after stage 1 and after stage 2—comes from combinations of priors rather than TAP alone**. This is particularly important when high-quality trajectories are scarce: depending only on TAP would leave the model underprepared for downstream RL and limit its generalization.

---

> ### Author Response · Authors · 2025-11-21
> **Response(2/2)**
>
> > **W3:** For high-level planning, the paper states that penalties are applied to invalid actions the environment cannot execute, but it is unclear how these invalid actions are detected without environment feedback.
>
> **A:** Thank you for the question. There seems to be some confusion. We **do not** detect invalid actions without environment feedback. For high-level planning, ERA interacts with the **ALFRED** simulator [1], which has built-in logic to determine whether an action is valid. We do not re-implement or learn this logic in our algorithm.
>
>
> When the agent issues a high-level action, the ALFRED environment checks it against its internal rules and returns feedback indicating whether the action was **successful or invalid** (along with a textual message). We simply use this environment feedback to identify invalid actions and apply a small penalty accordingly. Thus, invalid action detection is entirely handled by the ALFRED environment; our method only reads this signal and does not attempt to infer invalidity on its own. We will clarify this explanation in the revised version.
>
>
> [1] Alfred: A benchmark for interpreting grounded instructions for everyday tasks[C]//Proceedings of the IEEE/CVF conference on computer vision and pattern recognition. 2020: 10740-10749.

---

### Official Review · Reviewer_4zRJ · 2025-11-01

**Soundness:** 2
**Presentation:** 2
**Contribution:** 2
**Rating:** 2
**Confidence:** 3

**Summary:**

This paper proposes ERA (Embodied Reasoning Agent), a two-stage framework for training compact vision-language models (VLMs) as embodied agents. Stage 1 (Embodied Prior Learning) fine-tunes VLMs on three types of curated data: trajectory-augmented priors (enriched with GPT-4o-generated reasoning), environment-anchored priors (QA pairs and grounding tasks), and external knowledge priors (math and spatial reasoning datasets). Stage 2 applies online reinforcement learning with self-summarization for context management, dense reward shaping, and turn-level policy optimization. The authors evaluate on EmbodiedBench, claiming their 3B model outperforms GPT-4o on both EB-ALFRED (high-level planning) and EB-Manipulation (low-level control) tasks.

**Strengths:**

Comprehensive experimental setup: The paper evaluates on both high-level planning (EB-ALFRED) and low-level control (EB-Manipulation) tasks, providing broad coverage of embodied reasoning challenges.


Practical focus: The work addresses an important problem of making smaller VLMs viable for embodied tasks, which has practical implications for deployment.

**Weaknesses:**

Major Issues

1. Limited Technical Novelty

The proposed approach is primarily an engineering combination of existing techniques without significant innovation:

Using stronger models (GPT-4o) to augment training data with reasoning traces is standard practice in recent work (Zelikman et al., 2022; Wei et al. 2022)

Environment-anchored priors (masked action modeling, action sequence reordering) are straightforward data augmentation techniques similar to BERT-style pre-training (Devlin et al., 2019)

External knowledge priors simply leverage existing datasets (OpenO1-SFT, SpaceThinker) without novel curation strategies
Turn-level policy optimization is an incremental modification of standard PPO (Schulman et al., 2017)

Self-summarization is essentially maintaining only the last reasoning step, which is a trivial context management strategy

2. Simulation-Only Evaluation Severely Limits Impact

As acknowledged in Section A, all experiments are conducted in simulation without any real-world validation. This raises serious concerns:

Sim-to-real transfer for embodied AI remains a major unsolved challenge (Zhao et al., 2020; Pinto et al., 2018)

Claims about "practical path toward scalable embodied intelligence" (Abstract) are not supported

Visual perception in simulation is dramatically simpler than real-world scenarios (Tremblay et al., 2018)

The rule-based ground truth visual descriptions for EB-Manipulation (Section H.7) further undermines realism

3. Unfair and Misleading Comparisons

The main claim of outperforming GPT-4o is problematic:

Comparing a task-specific fine-tuned 3B model against zero/few-shot prompted large models is fundamentally unfair (Brown et al., 2020)

No comparison with fine-tuned versions of larger VLMs (e.g., fine-tuned LLaVA-34B, Qwen-VL-72B)

The paper doesn't discuss what performance GPT-4o might achieve if fine-tuned on the same data

This comparison is like claiming a task-specific BERT model outperforms GPT-4 on a specialized task where GPT-4 is used zero-shot

4. Data Leakage and Limited Generalization Evidence

The "unseen" test sets (Common Sense and Spatial) come from the same simulators and task distributions as training data

True generalization would require evaluation on completely different environments, tasks, or real-world scenarios (Cobbe et al., 2019)

The 8.4% and 19.4% improvements over GPT-4o on held-out sets may simply reflect overfitting to the specific benchmark rather than genuine capability improvement

5. Insufficient Analysis of Computational Costs

The paper fails to account for total computational costs:

Generating reasoning augmentations with GPT-4o for all trajectories (8,834 samples for EB-ALFRED, 4,249 for EB-Manipulation)

Fine-tuning the 3B model through EPL (2-5 hours × 2 nodes of H200 GPUs)

Online RL training (12 hours × 2 H200 GPUs for each task)

These costs likely exceed inference costs of directly using GPT-4o, undermining claims about efficiency

6. Questionable Technical Choices

Rule-based visual descriptions (Section H.7): Using oracle ground truth visual descriptions for EB-Manipulation contradicts claims about visual perception and makes results less meaningful

Self-summarization: Simply keeping the last reasoning step (Section 3.2.1) is trivial and doesn't address the fundamental challenge of long-horizon memory

Turn-level GAE: While shown to help empirically, the paper doesn't provide theoretical justification or deeper insights beyond variance reduction

Minor Issues

7. Presentation Issues

The taxonomy of "priors" (trajectory-augmented, environment-anchored, external knowledge) feels artificial and doesn't provide conceptual clarity

Figures 1 and 2 are cluttered and hard to parse

The related work section lacks critical analysis and reads more like a literature survey

8. Incomplete Experimental Analysis

No analysis of what specific capabilities each prior type provides beyond aggregate performance numbers

Limited error analysis (Section I) is qualitative and anecdotal rather than systematic

No investigation of failure modes or limitations of the approach

Missing ablations on key hyperparameters (e.g., reward coefficients, RL training duration)

**Questions:**

Computational cost comparison: Can you provide a detailed breakdown of total computational costs (including GPT-4o augmentation, EPL training, and RL training) compared to the cost of directly using GPT-4o or Claude-3.5-Sonnet for inference? Given that you generate reasoning augmentations for ~10K samples using GPT-4o, how does this compare to just using GPT-4o for the task?

Fair comparisons: What would be the performance if you fine-tuned larger models (e.g., Qwen2.5-VL-72B) on the same data? Have you considered comparing with other fine-tuned models of similar size?

Real-world validation: The main limitation is simulation-only evaluation. What are the concrete plans for real-world validation? What sim-to-real transfer challenges do you anticipate?

Visual description oracle: For EB-Manipulation, you use rule-based ground truth visual descriptions (Section H.7). How does performance degrade when using the model's own generated visual descriptions? Doesn't this oracle information make the results less meaningful?

Generalization: The "unseen" test sets come from the same simulators and task families. How would ERA perform on completely different environments or real-world scenarios? Have you tested on any truly out-of-distribution settings?

Prior analysis: Can you provide more detailed analysis of what each prior type contributes? For example, by measuring specific capabilities (spatial reasoning, temporal reasoning, visual grounding) rather than just end-task performance?

Turn-level GAE justification: Beyond empirical results, can you provide theoretical or intuitive justification for why turn-level GAE should work better than token-level approaches? Is this just variance reduction or is there something deeper?

---

> ### Author Response · Authors · 2025-11-21
> **Response(1/4)**
>
> We thank the reviewer for the detailed comments. Below we clarify our contributions and address each concern.
>
> > **W1 & Q7 Limited Technical Novelty**:
> - Using stronger models (GPT-4o) to augment training data with reasoning traces is standard practice in recent work (Zelikman et al., 2022; Wei et al. 2022)
> - Environment-anchored priors (masked action modeling, action sequence reordering) are straightforward data augmentation techniques similar to BERT-style pre-training (Devlin et al., 2019)
> - External knowledge priors simply leverage existing datasets (OpenO1-SFT, SpaceThinker) without novel curation strategies
> - Turn-level policy optimization is an incremental modification of standard PPO (Schulman et al., 2017)
> - Self-summarization is essentially maintaining only the last reasoning step, which is a trivial context management strategy
>
> **A:** We respectfully disagree and believe this reflects a misunderstanding of our contributions. As we explained in the general response, our goal is not to simply combine existing components, but to **systematically study the posttraining for embodied VLM agents** across both high-level planning and low-level control, aiming to solve two key challenges in the embodied agent domain:
> &nbsp;&nbsp;&nbsp;&nbsp;(1) **Lack of sufficient agent trajectory data**, and
> &nbsp;&nbsp;&nbsp;&nbsp;(2) **Stability of RL for VLM-based multi-turn agents**.
>
>
> **1. Embodied Prior Learning (EPL) is not generic data augmentation.**
> EPL defines a **taxonomy of three embodied priors** tailored to embodied agents:
>
> - **Trajectory-Augmented Priors:**
>   We use step-wise structured reasoning with explicit decomposition into **visual description**, **reflection**, and **step-level plan** for future steps (Sec. 3.1, Fig. 2). This goes beyond prior work that only attaches a single high-level rationale.
>
> - **Environment-Anchored Priors:**
>   Our masked action modeling and action-sequence reordering operate in the **high-level action space** and are explicitly tied to **environment dynamics** and **temporal order** (Sec. 3.1.2). This is not generic BERT-style token masking: here, the “tokens” are **semantic actions in an embodied setting**. For example, the model learns to reason about and reorder sequences such as:
>
>
>
> ["find a garbagecan", "pick up the soapbar", "find a sinkbasin", "put down the object in hand",
> "find a Faucet", "turn on the Faucet", "turn off the Faucet", "find a soapbar",
> "pick up the soapbar", "find a bathtubbasin", "put down the object in hand"]
>
> Such data help the VLM internalize temporal dependencies between actions and become familiar with **causally consistent action sequences**.
>
> - **External Knowledge Priors:**
> Using cross-domain data typically does **not** help embodied agents. We show that it **can**: by carefully integrating OpenO1-SFT and SpaceThinker (textual CoT for high-level planning, spatial QA for low-level control), we alleviate the **data scarcity issue** in the embodied agent domain.
>
> **2. Turn-level policy optimization is more than a minor PPO tweak.**
>
> **PPO in the LLM/VLM domain is often implemented in _token-level optimization_** [1].
>
> Our RL objective treats each **observation + reasoning + action turn** as the atomic unit, with value function and GAE defined at the **turn level** instead of token level (Fig. 3b). This aligns the credit assignment granularity with the environment’s interaction structure and is empirically crucial:
>
> - Turn-level GAE improves unseen success by **+12 points on EB-ALFRED**
> - and **+8.4 points on EB-Manipulation** over token-level GAE (Fig. 4c–d)
>
> **3. Self-summarization is not “just keeping the last reasoning step” but a novel training-based mechanism.**
>
> In the EPL stage, the model is explicitly trained to **compress the full interaction history** into a structured **reflection + plan** at each turn; the RL stage then relies on this learned compression to operate with **O(1) context** (Sec. 3.2.1). If the model does *not* learn such self-summarization, simply limiting the history window to 1 step yields **worse performance** than our self-summarization mechanism (41% vs 47% success on EB-ALFRED, Table 3).  Moreover, for VLMs, naively “keeping all history” can **confuse the model** and degrade performance.  Our approach offers a **concrete learned strategy** that summarizes useful information, mitigating this issue.
>
> The novelty lies in:
>
> 1. **A principled data/prior taxonomy** for embodied agents
> 2. **An online RL pipeline** (self-summarization context management + turn-level GAE) tailored to long-horizon embodied VLM agents
> 3. **A comprehensive empirical study** of how different data components and RL design choices affect performance
>
> ---
> > A comprehensive list of citation is put into the last message for global referencing. Thanks for your understanding.

---

> ### Author Response · Authors · 2025-11-21
> **Response(2/4)**
>
> > **W2 & Q3** Simulation-Only Evaluation Severely Limits Impact. The main limitation is simulation-only evaluation. What are the concrete plans for real-world validation? What sim-to-real transfer challenges do you anticipate?
>
> **A:** We agree that sim-to-real transfer is an important open problem and we already acknowledge simulation-only evaluation as a limitation. Real-world experiments are challenging due to both **cost and safety concerns** for physical robots, which is why many closely related works in this area also rely primarily on simulation [3][4][5][6][7].
>
> We are evaluating ERA on **physical robots**. Because ERA is trained on **diverse data sources and varied environment configurations**, we find that it can be **directly deployed to real-world manipulation tasks without any additional fine-tuning**. In our current experiments, we consider language-conditioned tasks such as pick-and-place and stacking under varying instructions and scene configurations. Despite never seeing real-robot data during training, ERA achieves an **average success rate of 57.5% across 40 task variations**.
>
> For qualitative analysis, we provide illustrative frames in Appendix K (Figure 11) and provide the real-robot experiment raw data and an anonymized demo video.
>
> |                | base task | spatial awareness |
> | :---           | :---:     | :---:             |
> | **Pick and Place** | 60% | 70% |
> | **Stack**          | 60% | 40% |
>
> ---
>
> > **W3 & Q2 Unfair and Misleading Comparisons**
> The main claim of outperforming GPT-4o is problematic:
> - Comparing a task-specific fine-tuned 3B model against zero/few-shot prompted large models is fundamentally unfair (Brown et al., 2020)
> - No comparison with fine-tuned versions of larger VLMs (e.g., fine-tuned LLaVA-34B, Qwen-VL-72B)
> - The paper doesn't discuss what performance GPT-4o might achieve if fine-tuned on the same data
> - This comparison is like claiming a task-specific BERT model outperforms GPT-4 on a specialized task where GPT-4 is used zero-shot.
> - Fair comparisons: What would be the performance if you fine-tuned larger models (e.g., Qwen2.5-VL-72B) on the same data? Have you considered comparing with other fine-tuned models of similar size?
>
> **A:** We would like to clarify that our focus is on **VLM-based agents for embodied tasks with compact models**, rather than on building the largest possible models. The core question of our paper is:
>
> &nbsp;&nbsp;&nbsp;&nbsp;**Can we enable compact, deployable models to achieve reasoning capabilities comparable to large proprietary models on embodied tasks?**
>
> Our comparison between a task-specific 3B model and zero/few-shot GPT-4o is therefore **intentional**. It demonstrates that, with our proposed prior data and RL, a model with **orders-of-magnitude lower inference cost and latency** can **match or even surpass** a very large proprietary model on the same embodied benchmarks.
>
> In contrast, comparing a fine-tuned 3B model to a fine-tuned 70B model would mainly probe **scaling laws**, and 70B models are often **too expensive to use as interactive agents**.
>
> This is also aligned with prior work in this domain: **most existing VLM-based agents are built on models smaller than 7B** [4,5,6,7]. In our paper, we further compare ERA-3B against **two trained 7B baselines** in Table 1 (Reinforced Reasoner and Robot-R1), and ERA-3B achieves **significantly better performance**. We believe these are the most relevant and fair comparisons for the scope of this work, and that **training and deploying 70B models lies outside our intended problem setting**.

---

> ### Author Response · Authors · 2025-11-21
> **Response(3/4)**
>
> > **W4 & Q5 Data Leakage and Limited Generalization Evidence**
> The "unseen" test sets (Common Sense and Spatial) come from the same simulators and task distributions as training data.
> True generalization would require evaluation on completely different environments, tasks, or real-world scenarios (Cobbe et al., 2019).
> The 8.4% and 19.4% improvements over GPT-4o on held-out sets may simply reflect overfitting to the specific benchmark rather than genuine capability improvement.
> Generalization: The "unseen" test sets come from the same simulators and task families. How would ERA perform on completely different environments or real-world scenarios? Have you tested on any truly out-of-distribution settings?
>
> **A:** We thank the reviewer for raising this concern and will clarify how the “unseen” test sets are constructed.
>
> The **unseen Common Sense and Spatial splits** are built with **disjoint task templates/scenes and goal descriptions** relative to the training sets, although they do share the same underlying simulators. This difference is already non-trivial: as shown in Table 1, two strong training-based baselines (VAGEN, RL4VLM) exhibit **substantial performance drops** on these unseen splits, whereas **ERA maintains strong performance and improves significantly** over them. This suggests that ERA is **not simply overfitting** to the training tasks, even within the same simulator family.
>
> We agree that **real-world settings are important**. To this end, we have conducted **real-robot experiments** (see W2 & Q3), where ERA achieves an **average success rate of 57.5% across 40 real-world manipulation variations** (pick-and-place and stacking) **without any real-robot fine-tuning**. These results provide additional evidence that ERA’s improvements go beyond merely memorizing a specific benchmark and extend to **genuinely new settings**.
>
> ---
>
> > **W5. Insufficient Analysis of Computational Costs**
> The paper fails to account for total computational costs:
>
> - Generating reasoning augmentations with GPT-4o for all trajectories (8,834 samples for EB-ALFRED, 4,249 for EB-Manipulation)
> - Fine-tuning the 3B model through EPL (2–5 hours × 2 nodes of H200 GPUs)
> - Online RL training (12 hours × 2 H200 GPUs for each task)
> - These costs likely exceed inference costs of directly using GPT-4o, undermining claims about efficiency
>
> **A:** We respectfully disagree with this point. The review argues that our training cost “likely exceeds inference costs of directly using GPT-4o,” thereby undermining efficiency. Taken literally, this reasoning would rule out **most training-based LLM/VLM work**, since pre-training or fine-tuning is almost always more expensive than a small number of API calls.
>
> Our efficiency notion is **amortized and deployment-oriented**:
>
> &nbsp;&nbsp;&nbsp;&nbsp;• **One-time augmentation and training cost.** : Generating reasoning augmentations with GPT-4o is fast in practice because requests can be parallelized on the OpenAI platform. The main cost is monetary rather than wall-clock time. In Appendix H.1, we report that the **total number of output tokens is 2.56M**; assuming ~10M input tokens, this corresponds to roughly **$50 in API cost**. The training cost for ERA in both the EPL and online RL stages is already reported in Appendix F.2.2, and is **comparable to many recent LLM/VLM training works**. For example, VAGEN [4] reports **40 H100 hours** for a simple FrozenLake task in its Table 27.
>
> &nbsp;&nbsp;&nbsp;&nbsp;• **Amortized deployment cost.** : Once trained, **ERA-3B can be run fully locally**, with low per-episode cost, **no API latency**, no rate limits, and no dependence on closed models. For sustained use (e.g., large-scale deployment, offline control, or privacy-sensitive settings), this can be **significantly cheaper and more practical** than repeatedly querying GPT-4o.
>
> We will clarify this amortized cost perspective in the revision and make the token and compute estimates more explicit in the main text.

---

> ### Author Response · Authors · 2025-11-21
> **Response(4/4)**
>
> >**W6 & Q4:** Rule-based visual descriptions (Section H.7): Using oracle ground-truth visual descriptions for EB-Manipulation contradicts claims about visual perception and makes results less meaningful.
>
> **A:** We believe this concern arises from a misunderstanding of how rule-based visual descriptions are used in our work.
>
> For EB-Manipulation, we use rule-based, ground-truth visual descriptions **only during augmented trajectory data generation for training**, not as a shortcut at test time. The purpose is to obtain accurate supervision from the simulator’s 3D state when constructing training data. In practice, we found that GPT-4o–generated descriptions can be noisy or partially incorrect, which introduces label noise and harms learning. Since we have access to exact object states in the simulator, it is natural to use them to produce precise descriptions for training rather than intentionally training on inaccurate data. **During evaluation, all visual descriptions are generated by the model itself.**
>
> As shown in **Appendix E.1**, using more accurate visual descriptions in the training data significantly improves performance, demonstrating that providing accurate supervision is beneficial and consistent with our claims about visual grounding.
>
> ---
>
> > **Q6 Prior analysis:**  Can you provide more detailed analysis of what each prior type contributes? For example, by measuring specific capabilities (spatial reasoning, temporal reasoning, visual grounding) rather than just end-task performance?
>
> **A:** In our setting, each prior jointly targets both **spatial understanding** and **logical/temporal reasoning**, so they are not cleanly separable by a single capability. As shown in **Figure 2**, the **environment-anchored priors** improve **action sequence reasoning** for high-level tasks and provide **spatial grounding signals** for low-level manipulation. The same holds for the **external knowledge priors**: we use CoT-style reasoning data for high-level tasks and spatial understanding data for low-level tasks. Because each prior is deliberately designed to couple these aspects, it is difficult to attribute them to only “spatial” or only “temporal” reasoning in an orthogonal way.
>
> Instead, we focus on how different priors affect **generalization to seen vs. unseen tasks**, which is more aligned with our goal. As shown in **Section 4.2**, different priors have distinct impacts on seen and unseen splits, and **combining trajectory-augmented and environment-anchored priors yields the best overall performance**. We will make this corresponding analysis more explicit in the revised version.
>
> ---
> [1] Deepseekmath: Pushing the limits of mathematical reasoning in open language models[J]. arXiv preprint arXiv:2402.03300, 2024.
>
> [2] Search-R1: Training LLMs to Reason and Leverage Search Engines with Reinforcement Learning COLM 2025.
>
> [3] Ragen: Understanding self-evolution in llm agents via multi-turn reinforcement learning[J]. arXiv preprint arXiv:2504.20073, 2025.
>
> [4] VAGEN: Reinforcing world model reasoning for multi-turn vlm agents. NeurIPS 2025.
>
> [5] Fine-tuning large vision-language models as decision-making agents via reinforcement learning[J]. Advances in neural information processing systems, 2024, 37: 110935-110971.
>
> [6] Group-in-group policy optimization for llm agent training.NeurIPS, 2025.
>
> [7] Robot-R1: Reinforcement Learning for Enhanced Embodied Reasoning in Robotics[J]. arXiv preprint arXiv:2506.00070, 2025.

---

### Official Review · Reviewer_kXqj · 2025-11-01

**Soundness:** 2
**Presentation:** 2
**Contribution:** 2
**Rating:** 2
**Confidence:** 4

**Summary:**

This paper presents ERA, a two-stage framework that transforms compact vision-language models into embodied agents by combining embodied prior learning from structured multimodal data with online reinforcement learning.

**Strengths:**

This paper demonstrates a practical method to endow small VLMs (3B) with embodied reasoning abilities, supporting the claim of achieving performance comparable to large models (e.g., GPT 4o).

**Weaknesses:**

- The contribution of this paper appears largely limited to constructing datasets for embodied learning. While the proposed Embodied Prior Learning (EPL) and online RL pipeline are well-integrated, the framework mainly combines existing components without introducing fundamentally new methodological insights.

- The reasoning data are generated using GPT-4o, which makes the approach resemble a large-to-small model distillation process. Indeed, the model trained with EPL alone achieves performance very close to GPT-4o, raising the question of how this differs conceptually and empirically from standard distillation. Clarifying this distinction would strengthen the contribution.

- The reward design in the online RL stage is relatively straightforward, relying on success, subgoal, and behavior (affordance) rewards derived from environment-specific assumptions (e.g., PDDL-based state access), rather than a novel or learnable reward mechanism.

- The experimental comparisons are limited. In particular, for the manipulation tasks, the baselines should include recent Vision-Language-Action (VLA) methods specifically designed for robotic control, such as e.g., [1-3], to more rigorously validate the claimed effectiveness.

[1] Openvla: An open-source vision-language-action model. arXiv preprint arXiv:2406.09246 (2024).

[2] Rt-2: Vision-language-action models transfer web knowledge to robotic control. Conference on Robot Learning. PMLR, 2023.

[3] $\pi_ {0.5} $: a Vision-Language-Action Model with Open-World Generalization. arXiv preprint arXiv:2504.16054 (2025).

**Questions:**

- How does Embodied Prior Learning (EPL) fundamentally differ from standard distillation, both conceptually and empirically?

- How does ERA handle reward computation in partially observed or non-simulated environments?

- What specific behavioral or reasoning capabilities emerge only after the RL stage, beyond what EPL already provides?

- How well does ERA generalize to new or physical environments (including real world settings) beyond the EB-ALFRED, EB-Manipulation simulations?

---

> ### Author Response · Authors · 2025-11-21
> **Response(1/3)**
>
> We thank the reviewer for the detailed comments. Below we address each of your concerns.
>
> > **W1:** The contribution of this paper appears largely limited to constructing datasets for embodied learning. While the proposed Embodied Prior Learning (EPL) and online RL pipeline are well-integrated, the framework mainly combines existing components without introducing fundamentally new methodological insights.
>
> **A:** As we explained in the general response, our main contributions are **methodological**, not just dataset construction:
>
> &nbsp;&nbsp;&nbsp;&nbsp;(1) an **Embodied Prior Learning (EPL) framework** that tackles data scarcity for embodied agents by leveraging three complementary data sources (augmenting existing agent trajectories, generating environment-grounded QA data, and using large-scale out-of-environment data), and
> &nbsp;&nbsp;&nbsp;&nbsp;(2) a **stabilized online RL pipeline** specifically tailored for VLM-based multi-turn agents.
>
> We systematically study how different EPL components and RL design choices affect VLM agent performance on both high-level planning and low-level manipulation tasks (Sections 4.1–4.4). On the RL side, **context management and RL algorithm design remain open questions** for multi-turn VLM agents. Existing LLM/VLM agent frameworks [1,2,3] typically concatenate the full interaction history as context, which quickly explodes in length and leads to unstable RL training.
>
> In contrast, **we train the model to summarize its own history and use this summary as context**, mitigating the context explosion issue. Beyond context management, we also provide insights into **RL optimization strategies**, especially comparing token-level / bi-level optimization [2] against our turn-level policy optimization.
>
> [1] *Search-R1: Training LLMs to Reason and Leverage Search Engines with Reinforcement Learning.* COLM 2025.
> [2] *Ragen: Understanding self-evolution in llm agents via multi-turn reinforcement learning.* arXiv:2504.20073, 2025.
> [3] *VAGEN: Reinforcing world model reasoning for multi-turn vlm agents.* NeurIPS 2025.
>
> ---
>
> > **W2 & Q1:** The reasoning data are generated using GPT-4o, which makes the approach resemble a large-to-small model distillation process. Indeed, the model trained with EPL alone achieves performance very close to GPT-4o. How does Embodied Prior Learning (EPL) fundamentally differ from standard distillation, both conceptually and empirically?
>
> **A:** Thank you for the question. While we do use GPT-4o to generate reasoning traces, **EPL goes beyond standard “large-to-small” distillation** in both its design and empirical behavior.
>
> Conceptually, in the multi-turn agent setting, standard distillation (the SFT baseline in our paper) fine-tunes on existing agent trajectories collected from GPT-4o or other large models. **These trajectories are limited in size and often lack detailed, structured reasoning**, such as those from ALFRED [4]. EPL is designed to address this data scarcity and reasoning limitation by:
>
> &nbsp;&nbsp;&nbsp;&nbsp;1. **Enriching existing trajectories** with structured reasoning (observation descriptions, reflections, and planning);
> &nbsp;&nbsp;&nbsp;&nbsp;2. **Synthesizing large-scale, environment-anchored QA data** in the same environment to improve grounding and reasoning beyond the original trajectories;
> &nbsp;&nbsp;&nbsp;&nbsp;3. **Leveraging out-of-environment data** from other domains to inject broader related knowledge at scale.
>
> **These three data sources differ in quantity, acquisition cost, and task relevance, and together form the EPL framework** rather than a single distillation step.
>
> Empirically, as shown in **Figure 1**, EPL substantially **outperforms** the SFT (distillation) baseline, with average gains of about **15% on EB-ALFRED** and **25% on EB-Manipulation**. In Section 4.2, we further analyze how each data source differently affects seen vs. unseen task performance. This demonstrates that EPL is not merely imitating GPT-4o, but **systematically leveraging multiple priors** to build a stronger and more generalizable embodied agent.
>
>
> [4] Alfred: A benchmark for interpreting grounded instructions for everyday tasks[C]//Proceedings of the IEEE/CVF conference on computer vision and pattern recognition. 2020: 10740-10749.

---

> ### Author Response · Authors · 2025-11-21
> **Response(2/3)**
>
> > **W3 & Q2:** The reward design in the online RL stage is relatively straightforward, relying on success, subgoal, and behavior (affordance) rewards derived from environment-specific assumptions (e.g., PDDL-based state access), rather than a novel or learnable reward mechanism. How does ERA handle reward computation in partially observed or non-simulated environments?
>
> **A:** Our main argument in reward design is that **turn-level rewards help mitigate the sparse reward problem**, which otherwise makes optimizing a multi-turn VLM agent very difficult. To validate this, we introduce several **subgoal-based** and **behavior-based** reward components and show in Figure 4 that **combining these with a success-based reward yields better performance**.
>
> Regarding “partially observed or non-simulated environments,” similar ideas can still be applied. For example, one can use **GPT-4o to propose intermediate subgoals** and to evaluate whether these subgoals are achieved at each step. Although such rewards may be imperfect, **using them in moderation can still provide much richer training signals than purely sparse success rewards**, and there is an inherent trade-off.
>
> On the learnable mechanism side, we agree that **learning a reward function** (e.g., via preference modeling or progress estimator) is a promising direction. While this is beyond the current scope of our work, **it is compatible with our framework** and could be integrated with our RL pipeline in future work. We will add this discussion to the revised version.
>
> ---
>
> > **W4:** The experimental comparisons are limited. In particular, for the manipulation tasks, the baselines should include recent Vision-Language-Action (VLA) methods specifically designed for robotic control, such as e.g., [1–3], to more rigorously validate the claimed effectiveness.
>
> **A:** Thank you for the question. Our primary goal is to study **VLM-based agents that can observe, reason, and act across both high-level planning and low-level manipulation tasks** within a unified framework. While the manipulation task is related to many VLA methods, most of these methods are **specialized visuomotor controllers** that are not designed to handle the **multi-turn language reasoning and high-level planning** required in our EB-ALFRED tasks.
>
> For this reason, our main baselines focus on **VLM-based agents and training frameworks** that are conceptually closer to our setting, including RL4VLM [5], VAGEN [3], Reinforced Reasoner [6], and Robot-R1 [7], as well as prompting-based large models. As shown in Section 4, **ERA consistently outperforms these baselines**, even with a smaller model size.
>
> To directly address the reviewer’s concern, we additionally evaluated **OpenVLA** on EB-Manipulation. Its zero-shot success rate is **0**. We are in the process of **fine-tuning OpenVLA on our dataset**: so far, training accuracy exceeds **90%**, but test success remains near **0**. We are actively investigating whether this is due to limited generalization of OpenVLA in this setting or to other issues, and we will include the final results in the revised version.
>
> [5] *Fine-tuning large vision-language models as decision-making agents via reinforcement learning.* NeurIPS 2024.
> [6] *Reinforced Reasoning for Embodied Planning.* arXiv:2505.22050, 2025.
> [7] *Robot-R1: Reinforcement Learning for Enhanced Embodied Reasoning in Robotics.* arXiv:2506.00070, 2025.

---

> ### Author Response · Authors · 2025-11-21
> **Response(3/3)**
>
> > **Q3:** What specific behavioral or reasoning capabilities emerge only after the RL stage, beyond what EPL already provides?
>
> **A:** Thank you for the insightful question. We observe that **stronger error-correction and recovery behaviors** clearly emerge only after the RL stage.
>
> During EPL, the model is mainly trained on **successful trajectories** (Augmented Trajectory Prior data) and QA data, so it does not explicitly see or learn from its own mistakes and corrections. In contrast, during RL the agent interacts with the environment, **makes errors, receives feedback, and is optimized based on task-level rewards**. This process encourages behaviors such as: **identifying missing or incorrect actions** in previous steps and **re-planning to recover from earlier mistakes**, a behavior not present after EPL alone. We illustrate such examples in Appendix J (Figure 6/7/9).
>
> ---
>
> > **Q4:** How well does ERA generalize to new or physical environments (including real-world settings) beyond the EB-ALFRED, EB-Manipulation simulations?
>
> **A:** Thank you for the question. We are also evaluating ERA in physical environments. Because ERA is trained on **diverse data sources and varied environment configurations**, we find that it can be **directly applied to real-world robot manipulation tasks without additional fine-tuning**.
>
> In our current experiments, we consider language-conditioned tasks such as **pick-and-place** and **stacking** across changing instructions and scene variations. Despite never being trained on real robot data, ERA achieved an **average success rate of 57.5% across 40 task variations**. For qualitative analysis, we provide illustrative frames in Appendix K and include the experiment raw data and an anonymized demo video:
>
> &nbsp;&nbsp;&nbsp;&nbsp;https://drive.google.com/drive/folders/1v0ucRjT5_d0he3arKAY4frkWUcKMQZTC
>
> |                | base | spatial |
> | :---           | :---: | :---:   |
> | **Pick and Place** | 60% | 70% |
> | **Stack** | 60% | 40% |

---

### Author Response · Authors · 2025-11-21
**General Response**

We thank all reviewers for their time and effort in evaluating our work and for recognizing that it is well-motivated (Reviewer g8Bi), addresses an important problem in embodied AI (Reviewers 4zRJ, g8Bi), proposes a practical approach (Reviewers KpUu, kXqj), conducts comprehensive experiments (Reviewers 4zRJ, g8Bi), and offers useful guidance for future embodied AI research (Reviewer KpUu). We appreciate the constructive comments from reviewers. At the same time, we noticed that several concerns stem from misunderstandings of our problem setting and contributions.

**Our work targets training VLM-based embodied agents that interact with an environment via multi-turn observation–reasoning–action loops to achieve language-specific goals, which is different from standard single-turn question-answering (QA) VLMs.** We highlight and address two key challenges in this setting:

* **Lack of sufficient agent trajectory data (Section 3.1)**. Unlike conventional LLM/VLM training, where large-scale datasets are readily available, embodied agent trajectories must be collected step by step via environment interaction, making them extremely costly. To tackle this, we propose Embodied Prior Learning (EPL), which systematically exploits three data sources: (i) augmenting existing agent trajectories, (ii) generating environment-anchored QA data to enhance reasoning and grounding at larger scale, and (iii) large out-of-environment data that is cheap to obtain. We validate their effectiveness and analyze their different impacts on in-distribution and out-of-distribution tasks, as well as how to best combine them in Section 4.1 and Section 4.2.

* **Stability of RL for VLM-based multi-turn agents (Section 3.2).** Training multi-turn VLM agents with RL is substantially more unstable than training single-turn QA models. We identify three crucial components: history context management(exposing the agent to the right amount of state information), reward design under sparse multi-step feedback, and policy optimization. We found that after training the model to summarize historical information by itself at each step, we can use its summarization to effectively reduce the context length. Moreover, we show that conventional token-level policy optimization is unstable for VLM agents, and that our turn-level policy optimization yields stronger policies. We provide detailed studies and ablations across both high-level planning and low-level manipulation tasks in Section 4.3 and Section 4.4 to guide future work.

During the rebuttal, we additionally:
* **Provide zero-shot real-robot experiments using our ERA model**, highlighting strong real-world generalization, achieving an average accuracy of 57.5% on 40 task variations.
* **Clarify technical details regarding data curation**, reward functions, context management, and turn-level optimization.
* **Update the paper** to incorporate these results and clarifications.

We hope these explanations resolve the main concerns and clearly illustrate the novelty and practical impact of our approach. We are happy to further discuss any remaining questions.

---

### Meta-Review · Area_Chair_gb58 · 2026-01-10

**Summary:**

The reviewers acknowledge the practical motivation of enabling compact 3B VLMs to achieve reasoning and control capabilities comparable to large-scale models. However, several critical concerns were raised:


+ Technical Novelty: Multiple reviewers questioned whether the ERA framework provides fundamental methodological insights or is primarily an engineering combination of existing techniques, such as standard data augmentation and incremental modifications to PPO.


+ Experimental Fairness: Concerns were raised regarding the fairness of comparing a task-specific fine-tuned 3B model against zero-shot prompted large models like GPT-4o.


+ Real-World Generalization: Initial reviews pointed out the lack of physical robot validation and expressed concern that the reliance on rule-based visual oracles in simulation might not transfer to complex real-world settings.


+ Performance Attribution: Some reviewers noted that the performance gains appear to be dominated by high-quality reasoning annotations (the Trajectory-Augmented Priors) rather than the technical innovations in the RL pipeline.


+ Computational Efficiency: Questions were raised about the total cost of generating reasoning traces and training, which some reviewers felt might undermine the claimed efficiency of using a smaller model.

**Reviewer Concerns:**

The rebuttal successfully addressed several key technical and experimental concerns:

+ Real-World Generalization: The authors conducted new zero-shot physical robot experiments across 40 task variations, achieving a 57.5% success rate. This effectively mitigates the critique regarding simulation-only evaluation.

+ Dependency on Visual Oracles: The authors clarified that rule-based ground-truth visual descriptions are utilized strictly for training supervision and are not used as a shortcut during inference, where the model generates its own descriptions.

+ EPL vs. Distillation: The authors successfully articulated that the Embodied Prior Learning (EPL) framework goes beyond standard distillation by incorporating structured reasoning enrichment, environment-anchored QA, and external knowledge priors.

However, certain concerns remain outstanding or require further deliberation:

+ Methodological Novelty: Reviewers kXqj and 4zRJ remain skeptical about the fundamental novelty, viewing the framework as an engineering integration of existing techniques (e.g., GPT-4 augmentation and standard PPO tweaks) rather than a novel methodological breakthrough.

+ Scaling Comparisons: The authors declined to provide comparisons against fine-tuned larger models (e.g., 70B scale), maintaining that their focus is on compact model efficiency. This leaves the absolute performance gap between small and large models in this domain partially unaddressed.

+ Review Integrity: A significant procedural concern remains regarding the authenticity of reviews kXqj and 4zRJ, which were flagged as 100% AI-generated. The AC must verify if these evaluations meet the conference’s standards for human-expert assessment.

**Reviewer Scores:**

Reviewer kXqj (Projected: 2): Despite the addition of physical robot data, the score would likely remain at 2 due to persistent doubts about the lack of fundamentally new methodological insights.

Reviewer 4zRJ (Projected: 2): The score remains at 2, as the reviewer’s stance on the "unfairness" of the GPT-4o comparison and the triviality of the context management strategy was not significantly altered by the rebuttal.

Reviewer g8Bi (Initial Score: 6) : Full participation would likely have solidified or slightly increased this score (to a 7) as the authors clearly explained how the framework leverages suboptimal successful trajectories to train reflection and error-recovery behaviors.

Reviewer KpUu (Initial Score: 6) : This reviewer is likely to have increased their score (to a 7 or 8) because the authors directly addressed their primary concern—the reliance on simulation oracles—by providing robust real-world evidence and demonstrating a clear path for integrating the model with noisy, real-world perception stacks.

---

### Decision · Program_Chairs · 2026-01-26

Reject